# The *HASTER* lncRNA promoter is a *cis*-acting transcriptional stabilizer of *HNF1A*

Anthony Beucher [1,2] ✉, Irene Miguel-Escalada [1,2,3], Diego Balboa [2,3], Matías G. De Vas[1], Miguel Angel Maestro[2,3], Javier Garcia-Hurtado[2,3], Aina Bernal [2,3], Roser Gonzalez-Franco[1], Pierfrancesco Vargiu[4], Holger Heyn [2,5,6], Philippe Ravassard[7], Sagrario Ortega[4] and Jorge Ferrer [1,2,3] ✉

The biological purpose of long non-coding RNAs (lncRNAs) is poorly understood. Haploinsufficient mutations in HNF1A homeobox A (*HNF1A*), encoding a homeodomain transcription factor, cause diabetes mellitus. Here, we examine *HASTER*, the promoter of an lncRNA antisense to *HNF1A*. Using mouse and human models, we show that *HASTER* maintains cell-specific physiological HNF1A concentrations through positive and negative feedback loops. Pancreatic β cells from *Haster* mutant mice consequently showed variegated HNF1A silencing or overexpression, resulting in hyperglycaemia. *HASTER*-dependent negative feedback was essential to prevent HNF1A binding to inappropriate genomic regions. We demonstrate that the *HASTER* promoter DNA, rather than the lncRNA, modulates *HNF1A* promoter–enhancer interactions in *cis* and thereby regulates *HNF1A* transcription. Our studies expose a *cis*-regulatory element that is unlike classic enhancers or silencers, it stabilizes the transcription of its target gene and ensures the fidelity of a cell-specific transcription factor program. They also show that disruption of a mammalian lncRNA promoter can cause diabetes mellitus.

The transcription of genes is controlled by *cis*-acting promoter and enhancer sequences, many of which harbour disease variants. Mammalian genomes also contain >20,000 long non-coding RNAs (lncRNAs)[1,2]. Although the function of most lncRNAs has not been explored, some lncRNAs are known to regulate gene transcription[3,4]. A considerable number of lncRNAs are transcribed from evolutionarily conserved promoters located near genes encoding lineage-specific regulators[3,5–7], suggesting a *cis*-regulatory function. For some lncRNAs, knockdown experiments have revealed transcriptional effects on nearby genes[8–10], while genetic studies have demonstrated bona fide *cis*-regulatory

functions of selected lncRNAs[3,11–16]. There are nevertheless still major gaps in our understanding of the regulatory purpose of *cis*-acting lncRNAs and how they are fundamentally different from more established gene regulatory elements. Furthermore, the extent to which genetic disruption of *cis*-regulatory lncRNAs can lead to physiologically relevant phenotypes is unclear.

In this study, we examined *HASTER*, the promoter of an lncRNA at the HNF1A homeobox A (*HNF1A*) locus. Mutations in *HNF1A*, encoding a homeodomain transcription factor[17], cause maturity-onset diabetes of the young type 3, the most frequent form of monogenic diabetes

[1]Section of Genetics and Genomics, Department of Metabolism, Digestion and Reproduction, Imperial College London, London, UK. [2]Centre for Genomic Regulation, Barcelona Institute of Science and Technology, Barcelona, Spain. [3]Centro de Investigación Biomédica en Red de Diabetes y Enfermedades Metabólicas Asociadas, Madrid, Spain. [4]Transgenics Unit, Spanish National Cancer Research Centre, Madrid, Spain. [5]CNAG–CRG, Centre for Genomic Regulation, Barcelona Institute of Science and Technology, Barcelona, Spain. [6]Universitat Pompeu Fabra, Barcelona, Spain. [7]Biotechnology and Biotherapy Team, Institut du Cerveau et de la Moelle, CNRS UMR7225, INSERM U975, University Pierre et Marie Curie, Paris, France. ✉e-mail: anthonybeucher@ochre-bio.com; jorge.ferrer@crg.eu

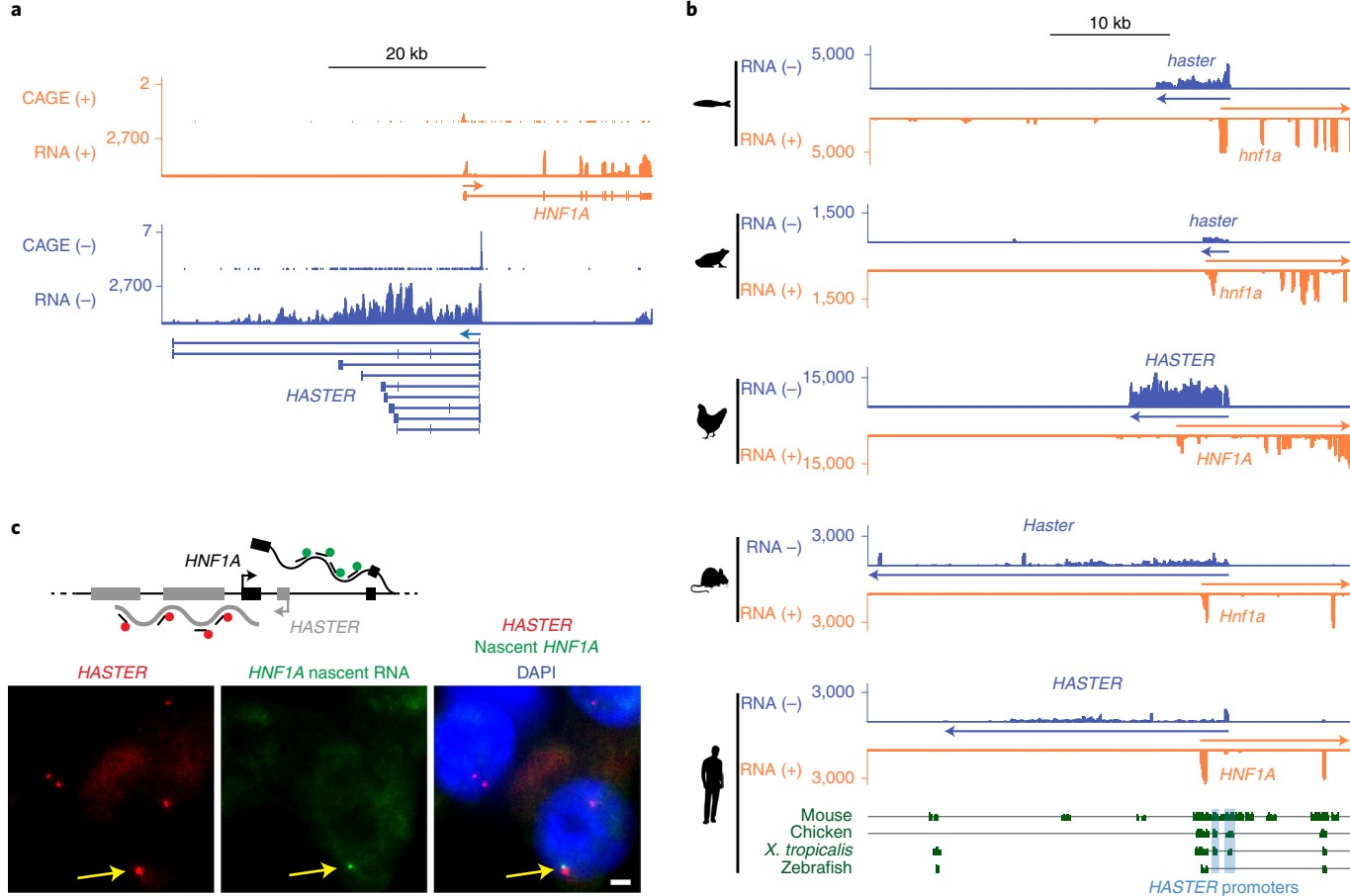

**Fig. 1 | HASTER transcribes an evolutionarily conserved nuclear RNA.**
**a**, Human islet RNA-seq (reads per kilobase per million reads, RPKM) and CAGE (normalized tag counts, TPM) showing overlapping and divergent transcription of *HNF1A* and *HASTER* (representative examples from four biological replicates). *HASTER* isoforms were detected by 3′ RACE from human islets. **b**, Liver strand-specific RNA-seq (RPKM) and Multiz alignments in the indicated species. **c**, Single-molecule fluorescence in situ hybridization for *HASTER* (exonic probes) and *HNF1A* nascent transcripts (intronic probes) in EndoC-βH3 β cells. The yellow arrows indicate co-localization of *HASTER* and nascent *HNF1A* transcripts. Quantifications are shown in Extended Data Fig. 2. Scale bar, 2 μm.

mellitus[18], while rare and common variants predispose to type 2 diabetes[19,20]. Studies of homozygous *Hnf1a* null mutant mice have shown that HNF1A is essential for differentiated cell programs in various organs, whereas human *HNF1A* haploinsufficiency causes diabetes due to selective abnormalities in pancreatic β cells, indicating that the gene dosage sensitivity of *HNF1A* is cell specific[18,21–26]. We now show that *HASTER* is a cell-specific *cis*-acting transcriptional stabilizer of HNF1A and demonstrate that disruption of this function causes diabetes mellitus in mice.

## Results

### Evolutionarily conserved co-expression of *HNF1A* and *HASTER*

*HNF1A-AS1*, or *Hnf1a-os1* and *Hnf1a-os2* in mice, is a putative non-coding transcript that is transcribed from intron 1 of *HNF1A* and runs in antisense configuration (Fig. 1a). In the present study, we focus on the regulatory function of the promoter of *HNF1A* antisense transcripts. We named this DNA region *HASTER* (*HNF1A* stabilizer). *HNF1A* antisense transcripts, which we refer to as HASTER RNAs, have previously been proposed to exert *trans*-regulation of proliferation in cell-based models[14,27–31], but so far the transcriptional *cis*-regulatory function of the lncRNA or its promoter have not been characterized with genetic tools.

We used cap analysis gene expression sequencing (CAGE-seq), RNA sequencing (RNA-seq) and 3′ rapid amplification of complementary DNA ends (RACE) to show that *HASTER* transcribes myriad transcript isoforms that originate from a major upstream transcriptional start

site in human islets and an additional downstream start site in other tissues (Fig. 1a and Extended Data Fig. 1a). Both transcriptional start sites are located in evolutionarily conserved sequences that show active promoter chromatin (high H3K4me3 and low H3K4me1) in islets and liver (Fig. 1b and Extended Data Fig. 1a). *HASTER* is expressed exclusively in *HNF1A*-expressing tissues, including the liver, gut, pancreas and kidney, and has the same antisense configuration across species (Fig. 1b and Extended Data Fig. 1b,c). Subcellular fractionation of EndoC-βH3 human β cells showed that *HASTER* transcripts were associated with chromatin, and single-molecule fluorescence in situ hybridization showed that *HASTER* transcripts were exclusively present in one or two nuclear foci that co-localized with *HNF1A* nascent transcripts (Fig. 1c and Extended Data Fig. 2a–c). Therefore, *HASTER* transcribes an evolutionarily conserved nuclear lncRNA that is co-expressed with *HNF1A* across tissues.

### *HASTER* is a negative regulator of *HNF1A*

To study *HASTER* function, we created a 320-base-pair (bp) deletion of the main *HASTER* promoter (P1) in human embryonic stem cells (hESCs) (Fig. 2a) and differentiated them into hepatocyte-like cells[32]. In control cells, *HASTER* transcripts were already detected at maximal levels at the hepatoblast stage, while *HNF1A* messenger RNA (mRNA) increased gradually during maturation to hepatocytes (Fig. 2b). *HASTER*-deleted cells showed increased hepatocyte *HNF1A* mRNA (mean = 1.3- and 1.6-fold versus control cells for two independent deletions; *P* = 0.01

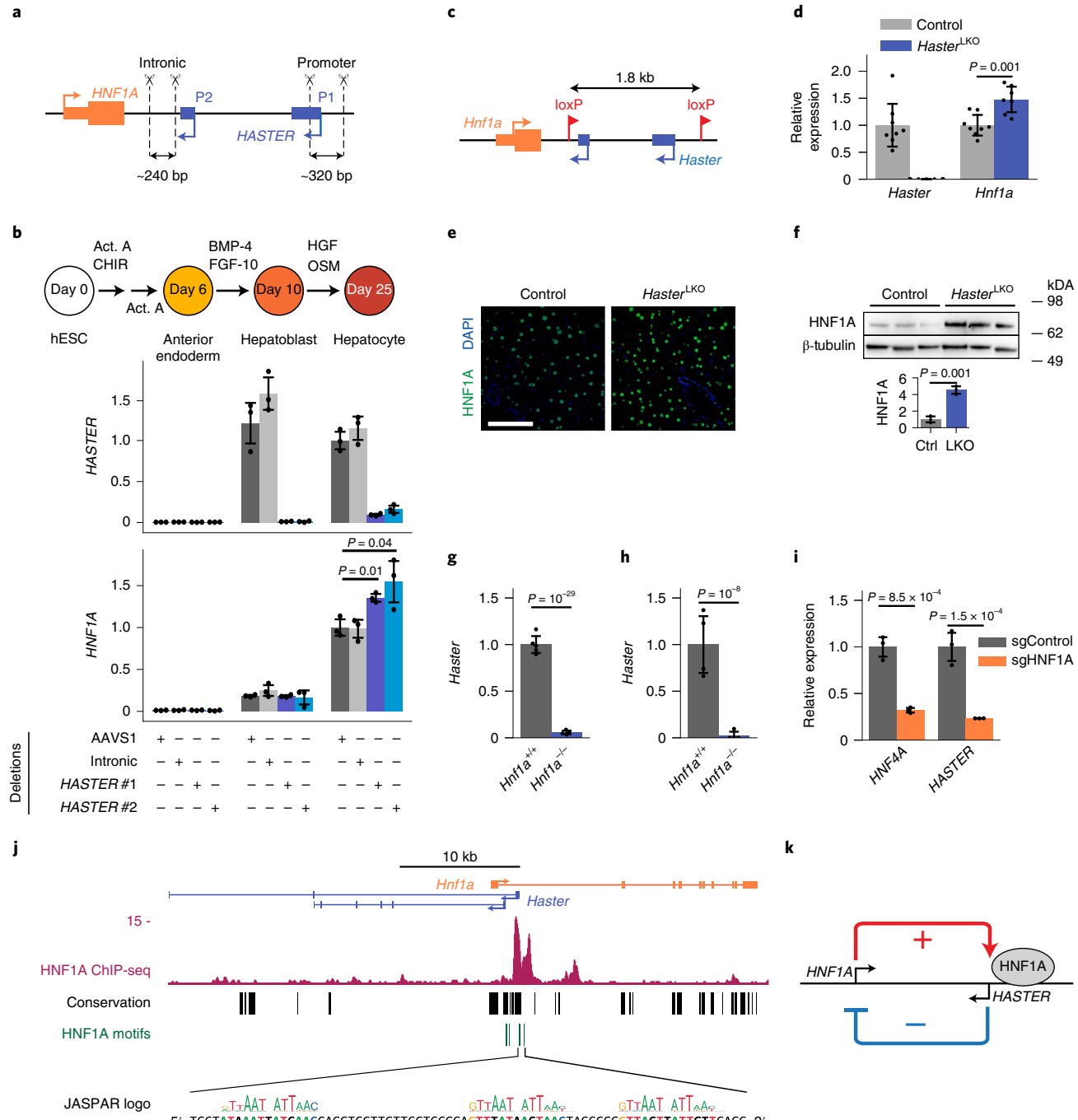

**Fig. 2 | *HASTER* negative feedback regulates HNF1A in mice and humans.**
**a**, Homozygous deletions of the *HASTER* promoter (two deletions with independent sgRNA pairs) or control deletions in *HNF1A* intron 1 or *AAVS1* were generated in hESCs. **b**, *HNF1A* mRNA was increased in differentiated hepatocytes from *HASTER* mutant hESCs (*n* = 3 independent clones per deletion). The bar graphs show *RPLP0*-normalized expression values (means ± s.d.). Statistical significance was determined by two-tailed Student's *t*-test. Act. A, Activin A; BMP-4, bone morphogenetic protein 4; HGF, hepatocyte growth factor; OSM, Oncostatin M. **c**, Schematic of the mouse *Haster^f^* allele. **d**, Liver RNA levels in seven *Haster*^LKO^ and eight control mice. The data represent *Tbp*-normalized values (means ± s.d.). Statistical significance was determined by two-tailed Student's *t*-test. **e**, Liver HNF1A immunofluorescence in the indicated genotypes. Scale bar, 50 μm. **f**, Western blot for HNF1A on liver extracts (*n* = 3 mice for each genotype). The bars represent relative expression levels (means ± s.d.).

Statistical significance was determined by two-tailed Student's *t*-test. Ctrl, control. **g**,**h**, *Haster* was decreased in *Hnf1a*^−/−^ islets (**g**; *n* = 4 *Hnf1a*^−/−^ and *n* = 5 *Hnf1a*^+/+^ mice) and liver (**h**; *n* = 4 mice per genotype). The bars represent relative expression levels (means ± s.d.). Statistical significance was determined by two-sided Wald test with adjusted *P* values. **i**, EndoC-βH3 cells carrying an indel in *HNF1A* exon 1 showed decreased *HASTER* as well as *HNF4A*—another HNF1A-dependent gene (*n* = 3 lentiviral transductions). The data represent means ± s.d. and are normalized to *TBP*. Statistical significance was determined by two-tailed Student's *t*-test. **j**, HNF1A binds the *Haster* promoter in mouse liver (representative example from three replicates, MACS2 *P* values). The locations of seven HNF1A motifs with a JASPAR CORE score of >0.8 are shown, along with the sequences of three motifs. See Extended Data Fig. 1 for information on transcriptional start sites. **k**, Schematic of the *HNF1A/HASTER* negative feedback loop.

and $P = 0.04$, respectively; Student's $t$-test) (Fig. 2b). Thus, *HASTER* exerts negative regulation of *HNF1A* in an in vitro human liver cell model.

To examine this function in vivo, we generated mice with LoxP sites flanking a 1.8-kilobase (kb) region containing *Haster* transcriptional start sites (Fig. 2c and Extended Data Fig. 3a,b) and used a liver Cre transgene[33] to breed liver-specific *Haster* homozygous deletions (*Haster*^LKO). *Haster*^LKO mice were born at Mendelian rates and showed normal organ formation, weight and glucose homoeostasis (Extended Data Fig. 3c,d). Consistent with human mutant cells, *Haster*^LKO mice showed increased liver *Hnf1a* mRNA ($1.5 \pm 0.3$-fold) and protein ($4.5 \pm 0.6$-fold) (Fig. 2d–f). Similar results were observed in germline *Haster* mutant mice (Extended Data Fig. 3e). Thus, *HASTER* negatively regulates *HNF1A* in mouse and human hepatic cells.

### HNF1A is a positive regulator of *HASTER*

The observation that *HASTER* modulates *HNF1A* hinted at a feedback mechanism. To examine whether HNF1A in turn regulates *HASTER*, we studied *HNF1A*-deficient cells. *HASTER* was strongly downregulated in pancreatic islets and liver from homozygous *Hnf1a* null mutant mice and in *HNF1A*-deficient EndoC-βH3 human β cells (Fig. 2g–i). *HASTER* transcripts seemed highly sensitive to HNF1A levels because partial *HNF1A* knockdown caused markedly decreased *HASTER* and only marginal changes in other HNF1A-dependent genes such as *HNF4A*[34] (Extended Data Fig. 4a). Conversely, upregulation of *Hnf1a* mRNA by ~30–80% through CRISPR–Cas9 synergistic activation mediator (CRISPR–SAM) led to ~50–120% increased *Haster* RNA (Extended Data Fig. 4b). This effect was probably direct because the *HASTER* promoter has seven HNF1A recognition sequences that are bound by HNF1A in mouse liver and human EndoC-βH3 β cells (Fig. 2j). These results suggested that the *HASTER* promoter functions as a HNF1A-sensing platform that drives *HASTER* transcription in accordance with HNF1A concentrations. Taken together, our observations revealed a negative feedback loop in which HNF1A positively regulates *HASTER* while *HASTER* negatively regulates *HNF1A* (Fig. 2k).

### *HASTER* negative feedback controls HNF1A pioneer-like activity

To investigate the consequences of disrupting this feedback loop, we performed RNA-seq on liver from *Haster*^LKO and control mice (Fig. 3a and Supplementary Table 1). Consistent with the increased HNF1A levels in *Haster*^LKO liver, deregulated transcripts and functional annotations were negatively correlated with those of *Hnf1a* knockout liver[22] (Fig. 3b,c and Extended Data Fig. 3f). A subset of genes that were most strongly upregulated in *Haster*^LKO liver were, however, specifically expressed in kidney or intestine—two other HNF1A-expressing organs (Fig. 3c and Extended Data Fig. 5a). Therefore, *Haster* mutations led to increased expression of HNF1A-dependent liver genes, but also activated ectopic transcription.

Next, we examined HNF1A genomic binding in *Haster*^LKO liver. Overall, the HNF1A binding strength was increased in *Haster*^LKO liver; 325 peaks showed increased HNF1A binding at a false discovery rate (FDR) of ≤0.05 (Fig. 3d). Remarkably, *Haster*^LKO liver showed HNF1A neo-binding sites at 105 regions that were not bound by HNF1A in control livers (Fig. 3d–f).

HNF1A can bind in vitro to nucleosomal DNA[35] and has been used to activate repressed liver genes in fibroblasts and reprogram them into hepatocytes[36]—two properties of pioneer transcription factors[37]. Although pioneer transcription factors have the ability to bind inaccessible chromatin, they typically show stable binding to different genomic regions across tissues[22,38], suggesting that cell-specific parameters, such as perhaps cellular transcription factor concentrations, might influence their in vivo binding selectivity and the capacity to create accessible chromatin. In keeping with this notion, HNF1A neo-binding sites did not show accessible chromatin in normal liver (Fig. 3e,f), whereas they showed classical active chromatin modifications (H3K4me3 and H3K27ac) in *Haster*^LKO liver (Fig. 3g and Extended

Data Fig. 5b–f). Interestingly, HNF1A neo-binding sites contained canonical high-affinity HNF1 binding motifs, suggesting that many could be bona fide HNF1A targets in other HNF1A-expressing tissues (Fig. 3h). Thus, increased HNF1A in *Haster*^LKO liver resulted in the creation of new binding sites, which led to the formation of new active chromatin regions.

Increased HNF1A binding at pre-existing active gene promoters in *Haster*^LKO liver led to increased gene expression; around one-quarter of genes in this class showed greater than twofold higher expression in *Haster*^LKO (Extended Data Fig. 5d). HNF1A neo-binding events in newly activated promoter regions led to ectopic activation of genes that are normally not expressed in liver, such as the kidney-enriched genes *Ggt* and *Tinag* (Fig. 3f and Extended Data Fig. 5d,e). Consistently, several HNF1A neo-binding sites did not show accessible chromatin in normal liver yet showed accessible chromatin in other HNF1A-expressing tissues such as kidney (Fig. 3c,f and Extended Data Fig. 5a,e). Some newly activated promoters did not overlap with any annotated mouse transcription start site, suggesting that increased HNF1A could also activate aberrant de novo promoters (Extended Data Fig. 5f,g).

In summary, genetic disruption of the *HASTER* feedback loop led to increased cellular HNF1A concentrations, which caused either super-activation of pre-existing HNF1A-bound promoters or the transformation of silent inaccessible chromatin into active promoters (Fig. 3i). This indicates that the *HASTER* feedback is crucial to control the pioneering-like activity of HNF1A, and to fine-tune the tissue specificity of HNF1A-dependent transcriptional programs.

### *Haster* inactivation causes diabetes

*HNF1A* haploinsufficiency leads to pancreatic β cell dysfunction and diabetes[18]. To examine *Haster* in pancreatic cells, we used a *Pdx1*-Cre transgene to excise *Haster* in all pancreatic epithelial lineages (*Haster*^pKO mice). *Haster*^pKO mice showed normal morphology and growth (Extended Data Fig. 6a), yet male mice displayed glucose intolerance with insulin deficiency by 8 weeks, as well as fasting hyperglycaemia (glycaemia = $137 \pm 16$ mM in *Haster*^pKO, $87 \pm 5$ mM in *Haster*^f/f littermates and $98 \pm 4$ mM in *Pdx1*-Cre; $t$-test $P < 0.05$) (Fig. 4a,b and Extended Data Fig. 6b). Male mice with germline mutations (*Haster*^−/−) were born at Mendelian rates and showed no overt manifestations, but also showed diabetes, glucose intolerance and hypoinsulinaemia (Fig. 4c–e and Extended Data Fig. 6c,d). Thus, inactivation of *Haster* in the germline or in the pancreas led to impaired insulin secretion and diabetes.

### *Haster* knockout leads to *HNF1A* induction or silencing in islet cells

*Haster*^pKO and *Haster*^−/− pancreas showed increased HNF1A immunoreactivity in all acinar cells and in many endocrine cells (Fig. 4f). This confirmed that *Haster* also acts as a negative regulator of *Hnf1a* in the pancreas. However, numerous other islet endocrine cells from 8- to 12-week-old *Haster*^pKO and *Haster*^−/− mice were completely devoid of HNF1A immunoreactivity (Fig. 4f).

To further understand *Haster*-dependent regulation of pancreatic HNF1A expression, we analysed mice in which *Haster* was deleted at different stages. At embryonic stage E11.5, most *Haster*^−/− multipotent pancreatic progenitors showed markedly heterogeneous HNF1A expression, with many cells showing low or no HNF1A expression, whereas HNF1A expression was uniform in surrounding primitive gut cells (Fig. 4g). At embryonic stage E15.5, β cells from *Haster*^−/− and *Haster*^pKO embryos also showed highly variable HNF1A levels, ranging from an apparent absence in many cells to marked overexpression in 1–5% of β cells (Extended Data Fig. 6e–h). This contrasted with highly uniform HNF1A staining in control embryonic β cells (Extended Data Fig. 6e,f). This dual phenotype became more evident if *Haster*^pKO and *Haster*^−/− mice were analysed postnatally, with more visible HNF1A-negative cells (62 and 80%, respectively) and more HNF1A-overexpressing cells

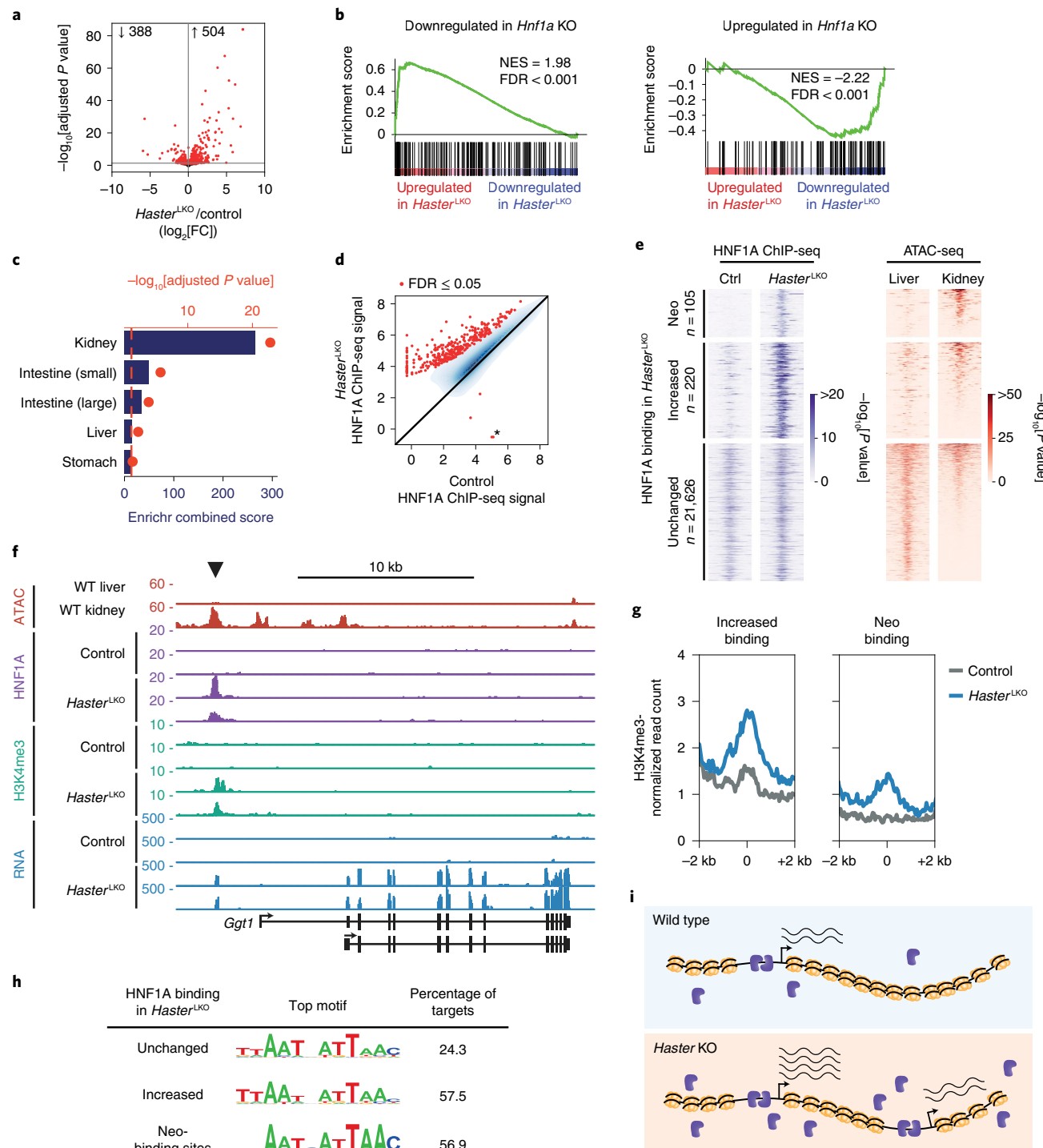

**Fig. 3 | *Haster* controls HNF1A pioneer-like activity. a**, RNA-seq in *Haster*[LKO] liver. Differentially expressed genes (adjusted $P \leq 0.05$) are highlighted in red and total numbers are indicated ($n = 5$ mice per genotype). FC, fold change. **b**, GSEA showing that genes up- or downregulated in *Hnf1a* KO liver have opposite expression patterns in *Haster*[LKO] liver. NES, normalized enrichment score. **c**, Enrichment of *Haster*[LKO] liver upregulated genes in different mouse tissues (Mouse Gene Atlas). The bars indicate Enrichr scores and the red dots show Fisher's exact $-\log_{10}$-adjusted $P$ values. **d**, HNF1A binding strength ($\log_2$[ChIP-seq normalized read count]) in *Haster*[LKO] and control liver ($n = 3$ mice). Red represents differentially bound sites (FDR $\leq 0.05$) whereas blue represents a kernel density of HNF1A-bound sites with FDR > 0.05. The asterisk denotes the *Haster*[LKO] deletion. **e**, Left, HNF1A occupancy in control and *Haster*[LKO] liver. Right, chromatin accessibility for the same regions in liver and kidney. Neo-binding

sites are bound by HNF1A only in *Haster*[LKO]. Increased bound sites include all of the other sites showing increased binding in *Haster*[LKO]. The heatmaps show the average signal of three replicates for ChIP-seq and two replicates for the assay for transposase-accessible chromatin with high-throughput sequencing (ATAC-seq). Windows were defined as peak centres ± 1 kb. $P$ values were obtained with MACS2. **f**, Activation of a kidney-specific gene in *Haster*[LKO] liver. $y$ axes represent MACS2 $P$ values for ChIP-seq and RPKM for RNA-seq. **g**, H3K4me3 in HNF1A-bound regions in *Haster*[LKO] and control samples (average of three mice). **h**, Top HOMER de novo motifs for the different categories of HNF1A peak. **i**, Model showing that *Haster* KO leads to increased HNF1A (blue), causing increased HNF1A binding and expression of HNF1A-bound genes (bottom left), as well as HNF1A neo-binding sites that lead to transformation of silent inaccessible chromatin into active promoters (bottom right).

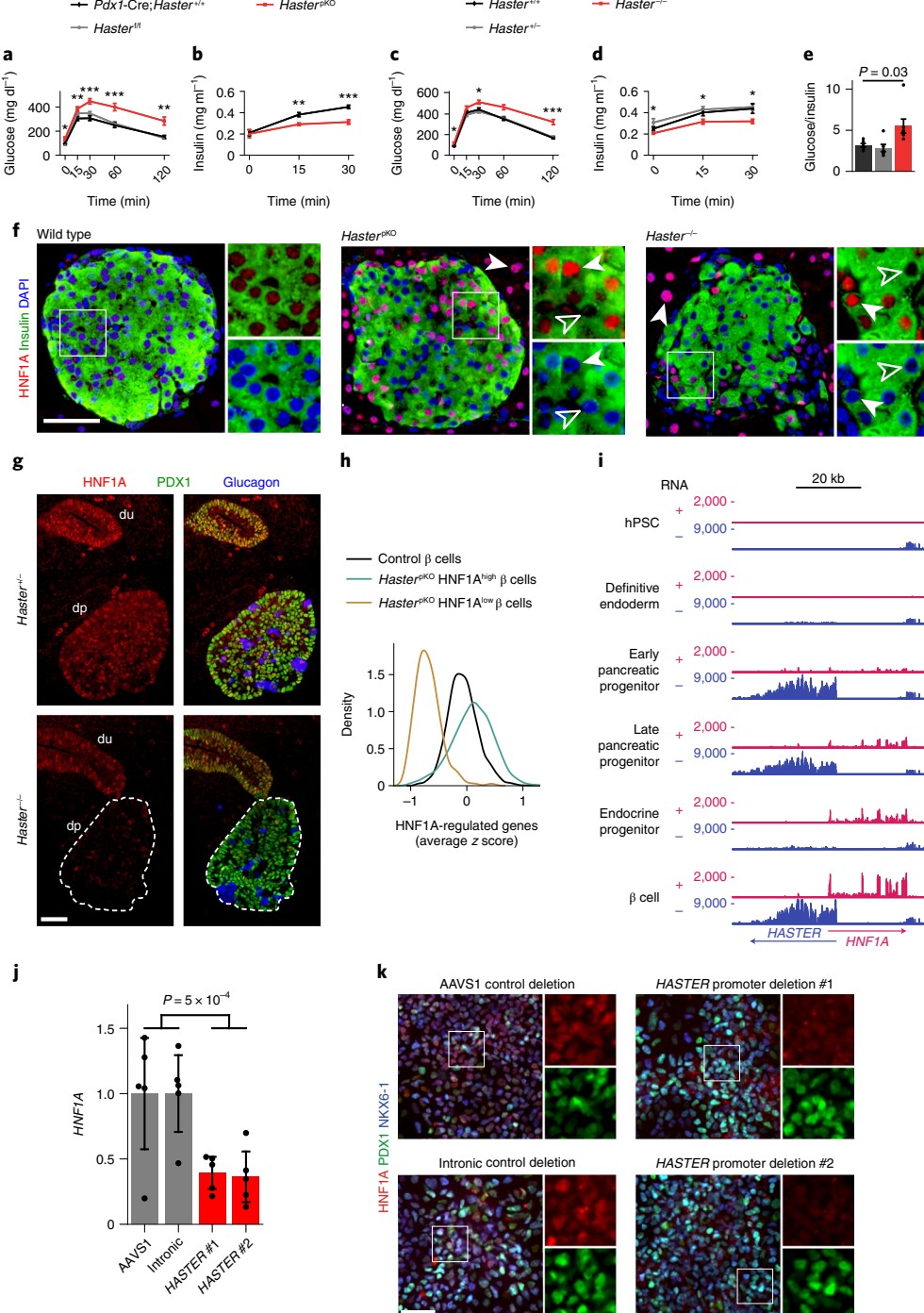

**Fig. 4 | *Haster* deletion causes islet cell HNF1A hyperactivation or silencing and diabetes. a**, Intraperitoneal glucose tolerance in 8-week-old male mice ($n = 8$ *Haster*^pKO, $n = 12$ *Pdx1*-Cre;*Haster*^{+/+} and $n = 8$ *Haster*^{f/f}). $P = 0.045, 8 \times 10^{-3}, 3 \times 10^{-4}, 4 \times 10^{-4}$ and $5 \times 10^{-3}$ at 0, 15, 30, 60 and 120 min, respectively. **b**, Plasma insulin of 8-week-old male mice ($n = 7$ *Haster*^pKO and $n = 6$ *Pdx1*-Cre;*Haster*^{+/+}). $P = 0.83, 2 \times 10^{-3}$ and $3 \times 10^{-4}$ at 0, 15 and 30 min, respectively. **c**, Intraperitoneal glucose tolerance in 8-week-old male mice ($n = 9$ *Haster*^{-/-}, $n = 12$ *Haster*^{+/-} and $n = 13$ *Haster*^{+/+}). $P = 0.048, 0.075, 0.011, 4 \times 10^{-4}$ and $2 \times 10^{-4}$ at 0, 15, 30, 60 and 120 min, respectively. **d**, Plasma insulin in 8-week-old male mice ($n = 7$ *Haster*^{-/-}, $n = 6$ *Haster*^{+/+} and $n = 6$ *Haster*^{+/-}). $P = 0.042, 0.045$ and $0.026$ at 0, 15 and 30 min, respectively. **e**, Glucose-to-insulin ratio in 8-week-old male mice ($n = 9$ *Haster*^{-/-}, $n = 12$ *Haster*^{+/-} and $n = 13$ *Haster*^{+/+}). In **a**–**e**, the data are presented as means ± s.e.m. and statistical significance was determined by two-tailed Student's *t*-test (*$P \leq 0.05$; **$P \leq 0.01$; ***$P \leq 0.001$). **f**, Immunofluorescence for HNF1A and insulin, showing either HNF1A overexpression (solid arrowheads) or no HNF1A expression (empty arrowheads) in endocrine cells of adult *Haster*^pKO

and *Haster*^{-/-} mice. Note that all acinar cells from mutant mice overexpressed HNF1A ($n = 3$ *Haster*^{+/+}, $n = 3$ *Haster*^pKO and $n = 2$ *Haster*^{-/-}). **g**, Immunofluorescence for HNF1A, PDX1 (a pancreatic and duodenal marker) and glucagon in *Haster*^{-/-} and control E11.5 embryos, showing low heterogeneous HNF1A in pancreatic but not gut progenitors. dp, dorsal pancreas (delineated by dashed lines in KO); du, duodenum. **h**, Kernel density estimation of HNF1A-regulated gene expression (average *z* score) showing either down- or upregulation of HNF1A-dependent genes in *Haster*^pKO HNF1A^low and HNF1A^high β cell clusters. **i**, RNA-seq (RPKM) from the indicated hESC-derived differentiation stages. **j**, *HNF1A* mRNA in hESC-derived pancreatic progenitors carrying *HASTER* P1 homozygous deletions (see Fig. 2a) ($n = 5$ independent differentiations). The data are presented as *TBP*-normalized relative expression (means ± s.d.). Statistical significance was determined by two-tailed Student's *t*-test. **k**, Immunofluorescence for HNF1A, PDX1 and NKX6-1 in hESC-derived pancreatic progenitors carrying the indicated deletions, showing downregulation of HNF1A ($n = 2$ per deletion). In **f**, **g** and **k**, the scale bars represent 50 μm.

(24 and 10%, respectively) (Fig. 4e,f). Inactivation of *Haster* after the formation of β cells, however, resulted in very few HNF1A-negative β cells and more frequent HNF1A overexpression (Extended Data Fig. 6i–k). Extended Data Fig. 6e summarizes the results from different models. Thus, *Haster* inactivation caused a unique variegated HNF1A expression phenotype in β cells, with co-existing silencing and overexpression. Therefore, *Haster* acts as a negative regulator of HNF1A in the pancreas, as in the liver, but also has a developmental cell-specific role to ensure HNF1A expression in early pancreatic progenitors and islet endocrine cells. Importantly, *Haster* is essential for β cell function and glucose homoeostasis.

## Variegation of *Haster*-deficient islet cell transcriptomes

Next, we defined the transcriptional impact of HNF1A expression heterogeneity. We performed single-cell RNA-seq of islet cells from *Haster*[pKO] and control mice (Supplementary Table 2) and used graph-based clustering to separate major endocrine cell types (Extended Data Fig. 7a–c). For each cell, we calculated the average normalized expression of known HNF1A-regulated genes. Consistent with HNF1A expression heterogeneity in *Haster*[pKO] β cells, we observed increased variability of HNF1A-regulated genes across *Haster*[pKO] β cells (interquartile range = 0.53 versus 0.34 for *Haster*[pKO] and control β cells, respectively; Brown–Forsythe; $P < 10^{-93}$) (Fig. 4h and Extended Data Fig. 7d–h). Further examination revealed that a large fraction of *Haster*[pKO] β cells showed increased expression of HNF1A-regulated genes, while another β cell cluster (β HNF1A[low]) showed strong downregulation of HNF1A-dependent genes, such as *Ttr*, *Tmem27*, *Slc2a2* and *Kif12* (Fig. 4h and Extended Data Fig. 8 and Supplementary Tables 3 and 4). This β HNF1A[low] cluster was specific to *Haster*[pKO] islet cells, constituted 5–21% of β cells and was discernible with independent clustering methods (Extended Data Fig. 7d–f). β HNF1A[low] cells were less abundant than expected from immunostainings, possibly due to a known propensity of *Hnf1a* knockout cells to dissociate during islet isolation. Thus, *Haster* mutations caused either functional *HNF1A* deficiency in pancreatic β cells, which is known to cause diabetes, or overexpression of *HNF1A*-dependent genes. *Haster*, therefore, acts to ensure the stability of β cell HNF1A-regulated programs.

## *HASTER* modulates *HNF1A* in human pancreatic progenitors

Next, we investigated whether *HASTER* also regulates *HNF1A* in human pancreatic cells. Analysis of published datasets showed that *HASTER* is activated during the early stages of hESC-derived pancreatic differentiation[39] (Fig. 4i). To test *HASTER* function in human pancreatic progenitors, we used the hESC clones carrying the 320-bp *HASTER* P1 deletion (Fig. 2a) and generated pancreatic progenitors[40]. In contrast with the results after hepatic differentiation, which showed increased *HNF1A* mRNA, *HASTER* knockout pancreatic progenitors showed a 62% decrease of *HNF1A* mRNA and low heterogenous HNF1A protein levels (Fig. 4j,k). These results showed that *HASTER* also acts as an essential organ-specific positive regulator of *HNF1A* in human early pancreatic multipotent progenitor cells.

## The *HASTER* promoter activates *HNF1A* in *cis*

Next, we explored how *HASTER* exerts positive and negative regulation of *HNF1A*, first focusing on the positive regulatory function. To assess whether *HASTER* acts in *cis* or *trans*, we bred compound heterozygous *Hnf1a*[+/−];*Haster*[+/−] mice. Single heterozygous *Haster*[+/−] or *Hnf1a*[+/−] mice do not develop hyperglycaemia[21] (in contrast with human *HNF1A* heterozygous mutations, which cause diabetes) (Fig. 5a). Remarkably, compound heterozygous *Hnf1a*[+/−];*Haster*[+/−] young mice developed severe fasting and fed hyperglycaemia with hypoinsulinaemia, but otherwise did not exhibit extra-pancreatic manifestations observed in homozygous *Hnf1a*-mutant mice[24,26] (Fig. 5a). This was accompanied by absent HNF1A expression in most β cells of 10-week-old *Hnf1a*[+/−];*Haster*[+/−] mice (Fig. 5b). Because the wild-type *Haster* allele was not able to activate the

wild-type *Hnf1a*, which was located on the alternative chromosome, this shows that *Haster* positively regulates *Hnf1a* in *cis* in islet cells. We also created hybrid-strain mice with a heterozygous *Haster* null allele and found decreased islet *Hnf1a* mRNA from the chromosome carrying the *Haster* null allele ($P < 0.02$) (Fig. 5c). Genetic experiments thus showed that *Haster* acts in *cis* to maintain *Hnf1a* expression in islet β cells.

Next, we examined whether *HASTER* transcriptional elongation, its RNA products or the promoter DNA are required to prevent *HNF1A* silencing. To this end, we created an allele with a transcriptional termination signal downstream of *Haster* (*Haster*[stop]; Fig. 5d). We bred this *Haster*[stop] allele in a hybrid-strain background and performed RNA-seq for strain-specific quantitation of *Hnf1a* mRNA in islets. As expected, we found severely diminished *Haster* transcripts from the *Haster*[stop] allele (93% reduction; Wilcoxon rank-sum; $P = 0.02$). However, we still detected abundant *Hnf1a* exon 1 transcripts from the stop allele (Fig. 5d). Thus, whereas deletion of the *Haster* promoter DNA caused islet cell *Hnf1a* silencing in *cis*, this was not recapitulated by blocking *Haster* transcription. This indicates that the *Haster* promoter, but not transcriptional elongation or RNAs, is an essential positive *cis*-acting element of *Hnf1a* in pancreatic islets.

## *HASTER* inhibits *HNF1A* in *cis*

Next, we examined how *HASTER* exerts negative regulation of *HNF1A*. To assess whether this function also occurs in *cis*, we again examined *Hnf1a*[+/−];*Haster*[+/−] mice, but this time focused on liver, where *Haster* deficiency causes uniformly increased HNF1A expression. Compound heterozygotes showed increased HNF1A in hepatocytes, indicating that increased expression of the *Hnf1a*[+] allele from the chromosome carrying the *Haster* deletion could not be compensated in *trans* by the *Hnf1a*[−];*Haster*[+] allele (Fig. 5e). Interestingly, pancreatic acinar cells showed similar behaviour to hepatocytes in compound heterozygotes, with increased HNF1A expression (Fig. 5b). We also examined *Haster*[+/−] mice bred on a hybrid-strain background and found that liver *Hnf1a* mRNA was selectively increased in *Haster* mutant chromosomes (Fig. 5f). Both findings showed that *Haster*-dependent inhibition of *HNF1A*, like its activating function, occurs in *cis*.

## The *HASTER* promoter, but not its RNA, is essential for *HNF1A* inhibition

Next, we examined the role of *HASTER* transcriptional elongation, RNA molecules or its promoter in this *cis*-inhibitory function. Hybrid-strain mice heterozygous for *Haster*[stop] showed that transcriptional blockage did not cause increased liver *Hnf1a* exon 1 transcripts in chromosomes carrying the *stop* allele (Fig. 5g). To further examine the role of the *HASTER* promoter versus transcripts, we generated clonal EndoC-βH3 cell lines with homozygous *HASTER* promoter deletions encompassing both transcriptional start sites (*HASTER*[ΔP/ΔP]) or a 320-bp deletion of the P1 promoter (*HASTER*[ΔP1/ΔP1]) (Extended Data Fig. 9a,b). Both deletions caused increased *HNF1A* mRNA (Extended Data Fig. 9a,b), recapitulating the phenotype of mice in which *Haster* was excised after the formation of β cells (Extended Data Fig. 6k). To study the role of *HASTER* transcription, we targeted deactivated Cas9 to the *HASTER* transcriptional start site (CRISPR interference (CRISPRi) *roadblock*[41]) or to a control intronic region located between *HASTER* and *HNF1A* promoters (Fig. 5h). Expectedly, targeting the *HASTER* promoter suppressed the formation of HASTER RNAs, although it did not influence *HNF1A* mRNA or *HNF4A*, an HNF1A-dependent transcript[34] (Fig. 5h). Similarly, degradation of HASTER nuclear transcripts using GapmeRs did not affect *HNF1A* or *HNF4A* mRNAs (Extended Data Fig. 9c). Conversely, CRISPR–dCas9–SAM activation of *HASTER* transcription in mouse or human β cell lines led to greater than fivefold levels of HASTER RNA without changing *HNF1A* or *HNF4A* mRNAs (Fig. 5i and Extended Data Fig. 9d). Thus, modulation of *HASTER* transcripts or transcriptional elongation did not recapitulate the inhibitory effects of *HASTER* on *HNF1A*.

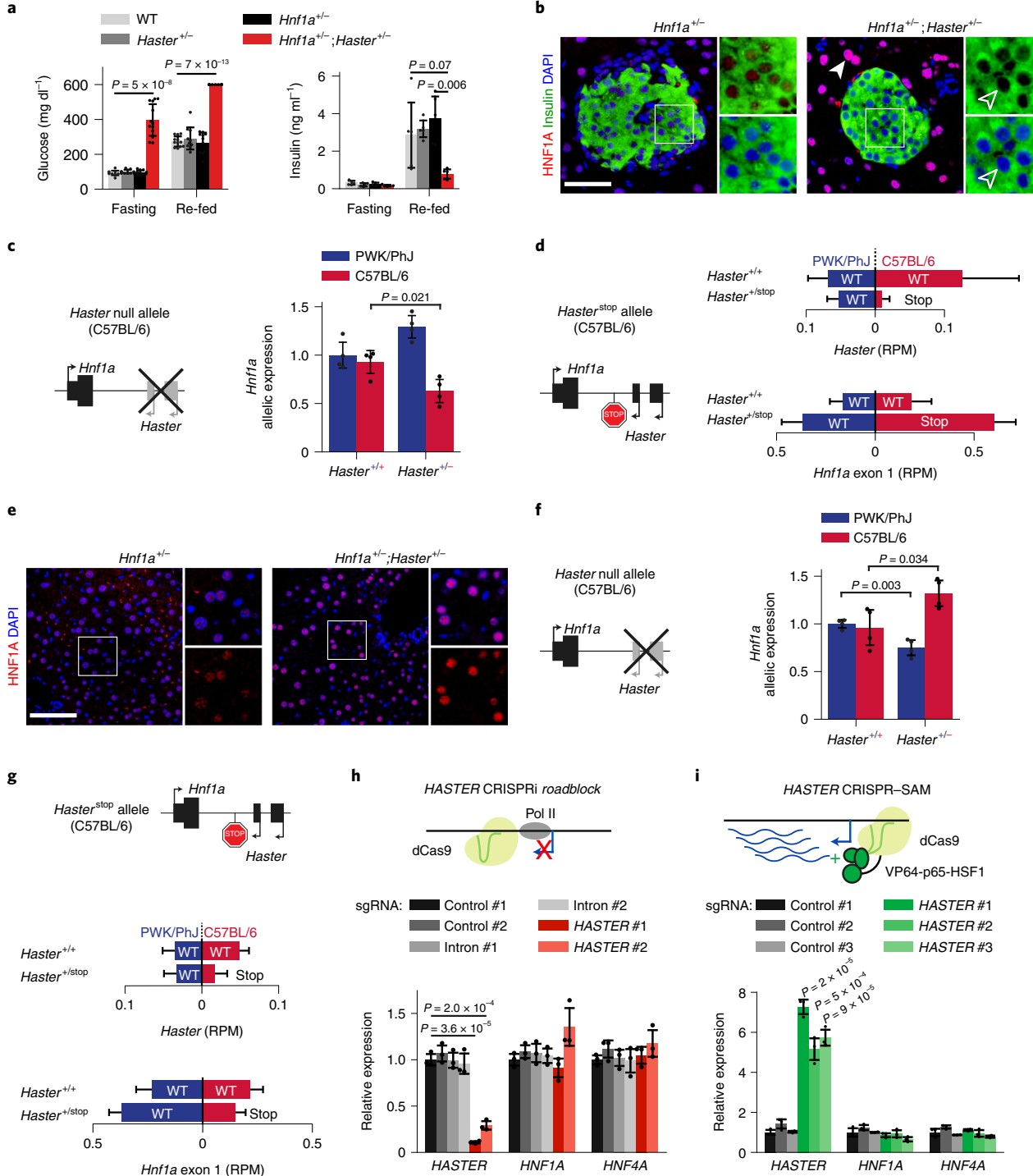

**Fig. 5 | The *HASTER* promoter is a positive and negative *cis*-acting element.**
**a**, Severe fasting and fed hyperglycaemia (left; $n = 12$ wild-type (WT) mice, $n = 10$ *Haster*[+/−] mice, $n = 11$ *Hnf1a*[+/−] mice and $n = 13$ *Hnf1a*[+/−];*Haster*[+/−] mice) and reduced insulin secretion (right; $n = 5$ mice per genotype) in *Hnf1a*[+/−];*Haster*[+/−] compound heterozygotes. The data are presented as means ± s.d. Statistical significance was determined by two-tailed Student's *t*-test. **b**, Immunofluorescence showing normal HNF1A in *Hnf1a*[+/−] islets and no expression in most islet cells from adult *Hnf1a*[+/−];*Haster*[+/−] mice ($n = 1$ per genotype). Solid arrowhead: HNF1A[high] acinar cell. Hollow arrowhead: HNF1A[low] β cell. Scale bar, 50 μm. **c**, Allele-specific *Hnf1a* mRNA in islets from hybrid-strain mice carrying the *Haster* mutation in the C57BL/6 chromosome. *Hnf1a* was quantified by strain-specific qPCR and normalized to *Tbp* ($n = 4$ mice per genotype). The data are presented as means ± s.d. Statistical significance was determined by two-tailed Student's *t*-test. **d**, Strain-specific RNA-seq analysis from *Haster*[+/stop] and *Haster*[+/+] PWK/PhJ;C57BL/6 hybrid islets ($n = 4$ mice per genotype). RPM, reads per million reads.

**e**, HNF1A overexpression in liver from *Hnf1a*[+/−];*Haster*[+/−] mice ($n = 1$ per genotype). Scale bar, 50 μm. **f**, Allele-specific *Hnf1a* mRNA in liver from *Haster*[+/−] hybrid-strain mice carrying the *Haster* mutation in the C57BL/6 chromosome. *Hnf1a* was quantified with strain-specific assays and normalized to *Tbp* ($n = 4$ mice per genotype). The data are presented as means ± s.d. Statistical significance was determined by two-tailed Student's *t*-test. **g**, Strain-specific RNA expression from *Haster*[+/stop] C57BL/6;PWK/PhJ hybrid mice, showing that reducing *Haster* elongation in liver failed to increase *Hnf1a* expression from the same C57BL/6 allele. The graphs show reads per million (RPM) (means ± s.d.). **h**, Targeting dCAS9 to the *HASTER* transcriptional start site blocked *HASTER* transcription in EndoC-βH3 cells but did not affect *HNF1A* or *HNF4A* mRNAs ($n = 3$ lentiviral transductions). **i**, CRISPR–SAM *HASTER* activation in EndoC-βH3 cells did not affect *HNF1A* and *HNF4A* ($n = 3$ lentiviral transductions). In **h** and **i**, the data represent normalized expression levels (means ± s.d.) and statistical significance was determined by two-tailed Student's *t*-test.

### *HASTER* inhibition of *HNF1A* requires HNF1A binding to *HASTER*

The observation that *HASTER* transcriptional activation was not essential was unexpected because our genetic findings showed a tight correlation between HNF1A-dependent *HASTER* transcription and negative regulation of *HNF1A*. To reconcile these findings, we activated *HASTER* through lentiviral doxycycline-inducible overexpression of HNF1A (Fig. 6a). As in the CRISPR–dCas9–SAM experiments, this led to increased *HASTER*, but this time we observed a tenfold decrease of endogenous *HNF1A* mRNA (Fig. 6a). Importantly, the inhibitory effects of HNF1A overexpression were almost completely suppressed after deletion of the *HASTER* promoter region (Fig. 6b). Therefore, these studies showed that inhibition of *HNF1A* was triggered selectively by HNF1A interactions with *HASTER* promoter DNA, but not by various other manoeuvres that influenced *HASTER* transcription.

### Uncoupling of HNF1A negative autoregulation and transactivation

To further establish whether HNF1A-dependent inhibition of its own promoter was dependent on its ability to activate *HASTER* transcription, we selectively modified the transactivation function of HNF1A. To this end, we examined the sequence of the transcriptional activation domain of HNF1A and identified an intrinsically disordered region (IDR); IDRs have been implicated in transcriptional activation through phase separation[42]. A selective deletion of this IDR led to decreased HNF1A-dependent *HASTER* transcription, but did not prevent inhibition of *HNF1A* (Fig. 6c,d). We also examined HNF1B, a paralogue with the same sequence recognition specificity. We found that while HNF1B is a weaker inhibitor of *HNF1A* than HNF1A itself, fusion of HNF1B to an unrelated IDR from the FUS protein increased *HASTER* activation, yet did not have a significant impact on HNF1B-dependent *HNF1A* inhibition (Fig. 6c,d). Therefore, the *HASTER* promoter is required for HNF1A-dependent transactivation of *HASTER*, as well as for *HNF1A* autoregulation, but these are two separable molecular mechanisms.

### *HASTER* restrains *HNF1A* enhancer spatial interactions

Next, we examined whether *HASTER* function entails changes in the local histone modification landscape. Chromatin from control liver expectedly showed localized H3K4me3 enrichment surrounding *Hnf1a* and *Haster* promoters. In contrast, *Haster*[LKO] H3K4me3 showed spreading from the *Hnf1a* promoter to an intronic E enhancer region (Fig. 7a). H3K4me3 was therefore significantly increased in this E region, as well as in an upstream CTCF-bound (C) region ($t$-test; $P < 0.05$) (Fig. 7b). This spreading of H3K4me3 in *Haster*[LKO] suggested that *Haster* might insulate the *Hnf1a* promoter from the intronic E enhancer, while an increase in H3K4me3 at the E and C regions in *Haster*[LKO] suggested that *Haster* might influence the proximity of E and C regions with the H3K4me3-rich *Hnf1a* promoter. We therefore hypothesized that the *HASTER* promoter could inhibit *HNF1A* by modulating three-dimensional (3D) chromatin contacts of *HNF1A* with local regulatory elements.

To test this, we performed quantitative chromosome conformation capture using unique molecular identifiers (UMI-4C)[43]. Mouse *Hnf1a* and *Haster* promoters, as well as the intronic E enhancer region, are all located within ~7 kb. To increase the ability to capture 3D chromatin interactions with the *Hnf1a* 5′ region, we selected one viewpoint ~6 kb upstream of *Hnf1a*, near the CTCF-bound C site (viewpoint 1) and another at the *Hnf1a* promoter (viewpoint 2) (Fig. 7a). UMI-4C experiments from *Haster*[LKO] versus control liver ($n = 6$ per genotype) showed that the *Haster* deletion caused greater than twofold increased contacts between both *Hnf1a* upstream regions and the intronic E enhancer (V1; $\chi^2$ test for pooled UMI-4C libraries; $P = 0.02$) (Fig. 7a,c and Extended Data Fig. 10a). Thus, the analysis of two viewpoints showed consistent changes in interactions between the *Hnf1a* upstream region and the intronic E enhancer in *Haster*[LKO] (Fig. 7d).

Likewise, we examined human EndoC-βH3 cells that had an intact or deleted *HASTER* promoter region and used the *HNF1A* promoter as a viewpoint for quantitative UMI-4C analysis. We found that *HASTER* deletions caused increased interactions between the *HNF1A* promoter and E regions ($\chi^2$ test; $P = 0.04$; pooled UMI-4C libraries from four experiments). Next, we asked whether HNF1A binding to *HASTER* can modulate such interactions. HNF1A overexpression using the doxycycline-inducible system expectedly decreased endogenous *HNF1A* mRNA and significantly decreased interactions between the *HNF1A* promoter and the E region in *HASTER*[+/+] cells ($\chi^2$ test; $P = 0.05$) (Fig. 7e,f and Extended Data Fig. 10b–d). This effect required an intact HASTER promoter, as no significant HNF1A-dependent 3D contact differences were observed in *HASTER* mutants ($\chi^2$ test; $P = 0.78$) (Fig. 7f and Extended Data Fig. 10b–d). Out of 33 enhancer-like regions in 1 megabase surrounding *HNF1A*, only E showed significant HNF1A-dependent changes (Extended Data Fig. 10b). Therefore, these results indicate that HNF1A overexpression limits 3D contacts between *HNF1A* and an intronic enhancer region, and this effect requires the *HASTER* promoter.

These findings imply that *HASTER* inhibition of *HNF1A* transcription involves modulation of interactions between *HNF1A* and the intronic E enhancer. Consistently, E deletions prevented increased *HNF1A* mRNA after deleting *HASTER*, but did not cause significant changes when *HASTER* was intact (Fig. 7g and Extended Data Fig. 10e,f). Taken together, these experiments show that *HASTER*-dependent negative feedback of *HNF1A* occurs through a *cis* function of the *HASTER* promoter that does not require *HASTER* transcription. Instead, HNF1A binding to *HASTER* modifies the local 3D chromatin landscape and insulates *HNF1A* from *cis*-acting intronic regulatory elements (Fig. 7h).

## Discussion

These studies have uncovered a *cis*-regulatory element that senses HNF1A concentrations and feeds back on *HNF1A* to ensure appropriate cell-specific expression levels (Fig. 7h). This is achieved through a dual activating and inhibitory function that is fundamentally different from conventional *cis*-acting enhancers or silencers that provide spatiotemporal ON or OFF switches, respectively (Fig. 7i).

We show that *HASTER*'s dual function emanates from a 320-bp promoter DNA sequence and does not require transcription. However, it remains possible that transcripts have additional effects that were not explored. *HASTER*'s inhibitory function was triggered by high concentrations of HNF1A, which modified *HNF1A* promoter–enhancer interactions (Fig. 7h). The activating function of *HASTER* is reminiscent of an intronic enhancer, because it activates transcription in *cis*, and has lineage-specific essential role in pancreatic endocrine cells, plausibly due to *cis*-regulatory redundancy in other cell types. This dual *HASTER* function was most compellingly illustrated by the pancreatic knockout phenotype, in which lack of *Haster* enhancer-like activity led to HNF1A silencing in some β cells, while lack of negative feedback caused overexpression in other β cells that succeeded in activating HNF1A.

*HASTER*-dependent feedback was critical to ensure that HNF1A selects appropriate binding sites in different cell types. Interestingly, a few lncRNAs have recently been shown to negatively regulate nearby genes through different mechanisms, including the heart transcription factor gene *Hand2* (refs. [12,44]), the c-MYC oncogene[12] or *CHD2* (ref. [15]). All such genes—*HAND2*, *MYC* and *CHD2*, as well as *HNF1A*—share in common that they are haploinsufficient and encode transcriptional regulators[15,18,45,46]. Furthermore, c-MYC, HAND2 and HNF1A have been used in misexpression systems for lineage reprogramming—a feature of transcription factors that can act on repressed chromatin[36,47]. These examples, and perhaps most clearly *HASTER*'s dual function, suggest that the principal function of a group of *cis*-acting lncRNA units may be to stabilize dosage-sensitive genes that encode proteins that have a capacity to transform cell-specific chromatin landscapes.

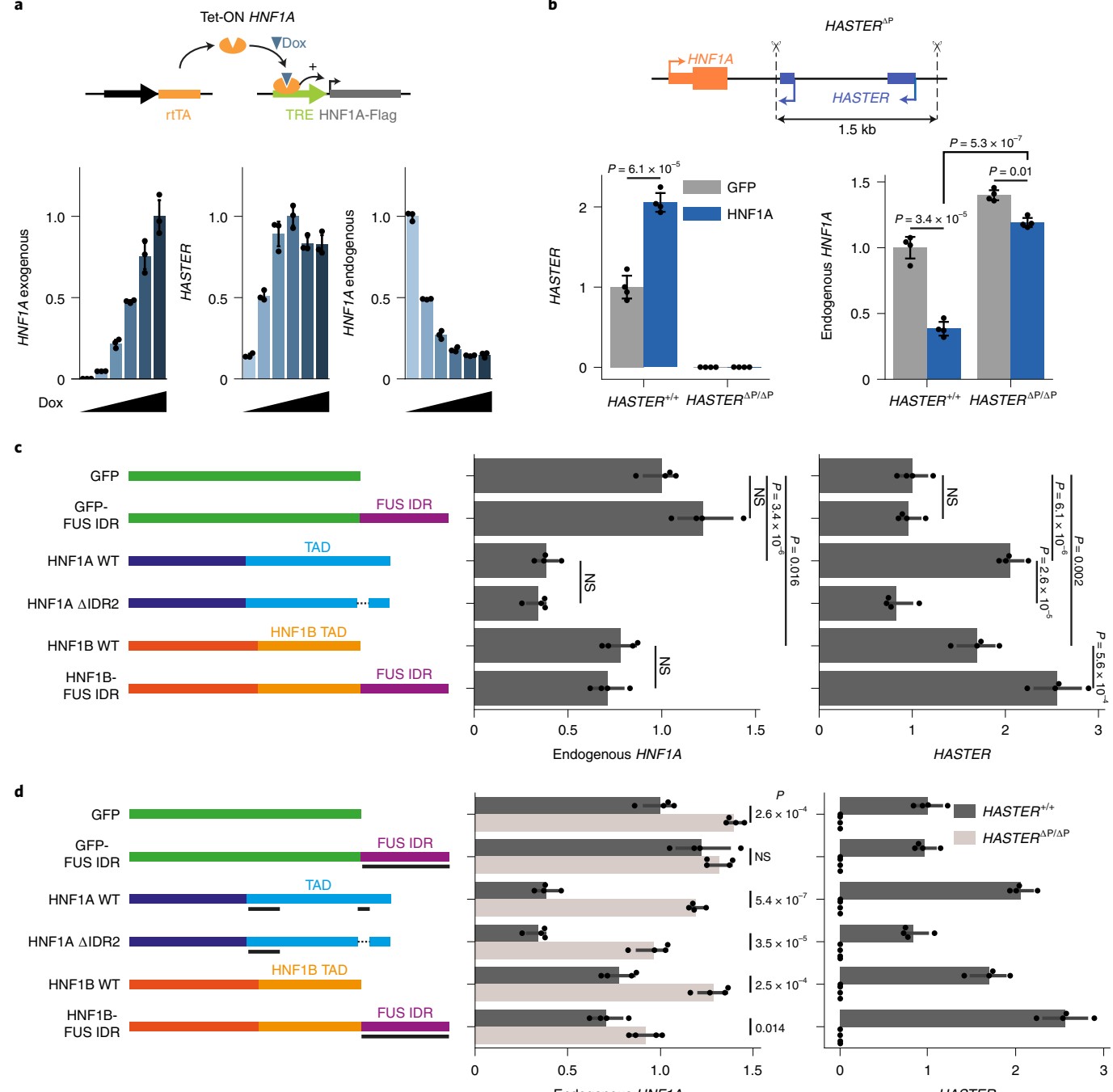

**Fig. 6 | HNF1A binding to *HASTER* mediates negative regulation of *HNF1A*.**
**a**, Doxycycline (Dox)-induced HNF1A overexpression in EndoC-βH3 cells
activated *HASTER* and blocked endogenous *HNF1A* ($n = 3$ independent
experiments). **b**, HNF1A overexpression in clonal EndoC-βH3 cell lines
with homozygous deletions of both *HASTER* promoters ($n = 4$ independent
experiments). **c**, HNF1A transactivation of *HASTER* is separable from repression
of its promoter. Wild-type HNF1A or HNF1A containing a deletion of an
endogenous IDR, HNF1B or HNF1B fused to an unrelated IDR were expressed in

EndoC-βH3 cells. Green fluorescent protein (GFP) and GFP fused to the unrelated
IDR are shown as controls ($n = 4$ independent experiments). NS, not significant.
**d**, Expression of HNF1A containing a deletion of an IDR and of HNF1B fused to
an unrelated IDR in EndoC-βH3 cells, essentially as represented in **b** with the
addition of experiments with *HASTER* promoter deletions to show that the effects
are dependent on HASTER ($n = 4$ independent experiments). In **a**–**d**, the data are
presented as *TBP*-normalized relative expression (means ± s.d.) and statistical
significance was determined by two-tailed Student's *t*-test.

Our studies exemplify a genetic defect in a mammalian lncRNA
promoter that causes an in vivo physiological phenotype. Remarkably,
the main manifestation of homozygous germline *Haster* mutations
was β cell dysfunction and diabetes. *HNF1A* heterozygous mutations
also cause selective β cell dysfunction and only subclinical altera
tions in other cell types[18], but homozygous *Hnf1a* mutations cause

severe liver and renal dysfunction, growth retardation, diabetes and
embryonic lethality[21,24]. The discovery of a transcriptional stabilizer
of *HNF1A* that has a selective function in β cells therefore provides a
lead to dissect cell-specific genetic mechanisms underlying *HNF1A*
haploinsufficient diabetes. It is also relevant for efforts to modulate
*HNF1A* function in β cells.

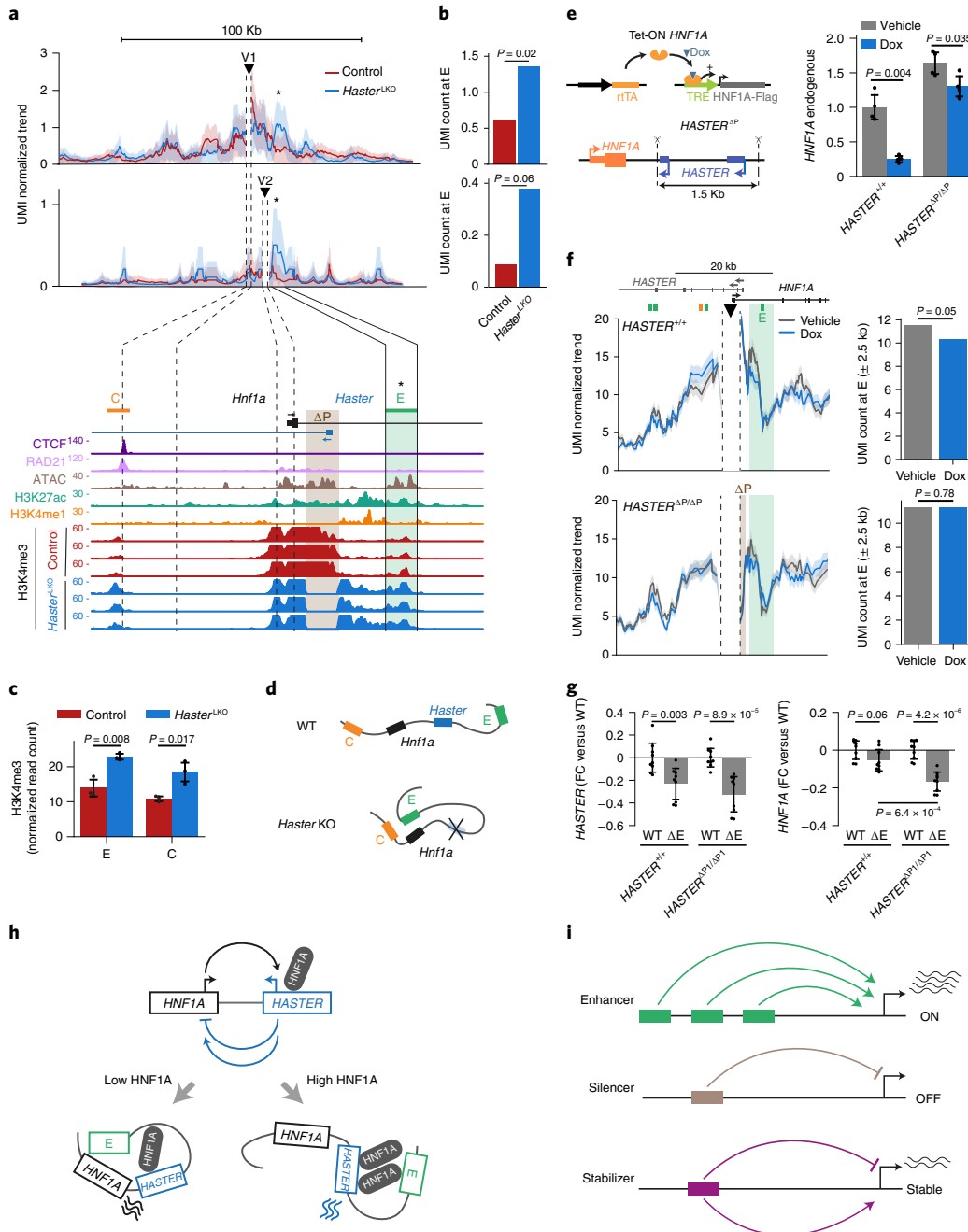

**Fig. 7 | *HASTER* remodels enhancer–*HNF1A* interactions. a,** *Haster*[LKO] liver shows increased contacts between *Hnf1a* upstream viewpoints and the intronic E enhancer. UMI-4C contact trends with binomial standard deviation for the V1 and V2 viewpoints are shown (*n* = 6 for the wild type and *n* = 3 for mutant livers). Triangles denote viewpoints (DpnII fragment ± 1 kb) and asterisks mark E. The bottom panel shows liver H3K4me3. The brown shading shows the region deleted in *Haster*[LKO]. **b,** UMI normalized counts at E showed increased contacts with upstream regions (V1 and V2) in *Haster*[LKO] liver. Statistical significance was determined by $\chi^2$ tests for *n* = 6 wild-type and mutant livers (V1) and *n* = 3 wild-type and mutant livers (V2). **c,** *Haster*[LKO] cells have increased H3K4me3 in C and E (*n* = 3 biological replicates). The data are presented as means ± s.d. Statistical significance was determined by two-tailed *t*-test. **d,** Schematic depicting increased *Hnf1a* promoter–E interactions in *Haster*[LKO] liver. **e,f,** Doxycycline-induced HNF1A overexpression in *HASTER*-deleted EndoC-βH3 cells (*n* = 4) showing (**e**) normalized *HNF1A* mRNA levels and (**f**) *HNF1A* promoter viewpoint (triangle) UMI-4C contacts. The green shading shows a 5-kb region centred on E that was used to quantify *HNF1A* promoter interactions. Normalized UMI counts and $\chi^2$ test *P* values calculated with umi4c are shown on the right. **g,** E deletions

prevent *HNF1A* increases in *HASTER*-deleted cells. *HASTER*[+/+] or *HASTER*[ΔP1/ΔP1] clones were used to create polyclonal cells containing a mix of homozygous and heterozygous E deletions (ΔE) or wild-type sgGFP controls (WT). *HASTER* and *HNF1A* RNAs are shown as the fold change relative to parental *HASTER*[+/+] or *HASTER*[ΔP1/ΔP1] cells. ΔE significantly reduced *HASTER* but not *HNF1A* in *HASTER*[+/+] cells, yet it reduced *HNF1A* in *HASTER*[ΔP1/ΔP1] cells. Identical results were observed with a different clone, whereas C mutations had no effect (Extended Data Fig. 10f) (pool of *n* = 3 independent experiments with three pairs of sgRNAs for each deletion). In **e** and **g**, the data are presented as *TBP*-normalized relative expression (means ± s.d.) and statistical significance was determined by two-tailed *t*-test. **h,** *HASTER* exerts negative and positive feedbacks. At low HNF1A concentrations, *HNF1A* promoter–E interactions and transcription are unhindered, whereas at high HNF1A concentrations, HNF1A binding to *HASTER* limits *HNF1A*–E contacts, thereby decreasing *HNF1A* transcription. *HASTER* also acts as an essential enhancer in pancreatic lineages. **i,** *HASTER* is distinct from classic enhancers or silencers and is instead a *cis*-acting stabilizer that prevents overexpression and silencing.

Finally, this finding has general implications for our understanding of non-coding genome defects in disease. Unlike transcriptional enhancers, which often form clusters that provide robustness to genetic disruption[48,49], our findings indicate that the 320-bp *HASTER* promoter region lacks functional *cis*-regulatory redundancy. This warrants a need to examine lncRNA promoter sequence variation in human genomes.

## Online content

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

## Methods

### Animal studies

Animal experimentation was carried out in compliance with EU Directive 86/609/EEC and Recommendation 2007/526/EC and enacted under Spanish law 1201/2005. Experiments were approved by the animal care committees of the University of Barcelona and Parc de Recerca Biomedica de Barcelona. *Haster f* (LoxP) and *stop* alleles were generated in C57Bl/6N JM8.F6 embryonic stem cells by homologous recombination. Briefly, mouse embryonic stem cells were electroporated with a linearized targeting plasmid containing a LoxP-flanking *Haster* promoter or a transcription termination (3× SV40 polyA) signal downstream of the *Haster* promoter, as well as a phosphoglycerate kinase/neomycin selection cassette flanked by FRT recombination sites (Extended Data Fig. 3a). Constructs were linearized by PmeI and SacII (conditional allele) or PacI and PmeI (stop allele). Electroporated embryonic stem cells were selected for the cassette with geneticin. Clones were analysed by Southern blot and targeted clones were injected into C57BL/6BrdCrHsd-Tyrc morulae (E2.5) to create chimeric mice that transmitted the recombined allele through the germline. The phosphoglycerate kinase/neomycin cassette was excised by crossings with Tg(CAG-Flp) mice. Mice were bred on C57BL/6 backgrounds unless otherwise specified.

To excise *Haster* in pancreatic epithelial cells, *Haster*[+/f] mice were crossed with *Pdx1*-Cre mice (Tg(Pdx1-Cre)6[Tuv])[50]. Constitutive excision in β cells was achieved with *Ins1*[Cre] knock-in mice (*Ins1*[tm1.1(cre)Thor])[51]. Inducible excision in β cells was achieved with the *Pdx1*-CreER transgene (Tg(Pdx1-Cre/Esr1*)[1Mga])[52] after 40 µg oral tamoxifen (Merck) twice, spaced by 4 d in 10- to 13-week-old mice, and analysed 12 weeks later. Early liver deletion was achieved with *Alb*-Cre mice, in which Cre is driven by an albumin promoter and alpha-fetoprotein enhancer (Alb Tg(Alb1-cre)1[Khk])[33]. *Haster* germline deletions were generated by breeding *Haster*[+/f] mice with Tg(EIIa-cre)[53]. *Hnf1a*[+/−] mice have been described[24]. Genotyping primers are provided in Supplementary Table 5.

Lines with LoxP alleles without Cre, Cre lines without LoxP alleles and wild-type littermates served as controls, as indicated. Experimental cohorts were maintained on a 12 h light/12 h dark cycle with free access to water and standard mouse chow. Before decapitation, mice were anaesthetized using isoflurane (Zoetis).

### Glucose tolerance

Animals were fasted overnight and received intraperitoneal glucose injections (2 g kg$^{-1}$) or were re-fed before blood glucose was collected at the indicated time points. Glucose was measured with a GlucoMen Aero 2K meter (Menarini Diagnostics). Plasma insulin was quantified with the Ultra Sensitive Mouse Insulin ELISA kit (Crystal Chem) using an Infinite M Plex (Tecan) plate reader. Standard curves were fitted using quadratic polynomial regression. Assays were performed in duplicate using 5 µl plasma from mouse tail, and mean values are reported.

### Islet isolation

Islet isolation was performed as described[54]. Briefly, ice-cold collagenase P solution (1 mg ml$^{-1}$ in Hanks' balanced salt solution (HBSS) buffer; Roche) was injected through the main duct. The inflated pancreas was dissected, incubated at 37 °C for 8 min with agitation, disaggregated by gentle suction through a needle, washed four times with cold HBSS with 0.5% bovine serum albumin (BSA) and resuspended in 7 ml 7:3 pre-cooled Histopaque 1077:Histopaque 1119 (Merck), then 7 ml HBSS with 0.5% BSA was layered on top. The gradient was centrifuged at 950*g* for 20 min at room temperature. The interphase containing islets was collected, washed three times with HBSS with 0.5% BSA and the islets were further enriched by aspiration under a stereomicroscope. Islets were cultured for 2 d in 11 mM glucose RPMI with 10% foetal calf serum and penicillin–streptomycin (1:100; Invitrogen) at 37 °C and under 5% $CO_2$.

### Vectors

We generated lentiviral vectors with a human insulin promoter driving the expression of Cas9 or dCas9, in addition to a U6-driven single guide RNA (sgRNA) (pLV-hIP-Cas9-BSD (plasmid 183230; Addgene) and pLV-hIP-dCas9-BSD (plasmid 183231; Addgene)). The EF1a promoter and puromycin resistance of lentiCRISPRv2 vector (plasmid 52961; Addgene) were replaced by a human insulin promoter and blasticidin-S deaminase (BSD) using Gibson Assembly (NEBuilder HiFi DNA Assembly Master Mix) to generate pLV-hIP-Cas9-BSD. The insulin promoter (343 bp) was amplified from EndoC-βH3 DNA and BSD from Addgene 61425. Cas9 from pLV-hIP-Cas9-BSD was replaced by dCas9 using Gibson Assembly to generate pLV-hIP-dCas9-BSD. dCas9 was amplified from pSp-dCas9-2A-GFP[8].

sgRNAs (20 nucleotides) for CRISPRi *roadblock* were designed within 100 bp downstream of the islet CAGE transcriptional start site using Cas-Designer (http://www.rgenome.net/cas-designer/) and cloned as described[55]. Briefly, oligonucleotides (Thermo Fisher Scientific) containing sgRNAs flanked by compatible overhangs were phosphorylated with T7 polynucleotide kinase (NEB) and annealed. Oligonucleotide duplexes were ligated into BbsI- or BsmBI-digested destination vectors. Ligated constructs were transformed into Stbl3 chemically competent *Escherichia coli* and clones were sequenced. For deletions, sgRNA pairs were cloned as described[56,57]. Briefly, a fragment containing the scaffold of sgRNA1 and the H1 promoter of sgRNA2 were amplified from the pScaffold-H1 donor (118152; Addgene) with primers containing the protospacer of the sgRNA1, sgRNA2 and BbsI restriction sites. The PCR fragment was digested with BbsI and ligated into the destination vector.

A TetOn-HNF1A lentiviral vector (pLenti-CMVtight-HNF1A-FLAG-Hygro; 183232; Addgene) was built by cloning human HNF1A-FLAG into pLenti CMVtight Hygro DEST (26433; Addgene). Reverse tetracycline-controlled transactivator (rtTA) was expressed from pLV-rtTA-zeo (183233; Addgene), built by amplifying the UbC promoter and rtTA-Advance cassette from pHAGE-TRE-dCas9-KRAB (50917; Addgene) and cloned into a lentiviral backbone upstream from the 2A-ZeocinR cassette. Sequences are listed in Supplementary Table 6.

HNF1A IDRs were predicted using MobiDB-lite (http://old.protein.bio.unipd.it/mobidblite/) from InterProt. IDR1 comprised amino acids 283–358 and IDR2 comprised amino acids 545–573. Only IDR2 deletions showed significantly decreased transcription and are thus shown. Vectors carrying deletions or fusions were built using Gibson Assembly (NEBuilder HiFi DNA Assembly Master Mix) using pcDNA3.1 as the backbone (183234–183239; Addgene). A carboxy-terminal FLAG tag and the 3′ untranslated region from the *Xenopus* globin gene were added. The FUS IDR[58] was codon optimized and synthetized as gBlock (IDT).

### Cell culture

EndoC-βH3 cells[59] were maintained on a 2 µg ml$^{-1}$ fibronectin- and 1% extracellular matrix-coated plate in Dulbecco's modified Eagle medium (DMEM) low glucose (1 g l$^{-1}$), sodium pyruvate (Thermo Fisher Scientific), 2% BSA Fraction V (Roche), 1% heat-inactivated foetal bovine serum (FBS; Labtech), 2 mM L-glutamine, 5.5 µg ml$^{-1}$ human transferrin, 1 mM sodium pyruvate 10 mM nicotinamide, 6.7 ng ml$^{-1}$ sodium selenite, 50 µM β-mercaptoethanol, 100 U ml$^{-1}$ penicillin and 100 µg ml$^{-1}$ streptomycin. DMEM was substituted with Advance DMEM/F-12 (Thermo Fisher Scientific) and FBS was omitted for the TetOn-HNF1A EndoC-βH3 cell line, as well as during the expansion of EndoC-βH3 clones.

293FT cells (Thermo Fisher Scientific) were maintained in DMEM, 10% heat-inactivated FBS, 0.1 mM MEM non-essential amino acids, 2 mM L-glutamine, 1 mM sodium pyruvate, 500 µg ml$^{-1}$ geneticin, 100 U ml$^{-1}$ penicillin and 100 µg ml$^{-1}$ streptomycin.

MIN6 cells[60] were maintained in DMEM, 4.5 g l$^{-1}$ glucose, 15% heat-inactivated FBS, 50 μM β-mercaptoethanol and 50 μg ml$^{-1}$ gentamicin.

## Gene perturbations in EndoC-βH3 cells

LNA GapmeRs (Exiqon; Supplementary Table 7) and plasmid vectors were nucleofected in EndoC-βH3 cells using Nucleofector B2 with an Amaxa Cell Line Nucleofection Kit V (Lonza) and program G-017. We used 2 million cells and 10 μg plasmid DNA per nucleofection for deletions or 1 million cells with 250 pg LNA GapmeRs. Cells were harvested 72 h after GapmeR or 48 h after plasmid nucleofection.

CRISPRi and CRISPR–SAM lentiviral particles were produced as described[56]. 293FT cells were seeded at 75,000 cells per cm$^2$ in T75 flasks and, 24 h later, transfected with CRISPR and packaging plasmids pMDLg/pRRE, pRSV-Rev and pMD2.G (12251, 12253 and 12259; Addgene) with PEIpro (Polyplus-transfection) in antibiotic-free media using a 1:1 ratio of total μg DNA to μl PEIpro. The medium was replaced with 9 ml fresh 293FT antibiotic-free media 18 h post-transfection and lentiviral particles were collected 72 h post-transfection. Immediately after collection, the supernatants were centrifuged for 5 min at 400$g$ and filtered using 0.45-μm pore size Steriflip-HV polyvinylidene fluoride filters (Millipore). The supernatant was supplemented with 1 mM MgCl$_2$ and treated with 1 μg ml$^{-1}$ DNase I (Roche) for 20 min at 37 °C. Viral particles were concentrated with 1:3 vol/vol of Lenti-X Concentrator (Clontech) at 4 °C overnight. On the following day, virus particles were collected for 45 min at 1,500$g$ and 4 °C, resuspended in 100 μl phosphate-buffered saline (PBS), aliquoted and stored at −80 °C. Transduction was carried out with 10 μl virus for 400,000 cells in 1 ml. Antibiotic selection was started 3 d later with 8 μg ml$^{-1}$ blasticidin, 100 μg ml$^{-1}$ hygromycin or 200 μg ml$^{-1}$ zeocin for EndoC-βH3 cells.

A CRISPR–SAM cell line was established by successive transduction of lentivirus dCAS-VP64_Blast (61425; Addgene) and MS2-P65-HSF1_Hygro (61426; Addgene). Cells were transduced with lentivirus sgRNA(MS2)_zeo (61427; Addgene) expressing gene promoter-targeting sgRNAs. The CRISPR–Cas9 and CRISPRi *roadblock* experiments were performed with lentivirus hIP-Cas9-BSD (pLV-hIP-Cas9-BSD plasmid) and hIP-dCas9-BSD (pLV-hIP-dCas9-BSD plasmid).

For the CRISPR–Cas9 clonal deletions, EndoC-βH3 cells were nucleofected with pSpCas9(BB)-T2A-HygR (118153; Addgene) containing sgRNAs. At 24 h after nucleofection, cells were selected using Hygromycin B (200 μg ml$^{-1}$; Thermo Fisher Scientific; 10687010) for 3 d. After 2 weeks, the cells were seeded at low density (2–9 cells per cm$^2$) in Advance DMEM/F-12-based EndoC-βH3 medium (Gibco). EndoC-βH3 clones were hand picked and transferred into 96-well plates. After genotyping, selected clones were expanded in DMEM-based 1% FBS medium.

Doxycycline-inducible HNF1A EndoC-βH3 cells were established by successive transduction with rtTA-2A-ZeoR-expressing lentivirus (pLV-rtTA-zeo) and TRE-HNF1A-FLAG lentivirus carrying the hygromycin resistance (pLenti-CMVtight-HNF1A-FLAG-Hygro). Cells were exposed to doxycycline (0, 25, 50, 100, 200 or 400 ng ml$^{-1}$) for 24 h and endogenous *HNF1A* mRNA was detected by PCR with oligonucleotides recognizing a 3′ untranslated region that is not present in HNF1A-FLAG. Exogenous HNF1A-FLAG was detected by PCR with oligonucleotides specific for the FLAG region.

## hESC genome editing

H9 hESCs were maintained in mTeSR1 medium (85870; STEMCELL Technologies) on a Matrigel (356231; Corning)-coated plate. For nucleofection, cells were dissociated with Accutase (Merck) for 8 min at 37 °C, diluted in 10 μM Y27632 (Merck) mTeSR1, centrifuged at 110$g$ for 3 min and resuspended in 10 μM Y27632 mTeSR1. A total of 10$^6$ cells were nucleofected using Human Stem Cell Nucleofector Kit 2 (program G-017; Lonza) with 5 μg pSpCas9(BB)-2A-puro (62988; Addgene) expressing two sgRNAs. After nucleofection, the cells were transferred to a 12-well plate containing 1 ml 10 μM Y27632 mTeSR1. After 24 h, the

cells were selected for puromycin resistance by replacing the medium with 10 μM fresh Y27632 mTeSR1 containing 0.5 μg ml$^{-1}$ puromycin for 24 h. After selection, the hESCs were cultured in mTeSR1 without Y27632. After two passages, the cells were dissociated and plated at low density. Isolated clones were transferred and maintained in 96-well plates until genotyping.

## hESC differentiation

H9 mutant clones were differentiated to hepatocytes using a protocol adapted from Hannan et al.[32]. Cells were seeded at 300,000 cells per 24-well plate in 10 μM Y27632 mTeSR1 and differentiation was started after 24 h. The following media were used for differentiation: (1) S1 medium[61] was prepared with MCDB 131 Medium (10372019; Thermo Fisher Scientific) supplemented with 8 mM D-(+)-Glucose (G7528; Merck), 2.46 g l$^{-1}$ NaHCO$_3$ (S3817; Merck), 2% BSA Fraction V (10735078001; Roche), 1:50,000 Insulin-Transferrin-Selenium-Ethanolamine (ITS-X) (51500056; Thermo Fisher Scientific), 2 mM GlutaMAX (35050061; Thermo Fisher Scientific) and 0.25 mM L-ascorbic acid (A4544; Merck); (2) RPMI/B27 medium was prepared with RPMI 1640 Medium, GlutaMAX Supplement (61870010; Thermo Fisher Scientific) supplemented with B-27 Supplement (17504044; Thermo Fisher Scientific) and MEM Non-Essential Amino Acids Solution (11140035; Thermo Fisher Scientific); and (3) hepatocyte growth medium (HGM) was prepared with HBM Basal Medium (CC-3199; Lonza) supplemented with 3.75 g ml$^{-1}$ BSA Fraction V, 250 μg ml$^{-1}$ L-ascorbic acid, 10 μg ml$^{-1}$ holo-Transferrin (T0665; Merck), 0.5 μg ml$^{-1}$ Hydrocortisone (H0888; Merck), 5 μg ml$^{-1}$ human Insulin and 10 ng ml$^{-1}$ epidermal growth factor (236-EG-200; R&D Systems). During differentiation, the medium was changed every day, or every 2 d after day 11, using the following media: S1 medium with 100 ng ml$^{-1}$ Activin A (338-AC-050; R&D Systems) and 3 μM CHIR 99021 (04-0004; Tocris Bioscience) for day 1; S1 medium with 100 ng ml$^{-1}$ Activin A for days 2 and 3; RPMI/B27 medium with 50 ng ml$^{-1}$ Activin A for days 4–6; RPMI/B27 medium with 20 ng ml$^{-1}$ BMP-4 (314-BP-010; R&D Systems) and 10 ng ml$^{-1}$ FGF-10 (ABE1324; Source BioScience) for days 7–10; and HGM medium with 30 ng ml$^{-1}$ Oncostatin M (295-OM-010; R&D Systems) and 50 ng ml$^{-1}$ HGF (100-39; PeproTech) for days 11–25. The definitive endoderm stage was reached at day 4, the anterior endoderm stage was reached at day 7, the hepatoblast stage was reached at day 11 and the hepatocyte stage was reached at day 26.

Pancreatic differentiations were performed using a modification of a published protocol[40]. Dissociated hESCs were seeded at 2 million cells per 35 mm well coated with Matrigel in 5 μM Y27632 E8 medium (A1517001; Thermo Fisher Scientific). Differentiation was started the following day after washing the cells once with 1× PBS: Definitive endoderm induction was as follows: MCDB 131, 2 mM GlutaMax (35050038; Thermo Fisher Scientific), 1.5 g l$^{-1}$ NaHCO$_3$, 0.5% BSA Fraction V (7500804; Lampire Biological Laboratories), 10 mM final glucose, 100 ng ml$^{-1}$ Activin A (QK001; Qkine) and 3 μM CHIR 99021 (4423; Tocris Bioscience) on day 0; as for day 0 but reducing CHIR 99021 to 0.3 μM on day 1; and as for day 1 but with no CHIR 99021 on day 2. Stage 2 posterior foregut induction was as follows: MCDB 131, 2 mM GlutaMax, 1.5 g l$^{-1}$ NaHCO$_3$, 0.5% BSA Fraction V, 10 mM final glucose, 0.25 mM ascorbic acid (A4544; Sigma–Aldrich) and 50 ng ml$^{-1}$ FGF-7 (Z03407-1; GenScript) on days 3–5. Stage 3 pancreatic endoderm induction was as follows: MCDB 131, 2 mM GlutaMax, 2.5 g l$^{-1}$ NaHCO$_3$, 2% BSA Fraction V, 10 mM final glucose, 0.25 mM ascorbic acid, 50 ng ml$^{-1}$ FGF-7, 0.25 μM SANT-1 (S4572; Sigma–Aldrich), 1 μM retinoic acid (R2625; Sigma–Aldrich), 100 nM LDN193189 (S2618; Selleckchem), 1:200 ITS-X (51500056; Thermo Fisher Scientific) and 200 nM TPB (sc-204424; Santa Cruz Biotechnology) on days 6 and 7. Stage 4 pancreatic progenitor induction was as follows: MCDB 131, 2 mM GlutaMax, 2.5 g l$^{-1}$ NaHCO$_3$, 2% BSA Fraction V, 10 mM final glucose, 1:200 ITS-X, 0.25 mM ascorbic acid, 2 ng ml$^{-1}$ FGF-7, 0.25 uM SANT-1, 0.1 μM retinoic acid, 200 nM LDN, 100 nM TPB, 100 ng ml$^{-1}$ epidermal growth factor

(AF-100-15; PeproTech), 10 mM nicotinamide (N0636; Sigma–Aldrich), 10 ng ml$^{-1}$ Activin A and 10 μM Y27632 on days 8–11. Cells were dissociated with TrypLE and seeded in AggreWell 400 plates (34425; Stem Cell Technologies) on day 10.

## Reverse transcription quantitative PCR

RNA was prepared using an RNeasy Mini Kit (Qiagen) and DNAse I (Qiagen) and retrotranscribed with SuperScript III (Thermo Fisher Scientific) and random hexamers (Thermo Fisher Scientific). Quantitative PCR was performed with Universal Probe Library assays (Roche). Reactions were carried out in duplicate in a QuantStudio 12K Flex (Applied Biosystems) with 1× TaqMan Fast Advanced Master Mix (Thermo Fisher Scientific), 1 μM forward and reverse primers and 250 nM Universal Probe Library probe, or 1× TaqMan assay. Quantification was performed using standard curves, with duplicate means reported, normalized by *TBP* or *RPLPO*, as indicated. Oligonucleotides are listed in Supplementary Table 5.

## Single-molecule fluorescence in situ hybridization

Single-molecule fluorescence in situ hybridization was performed as described[62]. A set of 48 probes (Supplementary Table 8), coupled with Quasar 570 (548/566) or Quasar 670 (647/670), were designed for each transcript (Stellaris RNA FISH probes; LGC Biosearch Technologies). EndoC-βH3 cells were grown on coated (2 μg ml$^{-1}$ fibronectin and 1% extracellular matrix; Merck) coverslips. Cells were fixed in 4% formaldehyde for 2 min, washed with 1× PBS and permeabilized with 70% ethanol at 4 °C for >1 h. Probes were hybridized overnight at 37 °C in the dark with 10% formamide, 100 mg ml$^{-1}$ dextran sulfate, 2× SSC and 12.5 μM probes. The following day, cells were washed for 30 min at 37 °C with 10% formamide and 2× SSC, followed by 30 min with 5 ng ml$^{-1}$ 4′,6-diamidino-2-phenylindole (DAPI). Coverslips were mounted using VECTASHIELD HardSet mounting media. Acquisitions were performed on a Zeiss Axio Observer inverted widefield microscope with light-emitting diode illumination. Z-stack acquisitions were taken with a 63× objective every 0.5 μm from a total depth of 40 μm and deconvoluted (Huygens Software) and maximal projections of whole stacks were used for counting (8–12 fields per sample).

## Immunofluorescence

Embryos and adult tissues were processed for immunofluorescence as described[63]. Briefly, tissues were fixed in 4% paraformaldehyde overnight at 4 °C, then washed in PBS before paraffin embedding. Deparaffinized sections (4 μm) were incubated for 30 min in antibody diluent (Dako) with 3% normal serum from the same species as the secondary antibody, incubated overnight at 4 °C with primary antibody and then overnight at 4 °C with secondary antibody, then DAPI stained and mounted with Mounting Medium (S3023; Molecular Probes). The primary antibodies were: HNF1A (1:400; D7Z2Q; Cell Signaling Technology), insulin (1:200; A0564; Dako), glucagon (1/1,000; 4030-01F; Millipore), Cytokeratin 19 (1/100; TROMA-III-c; Hybridoma Bank), PDX1 (1:200; AF2419; R&D Systems) and NKX6.1 (1:200; F55A10; Hybridoma Bank). The following secondary antibodies were used: Donkey anti-rabbit Alexa Fluor 488 (1/800; 711-545-152; Jackson ImmunoResearch) and Cy3 (1/400; 711-166-152; Jackson ImmunoResearch), Donkey anti-guinea pig Alexa Fluor 488 (1/800; 706-545-148; Jackson ImmunoResearch) and Cy5 (1/400; 706-175-148; Jackson ImmunoResearch), Donkey anti-goat Alexa Fluor 488 (1/800; 705-545-147; Jackson ImmunoResearch), Donkey anti-rat Cy3 (1/400; 712-165-153; Jackson ImmunoResearch) and Donkey anti-mouse Cy5 (1/400; 715-175-151; Jackson ImmunoResearch). Images were acquired using a Leica TSE confocal microscope for tissues and a Leica DMi8 for cell lines.

## Western blots

Proteins were extracted from frozen mouse livers with 9 M urea. Quantification was performed using a Microplate BCA Protein Assay Kit (23250; Thermo Fisher Scientific). Western blot was performed with 20 μg protein on 4–12% Bis-Tris gel (NP0335BOX; Thermo Fisher Scientific) using β-tubulin (2146; Cell Signaling Technology) and HNF1A antibodies (89670; Cell Signaling Technology) and Goat Anti-Rabbit IgG H&L (HRP) (1/2,000; ab97051; Abcam).

## Cellular fractionation

Cellular fractionation was performed as described[64]. Some 5 million EndoC-βH3 cells were incubated for 5 min on ice in 200 μl cold lysis buffer (10 mM Tris-HCl pH 7.5, 0.05% IGEPAL, 150 mM NaCl and 100 U ml$^{-1}$ SuperaseIn (Thermo Fisher Scientific)). The lysate was layered over 2.5 volumes of chilled sucrose solution (10 mM Tris-HCl pH 7.5, 0.05% IGEPAL, 150 mM NaCl, 24% sucrose and 100 U ml$^{-1}$ SuperaseIn) then centrifuged for 10 min at 15,000*g* and 4 °C. The cytoplasmic supernatant was kept and the pellet was washed with 500 μl wash buffer (1 mM ethylenediaminetetraacetic acid (EDTA) in PBS pH 7.5), then centrifuged for 10 min at 15,000*g* and 4 °C. This pellet was resuspended in 100 μl cold glycerol buffer (20 mM Tris-HCl pH 7.5, 75 mM NaCl, 0.5 mM EDTA, 0.85 mM dithiothreitol, 1× protease inhibitor cocktail (Roche), 50% glycerol and 100 U ml$^{-1}$ SuperaseIn), and 100 μl cold nuclei lysis buffer (10 mM HEPES pH 7.5, 1 mM dithiothreitol, 7.5 mM MgCl$_2$, 0.2 mM EDTA pH 8, 0.3 mM NaCl, 1 M urea and 1% IGEPAL) was added to the nuclei suspension, vortexed and left on ice for 2 min. Nuclear lysate was centrifuged for 2 min at 15,000*g* and 4 °C and the supernatant was collected as nucleoplasmic fraction. The pellet (chromatin fraction) was washed with 500 μl wash buffer and resuspended in 300 μl chromatin DNase buffer (20 mM Tris-HCl pH 7.5, 50 mM KCl, 4 mM MgCl$_2$, 0.5 mM CaCl$_2$, 2 mM TCEP (Merck), 1× protease inhibitor cocktail, 0.4% sodium deoxycholate, 1% IGEPAL and 0.1% *N*-lauroylsarcosine). Next, 15 μl murine RNase inhibitor (NEB) and 30 μl TURBO DNase (Ambion) were added and the reaction was incubated for 20 min at 37 °C. DNase was inactivated with 12.5 μl 25× Stop Solution (250 mM EDTA and 125 mM ethylene glycol tetraacetic acid). Proteins were digested with 7.5 μl proteinase K (Ambion) for 1 h at 37 °C. RNA from the different fractions was purified using an RNA Clean & Concentrator-25 Kit (Zymo Research).

## 3′ RACE

3′ RACE was performed as described[65]. Human islet RNA (240 ng was retrotranscribed with Q$_T$ primers using SuperScript III. Nested PCRs were performed with Q5 polymerase (NEB). The first PCR used one-twentieth of complementary DNA with a gene-specific forward primer 1 and a Q$_O$ reverse primer, while the second PCR used 1 μl of a 1:5 dilution of the first PCR with a gene-specific forward primer 2 and a Q$_I$ reverse primer. The resulting fragments were cloned and Sanger sequenced. Oligonucleotides are provided in Supplementary Table 5.

## Chromatin immunoprecipitation

Liver was collected after perfusion of ice-cold PBS and minced with a razor blade. Minced liver (100 mg) or 100–500 mouse islets were incubated with 1% formaldehyde (Agar Scientific) for 10 min at room temperature, then one-tenth of 1.25 M glycine was added for 5 min at room temperature, pelleted at 800*g* and 4 °C for 3 min and washed twice with PBS. Aliquots containing 20 mg initial liver or all processed islets were snap-frozen and stored at −80 °C until use. Crosslinked samples were lysed using ice-cold 2% Triton X-100, 1% sodium dodecyl sulfate (SDS), 100 mM NaCl, 10 mM Tris-HCl pH 8, 1 mM EDTA pH 8 and 1× protease inhibitor cocktail for 15–20 min on ice. Chromatin was sonicated with a Covaris S220 Focused-ultrasonicator (2% duty factor; 105 W peak incident power; 200 cycles per bust; 16 min). Sheared chromatin was centrifuged at full speed for 10 min at 4 °C to remove debris and insoluble chromatin and the supernatant was transferred to a fresh low-binding tube. For liver, the chromatin equivalent of 5 μg DNA was used for one-histone-mark chromatin immunoprecipitation (ChIP) and 10 μg was used for transcription factor ChIP. Chromatin was diluted four times with ChIP Dilution Buffer (0.75% Triton X-100,

0.1% sodium deoxycholate, 140 mM NaCl, 50 mM HEPES pH 8, 1 mM EDTA and 1× protease inhibitor cocktail) and 5% was used as input. Dynabeads Protein G (30 µl; Thermo Fisher Scientific) were blocked with BSA overnight at 4 °C. HNF1A antibody (10 µl; D7Z2Q; Cell Signaling Technology), 2 µg H3K27ac antibody (ab4729; Abcam) and 2 µg H3K4me3 antibody (15-10C-E4; Merck) or 2 µg H3K4me1 antibody (ab8895; Abcam) were added to 500 µl samples and incubated overnight with rotation at 4 °C. Magnetic beads (30 µl) were added to the samples and rotated at 4 °C for 2 h.

For ChIP-quantitative PCR (ChIP-qPCR), antibody-incubated samples were washed with low-salt wash buffer (1% Triton X-100, 0.1% SDS, 150 mM NaCl, 20 mM Tris-HCl pH 8 and 2 mM EDTA pH 8), high-salt wash buffer (1% Triton X-100, 0.1% SDS, 500 mM NaCl, 20 mM Tris-HCl pH 8 and 2 mM EDTA pH 8), LiCl wash buffer (0.25 M LiCl, 1% IGEPAL, 1% sodium deoxycholate, 10 mM Tris-HCl pH 8 and 1 mM EDTA pH 8) and three times with TE buffer. Elution was performed with 200 µl 1% SDS and 0.1 M NaHCO$_3$ for 30 min at room temperature. Samples were placed on a magnet and the supernatant was transferred to a new tube. RNase A (1 µl; Thermo Fisher Scientific) was added to the eluate and incubated for 30 min at 37 °C. Reverse crosslink was performed by adding 8 µl 5 M NaCl and 3 µl proteinase K (Thermo Fisher Scientific) and incubation was performed for 1 h at 55 °C and 1,200 r.p.m., then overnight at 65 °C and 1,200 r.p.m. DNA was purified using a MinElute PCR Purification Kit (Qiagen). Quantitative PCR was carried out in duplicates as described for reverse transcription qPCR. Allele-specific qPCR was performed using Custom TaqMan SNP Genotyping Assays. Enrichment was subsequently normalized by the input.

For ChIPmentation, washes and tagmentation were performed as reported[66]. Antibody-incubated samples were washed twice with RIPA-LS (10 mM Tris-HCl pH 8, 140 mM NaCl, 1 mM EDTA pH 8, 0.1% SDS, 0.1% sodium deoxycholate and 1% Triton X-100), twice with RIPA-HS (10 mM Tris-HCl pH 8, 500 mM NaCl, 1 mM EDTA pH 8, 0.1% SDS, 0.1% sodium deoxycholate and 1% Triton X-100), twice with RIPA-LiCl (10 mM Tris-HCl pH 8, 250 mM LiCl, 1 mM EDTA pH 8, 0.5% IGEPAL and 0.5% sodium deoxycholate) and once with 10 mM Tris-HCl pH 8. Beads were resuspended in 20 µl tagmentation solution (10 mM Tris-HCl pH 8, 5 mM MgCl$_2$ and 10% vol/vol dimethylformamide) containing 1 µl Tn5 (Illumina) and incubated at 37 °C for 10 min. The reaction was stopped with 1 ml ice-cold RIPA-LS for 5 min on ice. Beads were washed twice with RIPA-LS and twice with TE buffer and resuspended in elution buffer (10 mM Tris-HCl pH 8, 5 mM EDTA pH 8, 300 mM NaCl and 0.4% SDS). Proteinase K was added to the elution and incubated for 1 h at 55 °C and 1,200 r.p.m., then overnight at 65 °C and 1,200 r.p.m. DNA was purified using a MinElute PCR Purification Kit (Qiagen). To estimate the number of cycles required for library amplification, 2 µl of the elution was used for SYBR Green qPCR, using KAPA HiFi polymerase (Kapa Biosystems). The resulting Ct value plus 1 cycle was used as the number of cycles to amplify the library. Libraries were amplified from 20 µl of elution with KAPA HiFi polymerase and Nextera custom primers (Supplementary Table 5). DNA clean-up was performed with 1.8× volume and size selection with a 0.65× volume of AMPure XP beads (Beckman Coulter). Libraries were sequenced on a HiSeq 2500 using 1 × 50 bp reads.

## ChIP sequencing
ChIP sequencing (ChIP-seq) reads were aligned with Bowtie 2 (version 2.3.5) on the GCRm38 genome and sorted using SAMtools (version 1.7). Alignment statistics are listed in Supplementary Table 2. Multi-mapped reads were discarded. Reads mapping to ENCODE blacklisted regions were removed using BEDTools (version 2.27.1) and duplicated reads were removed with Picard (version 2.6.0). Peak calling was performed using MACS2 (version 2.1.1) with an FDR ($q$ value) threshold of 0.05. The --broad flag was used for histone modifications. The MACS2 bdgcmp function was used to generate the local Poisson test −log$_{10}$[$P$ values]. $P$ value BedGraphs were converted to bigWig using bedGraphToBigWig. Differential binding was performed using DiffBind (version

2.8.0) on peaks called in at least two samples from any genotypes, using normalized read coverage from triplicates. Binding differences were determined at $q \leq 0.05$. HNF1A neo-binding sites were defined as peaks observed in at least two *Haster* knockout samples ($q \leq 0.05$), without significant peaks in any control sample and with average log$_2$-normalized ChIP read counts of ≤2 in control samples. Activated promoters were similarly defined as H3K4me3 peaks ($q \leq 0.05$) in two *Haster* knockout and no control samples, with log$_2$ normalized counts of <2 in controls and significant differential H3K4me3 enrichment ($q \leq 0.05$) in *Haster* versus controls. Coverage was calculated using deepTools (version 3.0.2) computeMatrix and the average of the three replicates was calculated for each bin. Peak intersections were performed with pybedtools (version 0.8.0).

## Motif analysis
Analysis of known and de novo transcription factor binding site motifs was performed using HOMER (version 3.12). Analyses were performed on the merge between overlapping consensus peaks defined by DiffBind, using a minimum overlap of 1 bp. Enrichment analysis of de novo transcription factor motifs was also performed with the findMotifsGenome.pl command on consensus peaks defined by DiffBind, for lengths of 8, 10 and 12 bp on the masked mm10 genome.

## Assay for transposase-accessible chromatin with high-throughput sequencing
Reads were trimmed to remove adaptors using Trim Galore and aligned with Bowtie 2 (version 2.3.5) on the GCRm38 genome. Multi-mapped and duplicated reads were removed using Picard (version 2.6.0). Mitochondrial and ENCODE blacklisted region reads were discarded. For visualization, MACS2 bdgcmp was used to generate the local Poisson test −log$_{10}$[$P$ value] bedGraphs. BedGraphs were converted to bigWig using bedGraphToBigWig. The coverage was calculated using deepTools (version 3.0.2) computeMatrix for 1-kb windows with 10-bp bins.

## RNA-seq
RNA from islets or liver was quantified with Qubit (Thermo Fisher Scientific) and verified with Bioanalyzer (Agilent). Libraries were prepared with a TruSeq Stranded mRNA Library Kit and sequenced on a HiSeq 4000 (2 × 75 bp reads). Reads were aligned to the GCRm38 genome with STAR (version 2.3.0). Transcript-level quantification was performed with Salmon (version 0.11) using GENCODE GCRm38 VM18 annotations (Supplementary Table 2). Gene-level normalization and differential expression were performed using the Bioconductor R (version 3.6.1) package DESeq2 (version 1.24.0), using adjusted $P \leq 0.05$ as a cut-off for differentially expressed genes. Fold changes were adjusted with lfcShrink using the apeglm option[67].

For differential expression, de novo transcripts from *Haster*$^{LKO}$ and control liver were assembled from RNA-seq using StringTie (version 2.0). Transcripts from *Haster*$^{LKO}$ and control replicates were merged in a single GTF file using gffcompare (version 0.10.1). Transcript quantification and differential transcript expression were performed using Salmon and DESeq2 as described above, using the merged *Haster*$^{LKO}$ and control liver transcriptome as a reference. Transcripts with low abundance (mean normalized transcripts per million< 3) were discarded. To define transcripts with an HNF1A-bound promoter, a minimum overlap of 1 bp between the transcription start site and an HNF1A peak was required.

Human, chicken, *Xenopus tropicalis* and zebrafish liver RNA-seq reads were aligned on the GCRh37 (hg19), galGal5, XenTro9 and GRCz11 (danRer11) genomes, respectively. Mouse kidney and small intestine RNA-seq reads were aligned to the GCRm38 (mm10) genome.

## Allele-specific RNA-seq
Stranded total RNA libraries from C57BL/6;PWK/PhJ F1 liver and islets were sequenced on a HiSeq 2500 using 2 × 125 bp reads. Reads were

aligned by STAR (version 2.7.6) using WASP[68], and reads that aligned to a different genomic region after swapping the C57BL/6 variant to the PWK/PhJ variant were discarded. PWK/PhJ single-nucleotide polymorphisms were obtained from the Mouse Genome Project (version 5)[69]. ASEReadCounter (GATK version 4.1.9.0) was used to count the reads overlapping *Hnf1a* and *Haster* PWK/PhJ exonic single-nucleotide polymorphisms.

## HNF1A-regulated gene set

To define a high-confidence islet HNF1A-dependent gene set, we intersected: (1) 115 downregulated genes from *Hnf1a*$^{-/-}$ islets[22]; and (2) 570 genes that showed increased expression (adjusted $P \le 0.05$ and mean normalized expression > 500) after CRISPR–SAM activation of *Hnf1a* in MIN6 mouse β cells. This MIN6-SAM cell line was generated by successive transduction of MIN6 cells with lentivirus dCAS-VP64_Blast (61425; Addgene), followed by blasticidin selection (1 μg ml$^{-1}$) and lentivirus MS2-P65-HSF1_Hygro (61426; Addgene), followed by hygromycin selection (100 μg ml$^{-1}$). MIN6-SAM cells were subsequently transduced with sgRNA expressing vector lenti sgRNA-(MS2)-zeo (61427; Addgene). RNAs from triplicates of two independent *Hnf1a*-activating sgRNAs and two independent control sgRNAs were used for RNA-seq. In total, 21 genes showed concordant downregulation and upregulation in both models (Supplementary Table 9).

## Gene set enrichment analysis and Enrichr

Gene set enrichment analysis (GSEA) was performed with GSEAPreranked (version 6.0; GenePattern)[70] on genes ranked by fold change, using default parameters over 10,000. Enrichments of functional annotations were performed with Enrichr[71].

## Tissue-specificity z score

Tissue-specificity $z$ scores were calculated for each gene by taking the average normalized gene expression in tissue minus the mean of all *Hnf1a*-expressing tissues divided by the standard deviation of all *Hnf1a*-expressing tissues[72].

## Single-cell RNA-seq

Cultured mouse islets were dissociated with Accutase (Merck) for 15 min at 37 °C. Islet cell suspensions were centrifuged at 600*g* for 3 min and resuspended in culture medium with DAPI before FACS sorting to remove dead cells and doublets. After sorting, cells were centrifuged at 600*g* for 3 min and resuspended in PBS/0.04% BSA. Single-cell libraries were generated with a 10X Genomics Chromium Single Cell 3′ Reagent Kit v3 following the manufacturer's instructions. Libraries were sequenced on a HiSeq 4000 using 2 × 75 bp reads.

## Single-cell RNA-seq analysis

Read alignments and UMI counts were performed with CellRanger (version 3.0.2) using the mm10 reference genome. Subsequent analyses were carried out with Seurat (version 3.0.1)[73] or scVI-tools (version 0.11.0)[74].

For Seurat, cells with <500 genes or >5% mitochondrial genes were filtered out (Supplementary Table 2). UMI counts were normalized using SCTransform[75]. To define shared populations between controls and knockouts, we performed an integrated analysis on the three control and three knockout datasets[76]. Briefly, the 3,000 most variable genes were used to find anchors (SelectIntegrationFeatures function using 50 dimensions). The first 50 principal components were used for t-distributed stochastic neighbour embedding projection (RunTSNE function) and clusters were defined by graph-based unsupervised clustering (FindClusters function) with a resolution of 0.5.

For scVI analysis, cells with <1,000 or >6,000 genes or >5% mitochondrial genes were filtered out. UMI counts were normalized for library size and log transformed. Integration of control and knockout samples was performed using the top 2,000 variable genes. The scVI

model was trained using ten dimensions of latent space and two hidden layers for the encoder and decoder neural network. The identification of HNF1A-deficient β cell clusters was robust to using Seurat or scVI (Extended Data Fig. 7).

Differential expression was performed with Seurat FindMarkers (min.pct = 0.1) for all combinations of controls versus knockouts. Wilcoxon rank-sum *P* values from the different combinations were combined using Fisher's method. Only genes with a consistent positive or negative fold change across all control or knockout combinations and with a combined $P \le 0.05$ were considered differentially expressed. All genes differentially expressed in endothelial cells were discarded. For differential expression, all β cell clusters with >250 cells were grouped in a single β cluster.

Seurat objects were exported as loom using as.loom of the loomR (version 0.2.0.1) library. Data visualization was performed with Python (version 3.7.3) and loompy (version 2.0.16), NumPy (version 1.15.4), pandas (version 0.25.0), Matplotlib (version 3.1.0) and seaborn (version 0.9.0) libraries. Statistics were computed using SciPy (version 1.1.0).

## UMI-4C

UMI-4C was performed as described[43] with modifications. Liver from three samples per genotype was crosslinked with 2% formaldehyde for 10 min, as described for ChIP. EndoC-βH3 cells were fixed with 1% formaldehyde for 10 min. Frozen pellets of ~10$^7$ cells were thawed on ice and resuspended in 5 ml cold lysis buffer (50 mM Tris-HCl pH 7.5, 150 mM NaCl, 5 mM EDTA, 1% Triton X-100, 0.5% IGEPAL and 1× protease inhibitor cocktail). After isolation, nuclei were resuspended in 650 μl nuclease-free water, 60 μl DpnII buffer and 15 μl 10% SDS and incubated at 37 °C and 900 r.p.m. for 1 h, with an additional hour after the addition of 75 μl 20% Triton X-100. The chromatin was digested at 37 °C and 900 r.p.m. for 24 h using 600 U DpnII (R0543L; NEB) and the enzyme was inactivated by incubating at 65 °C for 20 min. Ligation was performed in a final volume of 7 ml with 60 U T4 DNA ligase (Promega) and incubated at 16 °C overnight. The efficiency of the digestion and ligation was assessed by gel electrophoresis. Chromatin was reverse crosslinked with 30 μl proteinase K (10 mg ml$^{-1}$) overnight at 65 °C, followed by 45 min of incubation with 30 μl RNase A (10 mg ml$^{-1}$) at 37 °C. The DNA was purified by phenol–chloroform extraction followed by ethanol precipitation and resuspended in 10 mM Tris-HCl pH 8. Then, 10 μg DNA was sonicated using an S220 Focused-ultrasonicator (Covaris) to obtain 400- to 600-bp fragments. The DNA was end-repaired with 10 μl NEBNext End Repair Mix (E6050L; NEB) in a final volume of 200 μl, incubated for 30 min at 20 °C, purified with 2.2× AMPure XP beads (Beckman Coulter) and eluted in 10 mM Tris-HCl pH 8. A-tailing was performed with 200 U Klenow Fragment (M0212M; NEB) in 100 μl 1× NEBuffer 2 with 1 nM dATP. 5′ ends were dephosphorylated at 50 °C for 60 min with 20 U calf intestinal alkaline phosphatase (M0290S; NEB). The DNA was then cleaned with 2× AMPure XP beads. Adaptors were ligated with 0.4 μM Illumina-compatible forked indexed adaptors (Supplementary Table 5) and 10 μM quick ligase (M2200; NEB) in 160 μl 1× quick ligation buffer (M2200; NEB) for 15 min at 25 °C. DNA was denatured at 95 °C for 2 min and cleaned with 1× AMPure XP beads. To generate UMI-4C libraries, two nested PCRs were performed using GoTaq polymerase (Promega) with a final primer concentration of 0.4 mM. The first PCR used the upstream bait primer (Supplementary Table 5) and Illumina universal primer 2 and amplification was performed for 20 cycles. The DNA was cleaned with 1× AMPure XP and used for the second PCR with the downstream bait primer (Supplementary Table 5) and Illumina universal primer 2 for 16 cycles. After the second PCR, the DNA was cleaned and size selected with 0.7× AMPure XP beads. The size distribution of the libraries was controlled by Bioanalyzer and libraries were quantified with a KAPA Quantification Kit (07960166001; Roche). Libraries were sequenced on a HiSeq 2500 using 2 × 125 bp reads or a NovaSeq S4 using 2 × 150 bp reads.

## UMI-4C-seq analysis

Umi4cPackage (version 0.0.0.9000) was used as described[43]. FASTQ files from sequenced libraries were initially pooled by genotype. Paired-end reads were demultiplexed using fastq-multx from ea-utils (version 1.3.1). Reads were aligned and the number of UMIs extracted using p4cCreate4CseqTrack. A window of 1 kb around the viewpoint was removed from the analysis. 4C contact profiles from knockouts and controls were normalized for UMI coverage using the plotCompProf function and an adaptative smoothing method that controls window size so that no fewer than five molecules are included in each window. Assessment of differential contacts between knockout and wild-type 4C profiles in genomic regions of interest within a 0.5-megabase window surrounding the viewpoint was carried out using p4cIntervalsMean through a chi-squared test of normalized molecule counts.

## Statistics and reproducibility

No statistical method was used to predetermine sample size. No data were excluded from the analyses. The investigators were not blinded to allocation during the experiments and outcome assessment.

The results are shown as mean or median values, with error bars representing the s.e.m. or s.d., as stated in the figure captions. The numbers of biological replicates for each experiment are stated in the figure captions. $P$ values were calculated by $\chi^2$ test, two-sided Fisher's exact test, unpaired two-tailed Student's $t$-test or Wald or Wilcoxon rank-sum tests, as reported in the figure captions. The Brown–Forsythe test was used to test the equality of variances. For UMI-4C comparisons, $P$ values were calculated by $\chi^2$ test using umi4c. Statistical analysis of other epigenomic data is described in the appropriate Methods sections.

## Reporting summary

Further information on research design is available in the Nature Research Reporting Summary linked to this article.

## Data availability

Raw sequence reads from RNA-seq, small conditional RNA-seq and ChIP-seq, as well as ChIP-seq peaks, are available from ArrayExpress under accession codes E-MTAB-11463, E-MTAB-11471, E-MTAB-11472, E-MTAB-11473, E-MTAB-11474, E-MTAB-11475 and E-MTAB-11477. The following previously published data were re-analysed: mouse liver ChIP-seq for CTCF (GSE29184; ref. [77]), RAD21 (GSE102997; ref. [78]), FOXA2, CEBPB and HNF4A (GSE57559; ref. [38]), PPARA (GSE108689; ref. [79]), RXR (GSE35262; ref. [80]) and GATA4 (GSE49132; ref. [81]); mouse liver and kidney ATAC-seq (SRP167062; ref. [82]); and RNA-seq in humans (SRX218942; ref. [83]), chickens (SRX2704301 ref. [84]), *X. tropicalis* (SRX2704321; ref. [84]), zebrafish (E-MTAB-8959; ref. [85]), mouse kidneys (SRX2370375; ENCODE[86]), the small intestine (SRX2370402; ENCODE[86]); and human pluripotent stem cells differentiated into β cells (GSE140500; ref. [39]). All other data supporting the findings of this study are available from the corresponding author on reasonable request. Source data are provided with this paper.

## Code availability

All of the custom code used in this study is available upon reasonable request.

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

## Acknowledgements

This research was supported by the Medical Research Council (MR/ L02036X/1), Wellcome Trust (WT101033), European Research Council (Advanced Grant 789055), Spanish Ministry of Science and Innovation (BFU2014-54284-R and RTI2018-095666-B-I00) and National Institute for Health Research Imperial Biomedical Research Centre. Work in the Centre for Genomic Regulation was supported by the Centres de Recerca de Catalunya programme, Generalitat de Catalunya, Centro de Excelencia Severo Ochoa (CEX2020-001049) and by the Spanish Ministry of Science and Innovation aid to the European Molecular Biology Laboratory partnership. We thank the University of Barcelona School of Medicine animal facility, Center for Genomic Regulation and Imperial College London genomics units, Imperial College High Performance Computing Service, as well as K. Kaestner (University of Pennsylvania), M. Gannon (Vanderbilt University) and D. Tuveson (Cold Spring Harbor Laboratory) for the Cre lines. We thank J. Valcarcel and T. Graf (Center for Genomic Regulation) for comments on the manuscript, F. X. Real (Spanish National Cancer Research Centre) for insights and support, V. Grau and C. Roth for technical support and members of J.F.'s laboratory for valuable discussions.

## Author contributions

A. Beucher and J.F. conceived of, coordinated and supervised the study. A. Beucher performed the cell-based and computational studies and supervised the mouse analysis. A. Beucher, M.A.M. and J.G.-H. performed the analysis of mouse mutants. M.G.D.V. and P.R. designed and created the cell models. R.G.-F. performed the fluorescence in situ hybridization with supervision from A. Beucher. A. Beucher and I.M.-E. performed the UMI-4C studies. A. Beucher, A. Bernal and D.B. performed the stem cell studies. A. Beucher designed the mouse models with input from J.F., S.O. and P.V. S.O. and P.V. created the recombinant mice. H.H. participated in the single-cell genomics. A. Beucher and J.F. wrote the manuscript with input from the remaining authors.

## Competing interests

P.R. is a founder and consultant for EndoCells/UniverCell Biosolutions. All of the remaining authors declare no competing interests.

## Additional information

**Extended data** is available for this paper at https://doi.org/10.1038/ s41556-022-00996-8.

**Correspondence and requests for materials** should be addressed to Anthony Beucher or Jorge Ferrer.

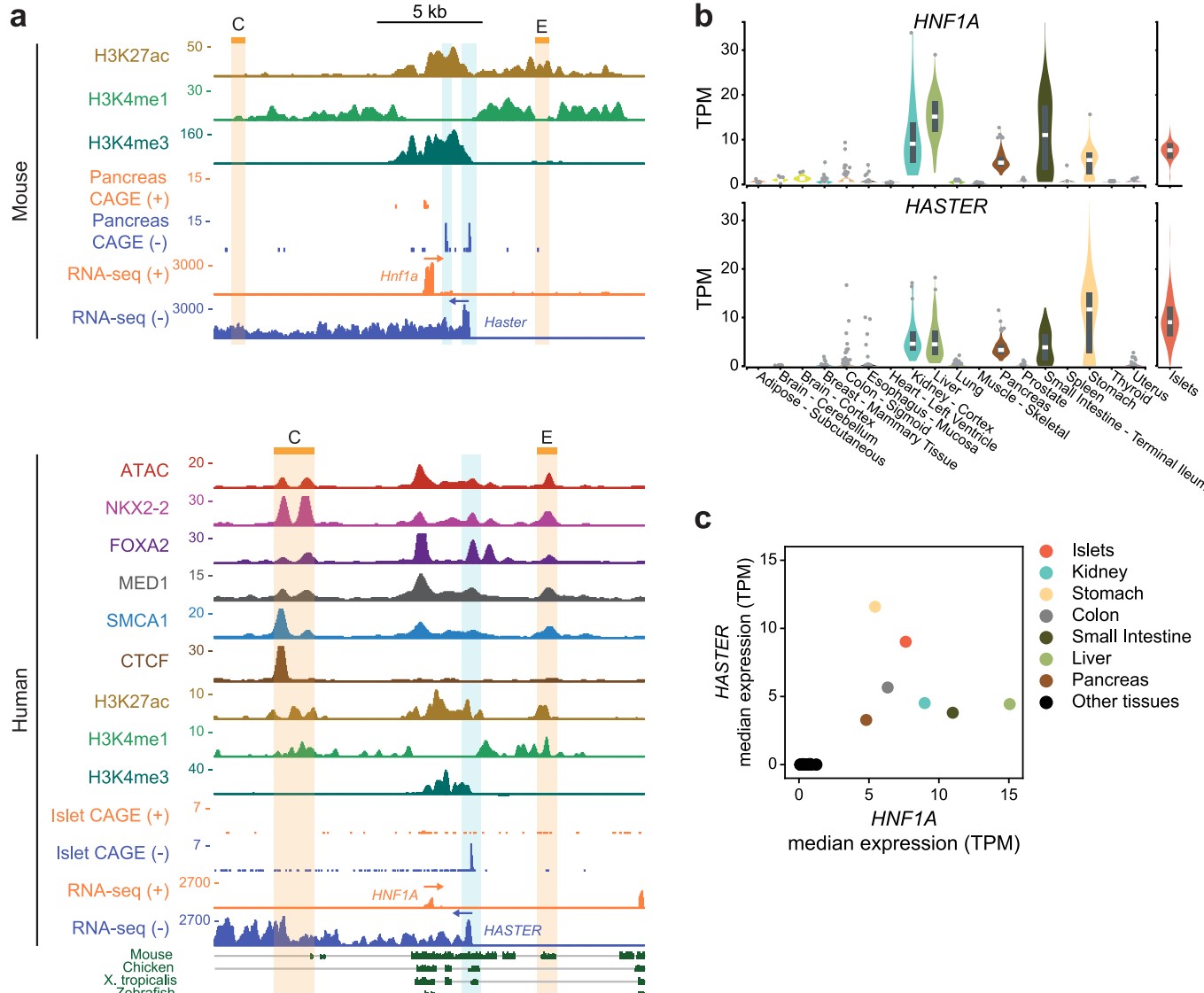

**Extended Data Fig. 1 | *HASTER* and *HNF1A* regulatory landscapes. a**, Chromatin and RNA maps in mouse (top) and human islets (bottom). Human CAGE is from islets, and mouse CAGE is from pancreas. The two mouse *Haster* transcriptional start sites are highlighted in blue, although only one transcriptional origin is apparent in human islets. The E islet enhancer, and CTCF-bound C region, both of which are bound by islet transcription factors, are highlighted in beige. **b**, *HNF1A* and *HASTER* expression across GTEx human tissues (Data Source: GTEx Analysis Release V8) and human islets (n = 130). Boxes show median and interquartile ranges. **c**, *HASTER* and *HNF1A* median transcript levels across tissues are negatively correlated, with the exception of whole pancreas.

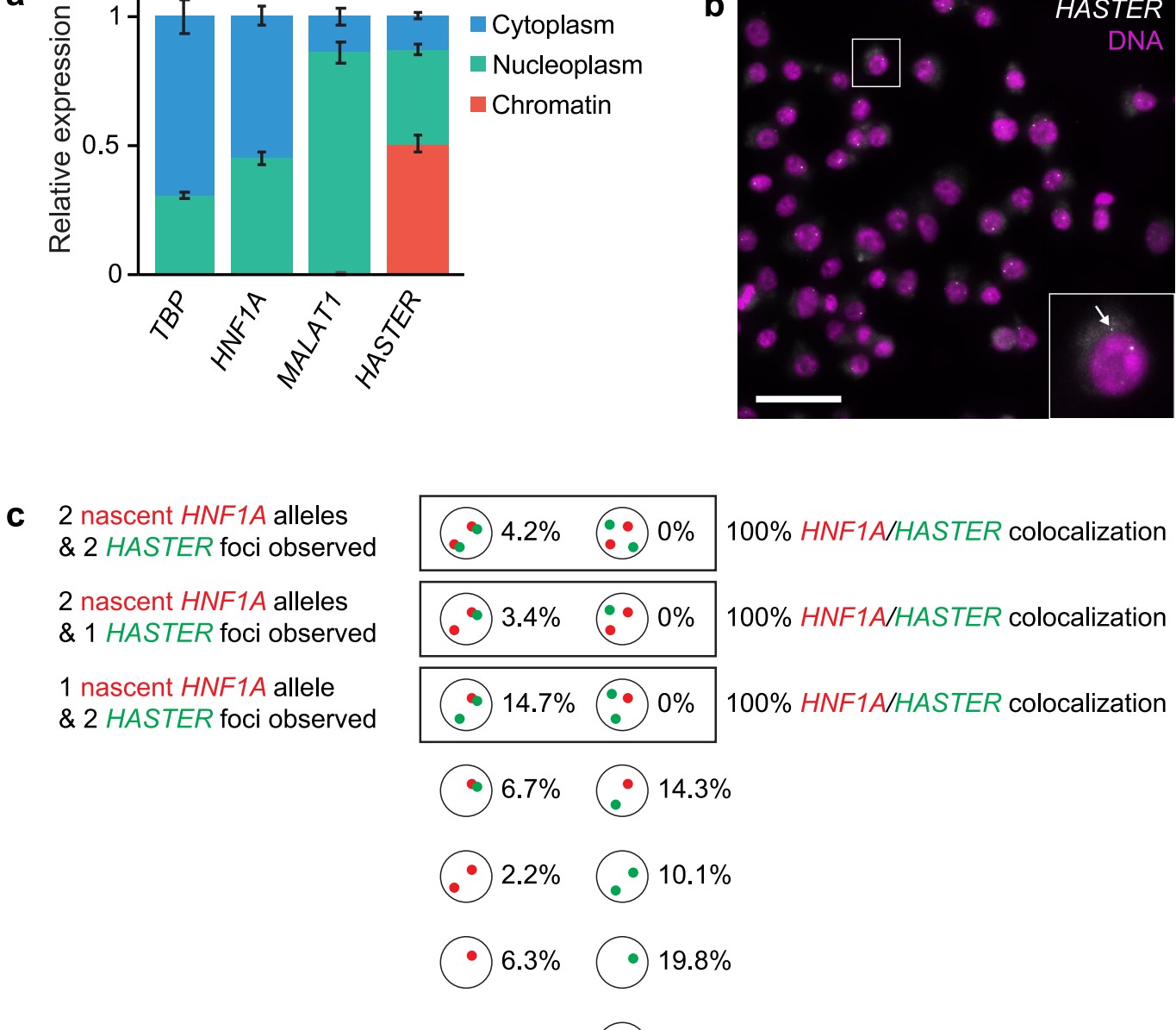

**Extended Data Fig. 2 | HASTER transcripts localize to the nucleus. a**, Relative subcellular expression of HASTER lncRNA in EndoC-βH3 cells, compared to control mRNAs (*TBP* and *HNF1A*) and the nuclear lncRNA *MALAT1*. Mean ± s.d., n = 3 biological replicates. **b**, Single molecule fluorescence in situ hybridization signals for HASTER. HASTER transcripts are almost exclusively observed in the nucleus (deconvoluted images). The inset shows a rare non-nuclear signal. (n = 5 independent experiments). **c**, Colocalization of single molecule fluorescent in situ hybridization signals for HASTER (exonic probes) and *HNF1A* (intronic probes for *HNF1A*) in human EndoC-βH3 cells. n = 496 cells. The degree to which HASTER and intronic *HNF1A* RNA molecules are located at the *HNF1A* locus can only be assessed when two *HNF1A* or two HASTER molecules are seen in the same nucleus. In all such instances *HNF1A* and HASTER were found to colocalize. Scale bar, 20 μm.

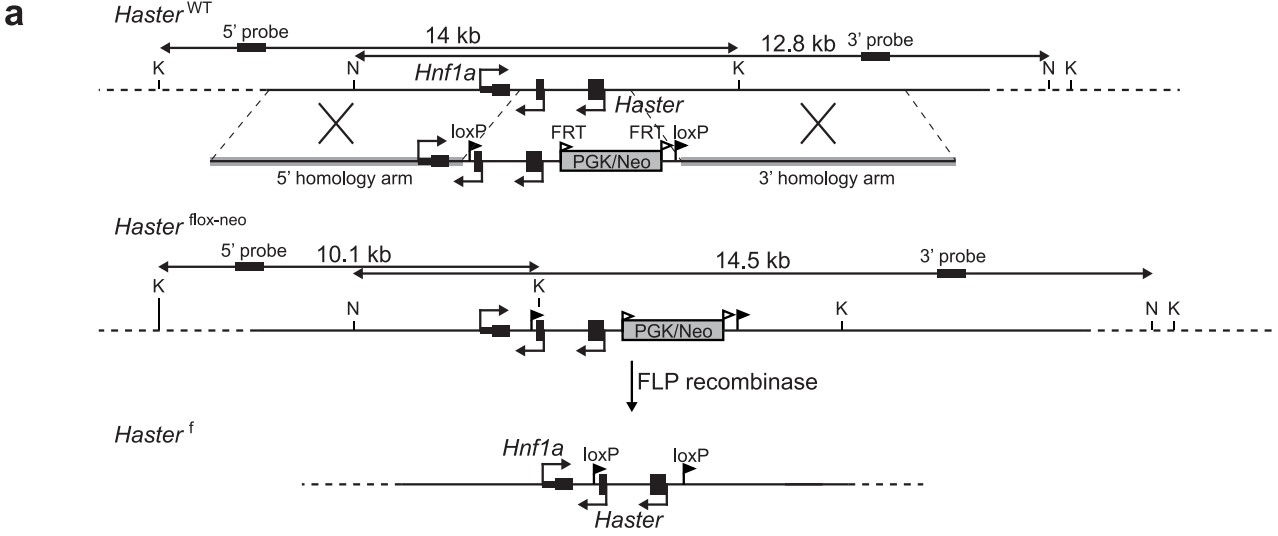

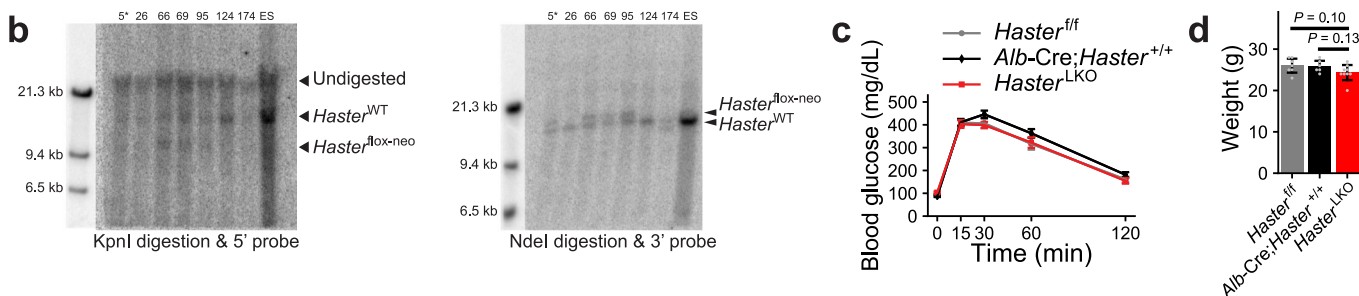

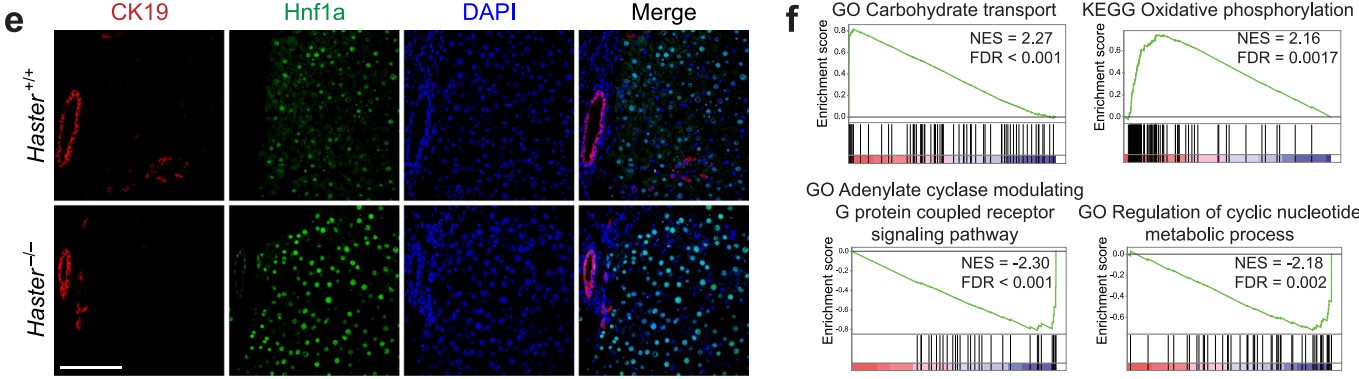

**Extended Data Fig. 3 | Conditional *Haster* allele and phenotypic analysis in liver. a**, Schematic of the targeted allele, digestion fragments and probes used for Southern blot analysis of different alleles. **b**, Southern blot with KpnI (left) and NdeI (right) digestion. Asterisk, Clone 5 was selected to establish the line. n = 1 Southern blot. K, KpnI; N, NdeI; ES, parental embryonic stem cell (C57BL/6). **c**, Intraperitoneal glucose tolerance test in *Haster*^LKO and control 8-week-old mice.

Mean ± s.e.m. **d**, Body weight at 8 weeks for *Haster*^LKO and controls. Mean ± s.d. **c**,**d**, n = 9 *Haster*^LKO, n = 6 *Alb*-Cre;*Haster*^+/+ and n = 7 *Haster*^f/f, two-tailed Student's t-test. **e**, Immunofluorescence showing HNF1A overexpression in *Haster*^-/- liver. n = 1 per genotype. Scale bar, 100 μM. **f**, GSEA displaying the enrichment of functional annotations in *Haster*^LKO upregulated (top panel) and downregulated (bottom panel) genes.

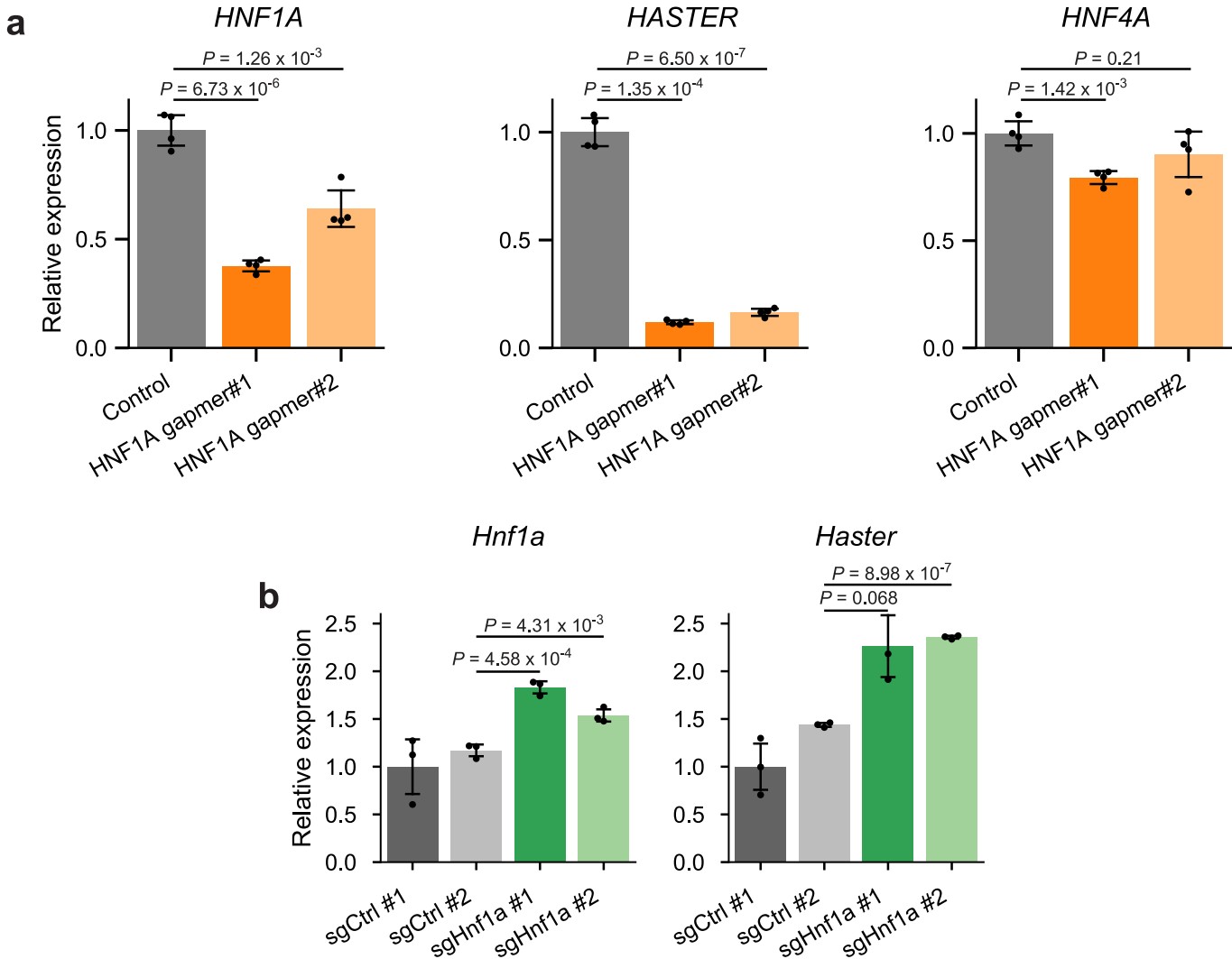

**Extended Data Fig. 4 | *HASTER* is sensitive to decreased or increased *HNF1A* expression. a**, Locked nucleic acid (LNA) GapmeR knockdown of *HNF1A* (#1, *HNF1A* exon 8; #2, *HNF1A* exon 1) in human EndoC-βH3 β cells led to decreased *HASTER* RNA, and minor changes in other HNF1A-dependent genes. n = 3 nucleofections, *TBP*-normalized mean ± s.d.; two-tailed Student's t-test. **b**, CRISPR-SAM activation of *Hnf1a* in mouse MIN6 β cells. n = 3 lentiviral transductions, representative of 2 independent experiments. *Tbp*-normalized mean ± s.d.; two-tailed Student's t-test.

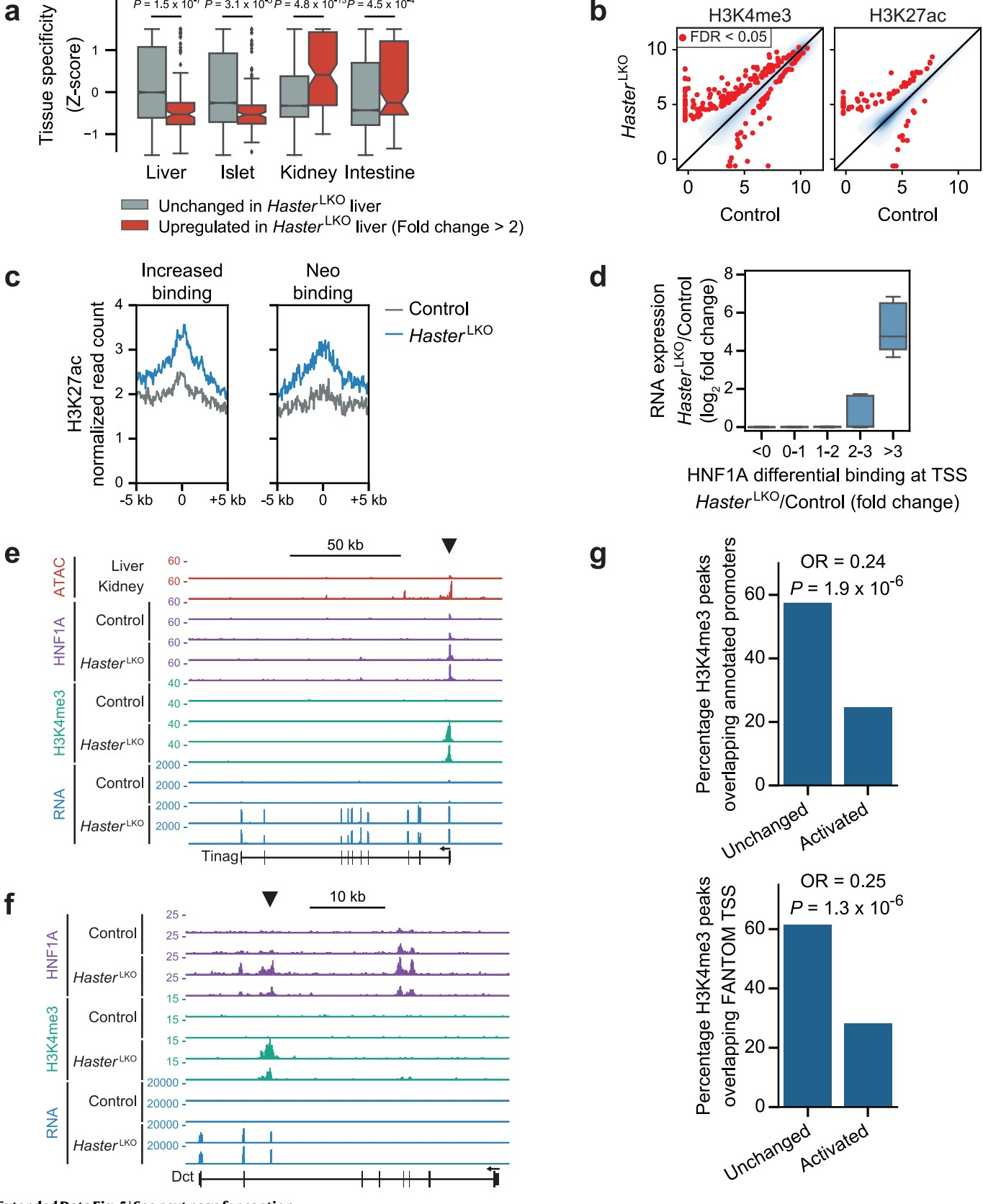

Extended Data Fig. 5 | See next page for caption.

**Extended Data Fig. 5 | *Hnf1a* upregulation perturbs HNF1A binding selectivity. a**, Tissue specificity of gene expression across HNF1A-expressing tissues for genes upregulated in *Haster*[LKO] liver. To quantify tissue specificity, for each gene and tissue we calculated a Z-score that represents the deviation of expression in that tissue relative to the average from all tissues. n = 3 kidney samples, n = 5 liver and small intestine samples and n = 6 pancreatic islet samples. Box plots show medians and interquartile ranges; whiskers, 1.5 times the interquartile ranges. Two-sided Wilcoxon rank-sum *P*-values. **b**, Liver H3K4me3 and H3K27ac in *Haster*[LKO] and control liver (log$_2$ normalized ChIP-seq read count; n = 3 mice per genotype). Red, differential H3K4me3 or H3K27ac sites (FDR ≤ 0.05); blue, kernel density of differential H3K4me3 or H3K27ac sites with FDR > 0.05. **c**, H3K27ac at HNF1A-bound regions in *Haster*[LKO] and controls (average of n = 3 mice per genotype). **d**, RNA fold change in *Haster*[LKO] vs. control liver of HNF1A-bound promoters for the different categories of HNF1A binding in *Haster*[LKO] liver. n = 5 mice per genotype for RNA and n = 3 mice per genotype for HNF1A ChIP. Box plots show medians and interquartile ranges; whiskers, 1.5 times the interquartile ranges. **e**, Examples of HNF1A neo-binding sites that lead to ectopic promoter and gene activation in *Haster*[LKO] liver. **f**, Ectopic activation of an intragenic promoter in *Haster*[LKO] liver (n = 2 mice per genotype). y axes in **e** and **f** represent MACS2 *P* values for ChIP-seq and RPKM for RNA-seq. **g**, *Activated* genomic regions that are bound by HNF1A and become active promoters in *Haster*[LKO], but are inactive in control liver, overlap less frequently with annotated promoter and FANTOM5 CAGE transcriptional start sites, compared with *unchanged* HNF1A-bound active promoters in control liver, suggesting that some may be aberrant promoters rather that repurposed from other cell types. Two-sided Fisher's exact test odd ratio (OR) and *P*-values, n = 3 mice per genotype.

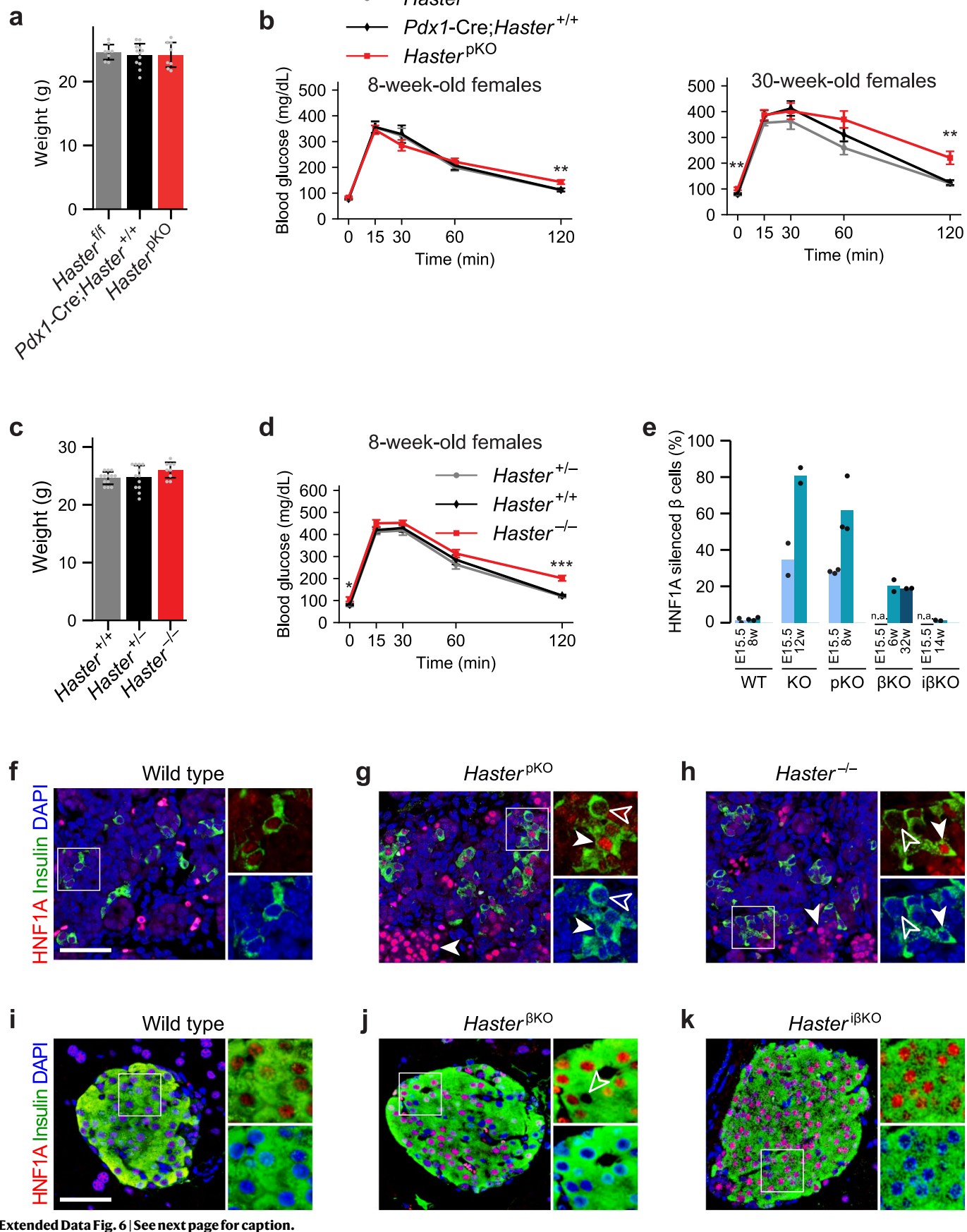

**Extended Data Fig. 6 | See next page for caption.**

**Extended Data Fig. 6 | Characterization of *Haster* mutants. a**, Body weight of males at 8 weeks of age. *Haster*^pKO (n = 8), *Pdx1*-Cre;*Haster*^+/+ (n = 12) and *Haster*^f/f (n = 8). Mean ± s.d. **b**, Intraperitoneal glucose tolerance test in 8-week-old and 30-week-old female *Haster*^pKO (n = 9), *Pdx1*-Cre;*Haster*^+/+ (n = 10) and *Haster*^f/f (n = 10). Mean ± s.e.m., *$P \leq 0.05$, **$P \leq 0.01$ and ***$P \leq 0.001$ (two-tailed Student's t-test). **c**, Body weight of 8- to 10-week-old male *Haster*^-/- (n = 9), *Haster*^+/- (n = 12) and *Haster*^-/- (n = 13). Mean ± s.d. **d**, Intraperitoneal glucose tolerance test in 8- to 10-week-old females *Haster*^-/- (n = 10), *Haster*^+/- (n = 10) and *Haster*^+/+ (n = 12). Mean ± s.e.m., *$P \leq 0.05$, **$P \leq 0.01$ and ***$P \leq 0.001$ (two-tailed Student's t-test). **e**, Relative quantification of HNF1A-negative β cells at the indicated age and genotype. Results show that HNF1A silencing correlates with time of *Haster* knockout, with higher silencing frequency after early deletion (*Haster* germline KO and *Haster*^pKO models). HNF1A silencing increased with time in β cells from germline KO and *Haster*^pKO models, but not when excision occurred in early β cells (βKO). No HNF1A silenced β cells were observed after *Pdx1*-CreER^TM-based tamoxifen-inducible excision in adult β cells (*Haster*^iβKO model). Mean ± s.d from 2-4 sections per mouse, 2-4 mice per condition. **f-h**, Immunofluorescence for HNF1A and insulin in E15.5 (**f**) *Haster*^+/+, (**g**) *Haster*^pKO, and (**h**) *Haster*^-/- pancreas. Solid arrowheads, insulin cells overexpressing HNF1A; hollow arrowheads, insulin cells lacking HNF1A. **i-k**, Immunofluorescence for HNF1A and insulin in adult (**i**) wild-type, (**j**) *Haster*^βKO and (**k**) *Haster*^iβKO pancreas. Arrows point to HNF1A-negative β cells. Most β cells from *Haster*^βKO and *Haster*^iβKO islets overexpress HNF1A. **e-k**, n = 2 wild type embryos and n = 3 adult wild type mice; n = 2 *Haster*^-/- embryos and adult mice; n = 3 *Haster*^pKO embryos and adult mice; n = 2 *Haster*^βKO 6- and 32-week-old mice; n = 2 *Haster*^iβKO mice. n.a.: not analyzed because deletions were performed at a later time point. Scale bar, 50 μM.

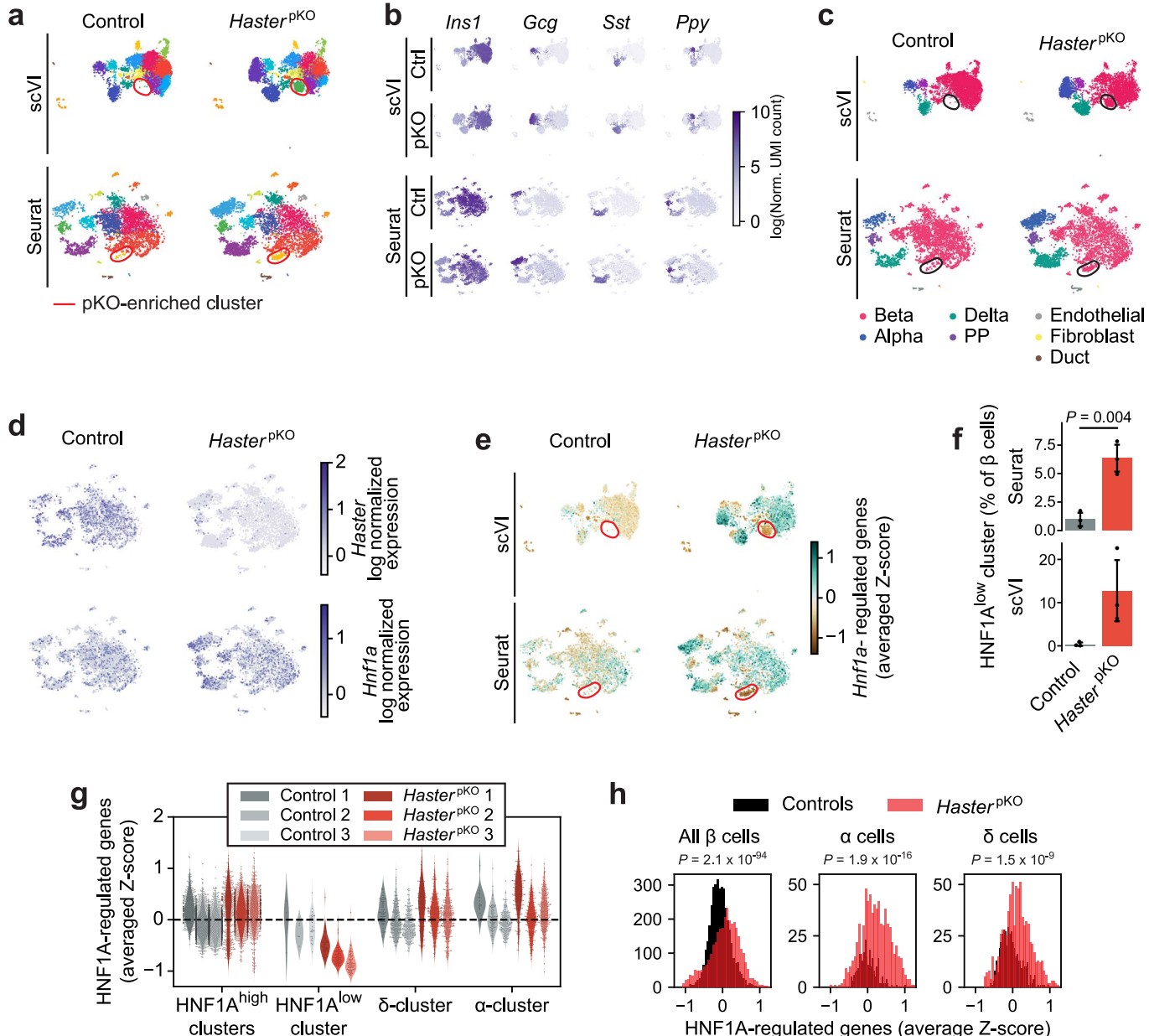

**Extended Data Fig. 7 | Single-cell RNA-seq of *Haster*<sup>pKO</sup> islets. a**, Clusters of islet cells from female *Haster*<sup>pKO</sup> (4961 cells from triplicates for Seurat; 4456 for scVI) and controls (4646 cells from triplicates for Seurat; 4460 for scVI) determined by Seurat (t-SNE projection) or scVI (UMAP projection). **b**, scVI UMAP and Seurat t-SNE projections showing hormone expression. **c**, Cell type assignment based on marker gene expression. **d**, *Haster* and *Hnf1a* mRNA (log normalized UMI count) in different cellular populations of control and *Haster*<sup>pKO</sup> islets (Seurat). **e**, HNF1A-regulated gene expression (average Z-score) showing lower expression of HNF1A-regulated genes in the *HASTER*<sup>pKO</sup>-enriched β cell cluster (HNF1A<sup>low</sup>) and high expression of HNF1A-regulated genes in other β cells in *HASTER*<sup>pKO</sup> cells. **f**, Relative proportions of β cells present in the *HASTER*<sup>pKO</sup>-enriched HNF1A<sup>low</sup>

β cell cluster. n = 3 mice per genotype. Mean ± s.d., two-tailed Student's t-test. We note these proportions are lower than observed in situ, plausibly because *Hnf1a*<sup>-/-</sup> islets have a marked propensity to dissociate upon collagenase digestion, which is expected to cause negative selection of HNF1A-deficient cells after digestion and FACS sorting of single cells. **g**, HNF1A-regulated gene expression (average Z-score) for different cell types in individual samples (Seurat). **h**, Histograms showing the distribution of HNF1A-regulated gene expression (average Z-score) for β, α and δ cells (Seurat). Bins = 40. The variance of HNF1A-regulated gene expression increased in *Haster*<sup>pKO</sup> β, α and δ cells, showing that HNF1A-regulated genes are either upregulated or downregulated in islet cells. Levene test *P*-values.

**a**

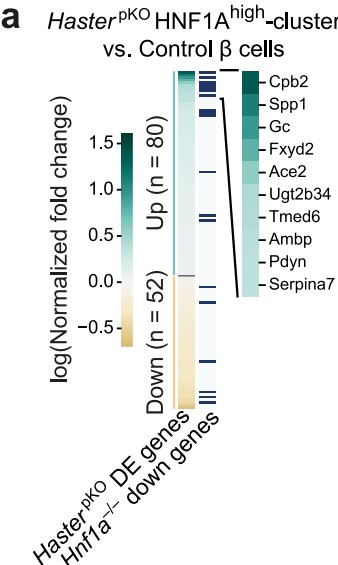

**b**

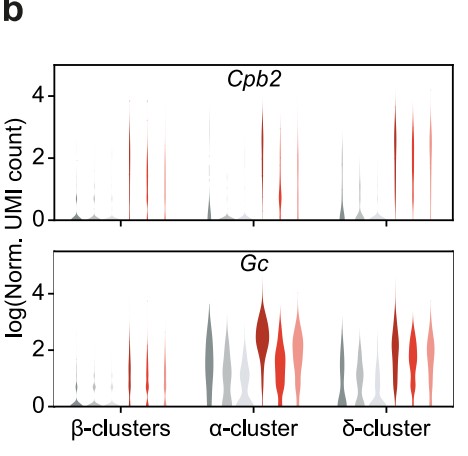

**c**

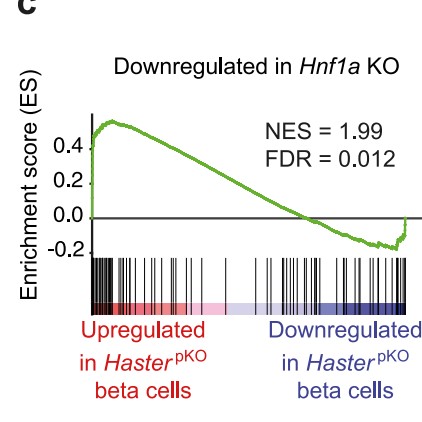

**d**

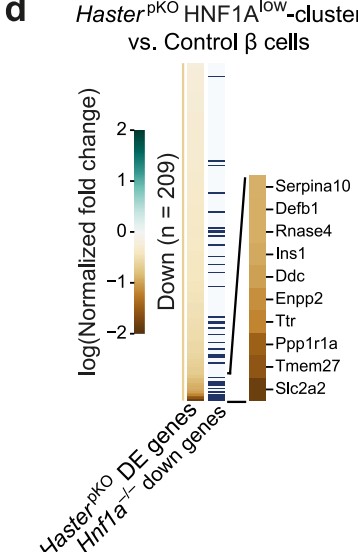

**e**

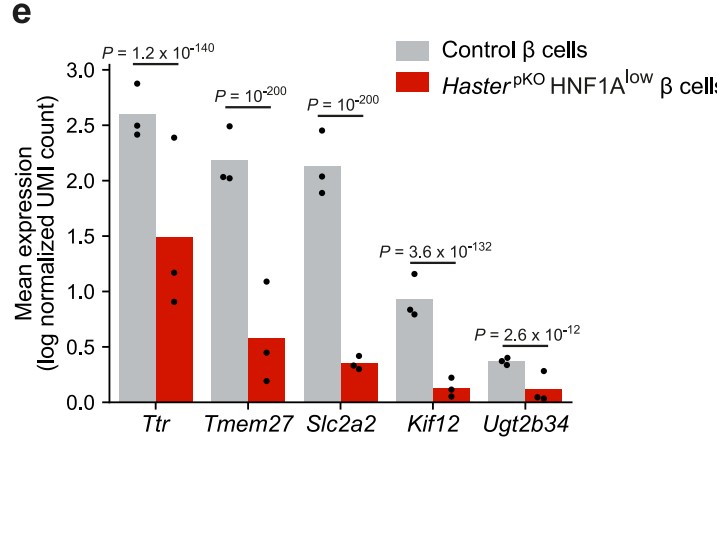

**Extended Data Fig. 8 | Differential gene expression in *Haster*^pKO β cells. a**, Genes differentially expressed in the major β-cell cluster of *Haster*^pKO islets. Many of the most upregulated genes in *Haster*^pKO islets are downregulated in *Hnf1a*^-/- islets (blue horizontal lines). **b**, Examples of two genes that are known to be downregulated in *Hnf1a*^-/- islets, *Cpb2* and *Gc*, and show increased expression in *Haster*^pKO β cells. **c**, GSEA showing upregulation in *Haster*^pKO β cells of genes downregulated in *Hnf1a* KO islets. **d**, Genes that are downregulated (combined *P* ≤ 0.05) in *Haster*^pKO HNF1A^low cells are often downregulated in *Hnf1a*^-/- islets (blue horizontal lines). **e**, Expression of selected genes that are known to be downregulated in *Hnf1a* KO islets and are downregulated in *Haster*^pKO HNF1A^low cells. Dots are medians of samples (log normalized UMI count) and bars are means of 3 replicates. Two-sided Wilcoxon Rank Sum test, *P*-values for the different biological replicates combined with Fisher's method.

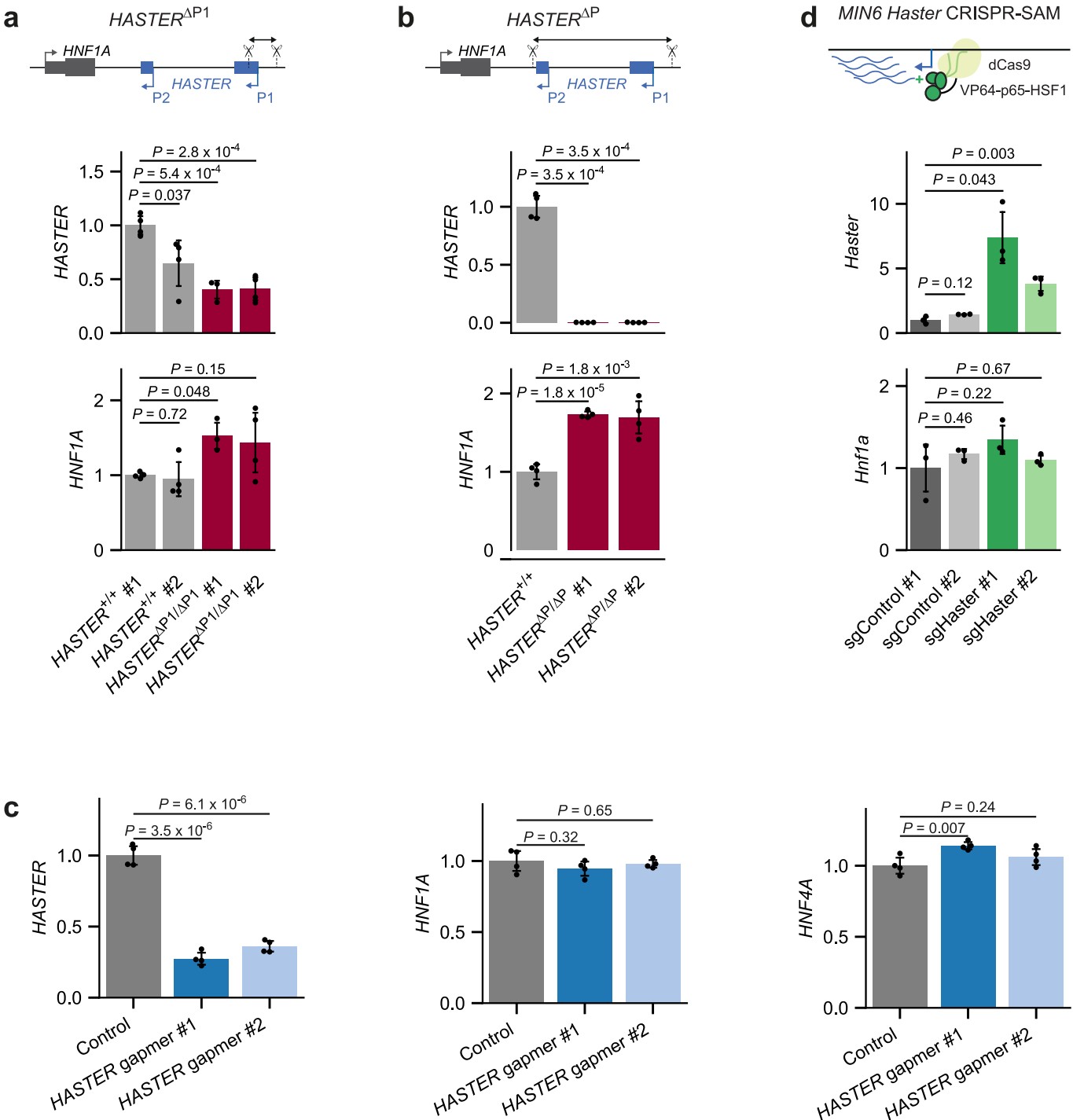

**Extended Data Fig. 9 | *HASTER* perturbations in β cells. a,b**, *HASTER* and *HNF1A* RNA in EndoC-βH3 cells with clonal homozygous deletion of (**a**) *HASTER* P1 promoter (*HASTER*ΔP1/ΔP1) or (**b**) *HASTER* P1 and P2 promoters (*HASTER*ΔP/ΔP). Deletion #1 and #2 were generated with independent pairs of sgRNAs, *HASTER*+/+ clones were transfected with sgRNAs targeting the AAVS1 locus. n = 4 clones per deletion. **c**, Two sets of LNA oligonucleotides (GapmeRs) were used to elicit *HASTER* degradation in EndoC-βH3 cells, without significant changes in *HNF1A* or *HNF4A* mRNA. n = 3 nucleofections. **a-c**, Expression normalized by *TBP*. Mean ± s.d., two-tailed Student's t-test. **d**, *Haster* activation by CRISPR-SAM in MIN6 mouse β cells had no effect on *Hnf1a* expression. n = 3 lentiviral transductions. Expression normalized by *Tbp*. Mean ± s.d., two-tailed Student's t-test relative to the control #1 sgRNA.

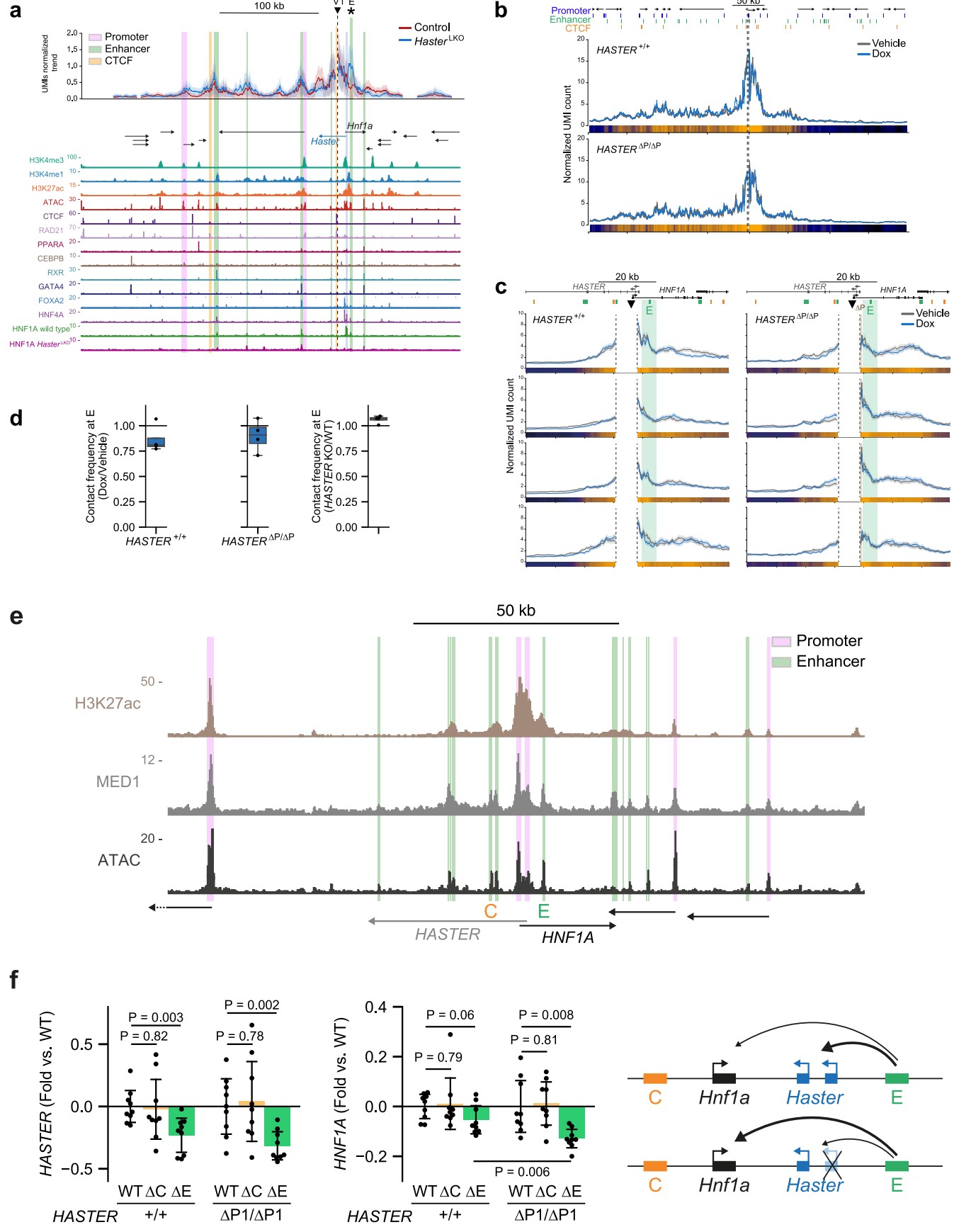

**Extended Data Fig. 10 | See next page for caption.**

**Extended Data Fig. 10 | HNF1A binding to *HASTER* promoter reduces *HNF1A* promoter – enhancer interactions. a**, Top, UMI-4C profile trends using a viewpoint region upstream of Hnf1a (V1), near a CTCF-bound C site, in adult liver from n = 3 wild type (blue) or mutant (red) mice. Bottom, chromatin features and transcription factor binding in adult mouse liver. The *Hnf1a* upstream region contacts several enhancers, promoters and CTCF/cohesin sites in control and *Haster*^LKO liver. The interaction between *Hnf1a* upstream region and E (asterisk) is increased in *Haster*^LKO liver (see also Fig. 7). The region deleted in *Haster*^LKO mice is highlighted in blue. **b**, UMI-4C profile trends of doxycycline-induced HNF1A overexpression in *HASTER*^+/+ and *HASTER*^ΔP/ΔP EndoC-βH3 cells with *HNF1A* promoter as viewpoint. Strongest contacts occurred within < 20 kb 3′ of *HNF1A* promoter, while weaker contacts were predominantly observed in a ~400 kb region 5′ of *HNF1A* promoter. Top tracks, genes and regulatory elements in human pancreatic islets (Miguel-Escalada, et al., 2019). Pool of libraries for n = 4 independent experiments. **c**, Individual UMI-4C profile trends from four individual experiments of doxycycline-induced HNF1A overexpression in *HASTER*^+/+ and *HASTER*^ΔP/ΔP EndoC-βH3 cells. In **a-c** shades represent estimated binomial standard deviation centered on the profile trend. **d**, *HNF1A* promoter – E interaction frequencies from individual replicates (n = 4 independent experiments). Interaction frequencies were measured at a 5 kb region centered on E highlighted with a green shade. Box plots show medians and interquartile ranges; whiskers, 1.5 times the interquartile ranges. **e**, Human islet chromatin marks showing the position of enhancers in the vicinity of *HNF1A*. **f**, *HASTER*^+/+ or *HASTER*^ΔP1/ΔP1 clone #1 cells carrying targeted deletions in C (ΔC), E (ΔE) or sgGFP as control (WT). *HASTER*^+/+ control and E deletion are identical to Fig. 7e. ΔC and ΔE were polyclonal deletions. Results are expressed as fold-differences relative to the parental *HASTER*^+/+ or *HASTER*^ΔP1/ΔP1 cells. This showed that ΔC has no effect on *HASTER* or *HNF1A*, ΔE had significant effects on *HASTER* but did not significantly affect *HNF1A* in wild type cells, yet showed a significant *HNF1A* reduction in *HASTER*^ΔP1/ΔP1 cells. This is shown in cartoon form in the right panel, whereby E predominantly enhances *HASTER* transcription, but enhances *HNF1A* in the absence of *HASTER*. Pool of n = 3 independent experiments with 3 pairs of sgRNAs for each deletion. *TBP*-normalized mean expression ± s.d.; two-tailed Student's t-test.

# nature research

# Reporting Summary

Nature Research wishes to improve the reproducibility of the work that we publish. This form provides structure for consistency and transparency in reporting. For further information on Nature Research policies, see our Editorial Policies and the Editorial Policy Checklist.

## Statistics

For all statistical analyses, confirm that the following items are present in the figure legend, table legend, main text, or Methods section.

| n/a | Confirmed | |
|---|---|---|
| ☐ | ☒ | The exact sample size (*n*) for each experimental group/condition, given as a discrete number and unit of measurement |
| ☐ | ☒ | A statement on whether measurements were taken from distinct samples or whether the same sample was measured repeatedly |
| ☐ | ☒ | The statistical test(s) used AND whether they are one- or two-sided *Only common tests should be described solely by name; describe more complex techniques in the Methods section.* |
| ☒ | ☐ | A description of all covariates tested |
| ☐ | ☒ | A description of any assumptions or corrections, such as tests of normality and adjustment for multiple comparisons |
| ☐ | ☒ | A full description of the statistical parameters including central tendency (e.g. means) or other basic estimates (e.g. regression coefficient) AND variation (e.g. standard deviation) or associated estimates of uncertainty (e.g. confidence intervals) |
| ☐ | ☒ | For null hypothesis testing, the test statistic (e.g. *F*, *t*, *r*) with confidence intervals, effect sizes, degrees of freedom and *P* value noted *Give P values as exact values whenever suitable.* |
| ☒ | ☐ | For Bayesian analysis, information on the choice of priors and Markov chain Monte Carlo settings |
| ☒ | ☐ | For hierarchical and complex designs, identification of the appropriate level for tests and full reporting of outcomes |
| ☒ | ☐ | Estimates of effect sizes (e.g. Cohen's *d*, Pearson's *r*), indicating how they were calculated |

*Our web collection on statistics for biologists contains articles on many of the points above.*

## Software and code

Policy information about availability of computer code

| Data collection | sra-tools (v.2.9.1_1)<br>loomR (v.0.2.0.1)<br>loompy (v.2.0.16) |
|---|---|
| Data analysis | bedGraphToBigWig (v4)<br>Bowtie2 (v.2.3.5)<br>Samtools (v.1.7)<br>BedTools (v.2.27.1)<br>MACS2 (v.2.1.1)<br>DiffBind (v.2.8.0)<br>deepTools (v.3.0.2)<br>pybedtools (v.0.8.0)<br>Homer (v.3.12)<br>Picard (v.2.6.0)<br>GSEAPreranked (v.6.0, GenePattern)<br>CellRanger (v.3.0.2)<br>Seurat (v.3.0.1)<br>scVI-tools (v.0.11.0)<br>python (v.3.7.3)<br>numpy (v.1.15.4)<br>pandas (v.0.25.0)<br>matplotlib (v.3.1.0)<br>seaborn (v.0.9.0)<br>scipy (v.1.1.0) |

STAR (v.2.3.0)
Salmon (v.0.11)
r-base (v.3.6.1)
DESeq2 (v.1.24.0)
StringTie (v.2.0)
gffcompare (v.0.10.1)
umi4cPackage (v 0.0.0.9000)
ea-utils (v.1.3.1)

For manuscripts utilizing custom algorithms or software that are central to the research but not yet described in published literature, software must be made available to editors and reviewers. We strongly encourage code deposition in a community repository (e.g. GitHub). See the Nature Research guidelines for submitting code & software for further information.

## Data

Policy information about availability of data

All manuscripts must include a data availability statement. This statement should provide the following information, where applicable:
- Accession codes, unique identifiers, or web links for publicly available datasets
- A list of figures that have associated raw data
- A description of any restrictions on data availability

Raw sequence reads from RNA-seq, scRNA-seq, UMI-4C and ChIP-seq are available from Arrayexpress, under accession number (E-MTAB-11463, E-MTAB-11471, E-MTAB-11472, E-MTAB-11473, E-MTAB-11474, E-MTAB-11475 and E-MTAB-11477).
Mouse liver ChIP-seq for CTCF was from GSE29184; RAD21 from GSE102997; FOXA2, CEBPB and HNF4A from GSE57559; PPARA from GSE108689; RXR from GSE35262; GATA4 from GSE49132. Mouse ATAC-seq was from SRP167062.
Liver RNA-seq for human is from SRX218942, chicken from SRX2704301, X. tropicalis from SRX2704321 and zebrafish from E-MTAB-8959.
RNA-seq of hPSC differentiated to β cells were obtained from GSE140500.
Mouse kidney (SRX2370375) and small intestine (SRX2370402) RNA-seq reads were obtained from the Mouse ENCODE project
GTex data was downloaded from https://gtexportal.org/home/
GRCm38 Bowtie2 index was downloaded from https://genome-idx.s3.amazonaws.com/bt/mm10.zip
GRCh37 human genome was downloaded from GENCODE https://www.gencodegenes.org/human/release_19.html
GRCm38 mouse genome was downloaded from https://www.gencodegenes.org/mouse/release_M10.html
GRCz11 (dnaRer11) zebrafish genome was downloaded from https://www.ncbi.nlm.nih.gov/assembly/GCF_000002035.6/
galGal5 chicken genome and XenTro9 Xenopus genome were downloaded from ENSEMBL (www.ensembl.org)
Processed data files are provided as Supplementary Data Sets and deposited at https://www.crg.eu/en/programmes-groups/ferrer-lab#datasets (pending at submission).

# Field-specific reporting

Please select the one below that is the best fit for your research. If you are not sure, read the appropriate sections before making your selection.

☒ Life sciences    ☐ Behavioural & social sciences    ☐ Ecological, evolutionary & environmental sciences

For a reference copy of the document with all sections, see nature.com/documents/nr-reporting-summary-flat.pdf

# Life sciences study design

All studies must disclose on these points even when the disclosure is negative.

| | |
|---|---|
| Sample size | For in vivo experiments, sample sizes were defined based on studies with similar experimental design and expected outcome. For glucose tolerance curves the numbers used are expected to yield statistically significant differences for >10% glucose tolerance effects of single gene defects bred on an inbred background. For in vitro studies, we ensured that any result is not only robust to technical replication of the read-out assay, but also to different attempts to perform the genetic modification. For CRISPR deletions, CRISPRi road block and CRISPR activation on cell lines, all experiments were performed using at least 2 different sgRNAs, or 2 different pair of sgRNA for deletions. Knock-down experiments were performed with 2 different LNA Gapmers. Likewise we used at least 2 independent mutant and control clones for hPSC mutations. |
| Data exclusions | No data were excluded. |
| Replication | In addition to using independent modifications as outlined above, each independent modification was studies with replicate experiments, as detailed for each result. Immunofluorescence stainings were performed on tissues from 2 or 3 mice for each genotype and age. Western blot and RNA-smFISH experiments were all replicated 2 to 3 times. All attempt of replication reported were successful. |
| Randomization | Not relevant for this study. Samples were allocated to a group based on their genotypes. |
| Blinding | While no proper blinding were performed, no particular attention was given to the genotypes during samples collection or measurements. |

# Reporting for specific materials, systems and methods

We require information from authors about some types of materials, experimental systems and methods used in many studies. Here, indicate whether each material, system or method listed is relevant to your study. If you are not sure if a list item applies to your research, read the appropriate section before selecting a response.

## Materials & experimental systems

| n/a | Involved in the study |
|-----|-----------------------|
| ☐ | ☒ Antibodies |
| ☐ | ☒ Eukaryotic cell lines |
| ☒ | ☐ Palaeontology and archaeology |
| ☐ | ☒ Animals and other organisms |
| ☒ | ☐ Human research participants |
| ☒ | ☐ Clinical data |
| ☒ | ☐ Dual use research of concern |

## Methods

| n/a | Involved in the study |
|-----|-----------------------|
| ☐ | ☒ ChIP-seq |
| ☒ | ☐ Flow cytometry |
| ☒ | ☐ MRI-based neuroimaging |

# Antibodies

| | |
|---|---|
| Antibodies used | Rabbit anti-β-Tubulin (#2146, Cell Signaling Technology)<br>Rabbit anti-HNF1A (D7Z2Q, #89670, Cell Signaling Technology)<br>Guinea pig anti-insulin (A0564, Dako, discontinued)<br>Guinea pig anti-glucagon (polyclonal, 4030-01F, Millipore, discontinued)<br>Rat anti-Cytokeratin 19 (Hybridoma Bank, TROMA III-c)<br>Goat anti-PDX1 (AF2419, R&D Systems)<br>Mouse anti-Nkx6.1 (F55A10, Hybridoma Bank)<br>Rabbit anti-H3K27ac (Abcam, ab4729)<br>Rabbit monoclonal anti-H3K4me3 (Merk, 05-745R, clone 15-10C-E4)<br>Rabbit anti-H3K4me1 (Abcam, ab8895)<br>Donkey anti rabbit Alexa Fluor 488 (711-545-152, Jackson Immunoresearch, 1/800)<br>Donkey anti rabbit Alexa Cy3 (711-166-152, Jackson Immunoresearch, 1/400)<br>Donkey anti guinea pig Alexa Fluor 488 (706-545-148, Jackson Immunoresearch, 1/800)<br>Donkey anti guinea pig Cy5 (706-175-148, Jackson Immunoresearch, 1/400)<br>Donkey anti goat Alexa Fluor 488 (705-545-147, Jackson Immunoresearch, 1/800)<br>Donkey anti rat Cy3 (712-165-153, Jackson Immunoresearch, 1/400)<br>Donkey anti mouse Cy5 (715-175-151, Jackson Immunoresearch, 1/400)<br>Goat Anti-Rabbit IgG H&L (HRP) (ab97051, Abcam, 1/2000) |
| Validation | The anti-β-Tubulin has been tested for Western blot by the provider using various cell lines extracts. This antibody is widely used (676 citations).<br>The anti-HNF1A was tested on section from acinar-specific Hnf1a KO mice (Kalisz et al, EMBO J, 2020)<br>The anti-insulin, anti-glucagon and anti-Cytokeratin 19 antibodies are widely used in the field (http://www.informatics.jax.org/antibody/key/1494).<br>The anti-PDX1 was validated by by Western blot and immunohistochemistry by manufacturer (https://www.rndsystems.com/products/human-pdx-1-ipf1-antibody_af2419)<br>Anti-Nkx6.1 (Generation and characterization of monoclonal antibodies against the transcription factor Nkx6.1, Jorgensen MC, The journal of histochemistry and cytochemistry : official journal of the Histochemistry Society 54.5 (2006 May): 567-74.<br>The anti-H3K4me3 was validated by dotblot for the ENCODE project (https://genome.ucsc.edu/ENCODE/validation/antibodies/human_H3K27ac_validation_Bernstein.pdf)<br>The anti-H3K4me3 was validated by ChIP-seq by the manufacturer "The highest 25% of peaks identified in the 05-745R and 07-473 datasets showed 99% overlap with peaks identified in the ENCODE H3K4me3 BROAD Histone track for HeLa S3." https://www.merckmillipore.com/GB/en/product/Anti-trimethyl-Histone-H3-Lys4-Antibody-clone-15-10C-E4-rabbit-monoclonal,MM_NF-05-745R?ReferrerURL=https%3A%2F%2Fwww.google.com%2F&bd=1<br>The anti-H3K4me1 was was validated by dotblot for the ENCODE project (https://genome.ucsc.edu/ENCODE/validation/antibodies/mouse_H3K4me1_ab8895_validation_Ren.pdf) |

# Eukaryotic cell lines

Policy information about cell lines

| | |
|---|---|
| Cell line source(s) | EndoC-βH3 cells were generated and provided by Philippe Ravassard, a co-author of this study (Benazra et al., Mol Metab., 2015).<br>MIN6 cells (Miyazaki, et al., Endocrinology, 1990) were provided by Rosa Gasa (IDIBAPS, Barcelona, Spain)<br>H9 cells were provided by WiCell.<br>293FT were purchased from from Invitrogen. |
| Authentication | Cell lines were authenticated by morphological inspection and qPCR measurement of cell-type-specific markers. |
| Mycoplasma contamination | EndoC-βH3 and H9 cells were tested and were negative for Mycoplasma. MIN6 and 293FT were not tested. |
| Commonly misidentified lines<br>(See ICLAC register) | None of the commonly misidentified lines were used. |

## Animals and other organisms

Policy information about studies involving animals; ARRIVE guidelines recommended for reporting animal research

| | |
|---|---|
| Laboratory animals | When not specified otherwise, 8- to 12-weeks old males were used for sample collection and glucose stimulated insulin secretion. The Haster Stop and Haster floxed alleles were generated for this study in C57BL/6 background. The Haster null allele was obtained by breeding the Haster floxed allele with the Tg(EIIa-cre) mice. Cell type specific deletions were performed using the Alb-Cre (Tg(Alb1-cre)1Khk), Pdx1-Cre (Tg(Pdx1-Cre)6Tuv), Ins1-Cre (Ins1tm1.1(cre)Thor) and Pdx1CreERTM (Tg(Pdx1-Cre/Esr1*)1Mga). For allele specific experiments, Haster +/Stop or Haster +/- mice were bred with PWK/PhJ mice and the F1 was analyzed. |
| Wild animals | This study did not involve wild animals. |
| Field-collected samples | This study did not involve samples collected from the field. |
| Ethics oversight | Animal experimentation was carried out in compliance with the EU Directive 86/609/EEC and Recommendation 2007/526/EC regarding the protection of animals used for experimental and other scientific purposes, enacted under Spanish law 1201/2005, and all experiments were approved by the Institutional Animal Care Committees of the University of Barcelona and Parc de Recerca Biomedica de Barcelona. |

Note that full information on the approval of the study protocol must also be provided in the manuscript.

## ChIP-seq

### Data deposition

☒ Confirm that both raw and final processed data have been deposited in a public database such as GEO.

☒ Confirm that you have deposited or provided access to graph files (e.g. BED files) for the called peaks.

| | |
|---|---|
| Data access links<br>*May remain private before publication.* | Processed data files are provided as Supplementary Data Sets and deposited at https://www.crg.eu/en/programmes-groups/ferrer-lab#datasets (pending at submission). |
| Files in database submission | Access from Arrayexpress pending at the time of submission |
| Genome browser session<br>(e.g. UCSC) | no longer applicable |

### Methodology

| | |
|---|---|
| Replicates | ChIP-seq were performed on 3 KO and 3 control mice for each antibody used. Between 84% to 98% of peaks from the replicate with the lowest number of peaks called were present in the other replicates. |
| Sequencing depth | Sample   Antibody   Read length   Total number of reads   Uniquely aligned reads (after removing duplicates)<br>Haster_LKO_1   HNF1A   50 SE   36,577,365   26,191,763<br>Haster_LKO_2   HNF1A   50 SE   36,412,249   26,168,607<br>Haster_LKO_3   HNF1A   50 SE   34,102,415   24,137,402<br>Control_1   HNF1A   50 SE   29,209,244   20,679,931<br>Control_2   HNF1A   50 SE   40,716,830   29,132,320<br>Control_3   HNF1A   50 SE   30,681,737   17,823,581<br>Haster_LKO_1   H3K4me3   50 SE   40,778,667   25,266,443<br>Haster_LKO_2   H3K4me3   50 SE   40,134,454   24,487,328<br>Haster_LKO_3   H3K4me3   50 SE   41,345,041   24,344,704<br>Control_1   H3K4me3   50 SE   23,870,270   16,172,657<br>Control_2   H3K4me3   50 SE   49,820,846   29,578,339<br>Control_3   H3K4me3   50 SE   30,669,001   17,572,781<br>Haster_LKO_1   H3K27ac   50 SE   40,318,283   33,806,790<br>Haster_LKO_2   H3K27ac   50 SE   51,441,794   42,494,143<br>Haster_LKO_3   H3K27ac   50 SE   48,839,781   40,112,808<br>Control_1   H3K27ac   50 SE   29,620,302   24,993,138<br>Control_2   H3K27ac   50 SE   51,638,898   42,321,953<br>Control_3   H3K27ac   50 SE   30,567,140   26,055,207 |
| Antibodies | Rabbit anti-HNF1A (D7Z2Q, Cell Signaling Technology)<br>Rabbit anti-H3K27ac antibody (Abcam, ab4729)<br>Rabbit monoclonal anti-H3K4me3 (Merk, 05-745R, clone 15-10C-E4) |
| Peak calling parameters | BOWTIE_INDEX=bowtie2-index/mm10/mm10<br>BLACKLISTED_REGIONS=mm10.blacklist.bed<br>#Aligning reads with Bowtie2<br>bowtie2 -p 16 --no-unal -x $BOWTIE_INDEX -U $FASTQ_FILE_PATH/$FASTQ_FILE -S $OUTPUT_DIR/${sample_name}_aligned.sam<br><br>#Converting sam file to bam format, filter for multimappings<br>samtools view -q 30 -bh -o $OUTPUT_DIR/${sample_name}_aligned.bam $OUTPUT_DIR/${sample_name}_aligned.sam |

```
echo "Sort bam"
samtools sort -@ 16 -T $TMPDIR/${sample_name}_aligned.tmp.bam -o $OUTPUT_DIR/${sample_name}_sorted.bam $OUTPUT_DIR/
${sample_name}_aligned.bam

# Remove duplicates
picard MarkDuplicates \
    I=$OUTPUT_DIR/${sample_name}_sorted.bam \
    O=$OUTPUT_DIR/${sample_name}_nodup.bam \
    M=$OUTPUT_DIR/${sample_name}_nodup.log \
    REMOVE_DUPLICATES=true

#Filter out blacklisted regions
intersectBed -abam  $OUTPUT_DIR/${sample_name}_nodup.bam -b $BLACKLISTED_REGIONS -v > $OUTPUT_DIR/
${sample_name}_clean.bam

# Indexing bam file
samtools index $OUTPUT_DIR/${sample_name}_clean.bam

#Peak calling for transcription factor ChIP
macs2 callpeak -t $OUTPUT_DIR/${sample_name}_clean.bam -c $INPUT_CHIP -f BAM -g mm -n ${MACS2_OUTPUT} --keep-dup all -B
-q 0.05 --outdir $OUTPUT_DIR

#Peak calling for transcription factor ChIP
macs2 callpeak -t $OUTPUT_DIR/${sample_name}_clean.bam -c $INPUT_CHIP -f BAM -g mm -n ${MACS2_OUTPUT} --keep-dup all -B
-q 0.05 --broad --outdir $OUTPUT_DIR
```

Data quality

Peaks called at in 2 replicates were merged and used for differential binding and subsequent analysis.

Sample   Antibody   Percentage of peaks called at 5% FDR with >5-fold enrichment
Haster_LKO_1   HNF1A   51.7%
Haster_LKO_2   HNF1A   54.7%
Haster_LKO_3   HNF1A   60.6%
Control_1   HNF1A   69.4%
Control_2   HNF1A   48.9%
Control_3   HNF1A   75.6%
Haster_LKO_1   H3K4me3   46.8%
Haster_LKO_2   H3K4me3   50.4%
Haster_LKO_3   H3K4me3   52.2%
Control_1   H3K4me3   54.4%
Control_2   H3K4me3   46.6%
Control_3   H3K4me3   58.7%

Software

Reads were aligned using Bowtie2 (v.2.3.5)
Duplicates were removed using Picard (v.2.6.0)
Peaks were called using MACS2 (v.2.1.1)
Intersections were performed using BedTools (v.2.27.1)
Differential binding sites were determined using DiffBind (v.2.8.0)
Read coverages were computed using deepTools (v.3.0.2)
Motif analysis was performed using Homer (v.3.12)

