## [Peer Review File · Nature Cell Biology]

Peer Review Information

Journal: Nature Cell Biology

Manuscript Title: HASTER lncRNA promoter is a cis-acting transcriptional stabilizer of HNF1A

Corresponding author name(s): Jorge Ferrer

Reviewer Comments & Decisions:

Decision Letter, initial version:
--

*Please delete the link to your author homepage if you wish to forward this email to co-authors.

Dear Professor Ferrer,

Your manuscript, "HASTER is a transcriptional stabilizer of HNF1A", has now been seen by 3 referees, who are experts in lncRNA (referees 1 and 2) and diabetes and beta cells (referee 3). As you will see from their comments (attached below) they find this work of potential interest, but have raised substantial concerns, which in our view would need to be addressed with considerable revisions before we can consider publication in Nature Cell Biology.

Nature Cell Biology editors discuss the referee reports in detail within the editorial team, including the chief editor, to identify key referee points that should be addressed with priority, and requests that are overruled as being beyond the scope of the current study. To guide the scope of the revisions, I have listed these points below. I should stress that the referees' concerns point to a premature dataset and these points would need to be addressed with experiments and data, and reconsideration of the study for this journal and re-engagement of referees would depend on strength of these revisions.

In particular, it would be essential to:

a) explore the mechanism underlying the opposite effects of HASTER on HNF1A transcription in different cell types, along with the experimental suggestions raised by referee 2:

Weaknesses include (1) lack of underlying molecular mechanism to explain the variegated expression pattern of HNF1A in a homozygous HASTER knockout pancreatic endocrine context; (2) the enhancer hijacking mechanism proposed by the authors reasonably explains the inhibitory effect of HASTER promoter on HNF1A transcription. However, what's the mechanism contributing to HASTER activation function to HNF1A transcription shown in pancreatic endocrine context? More intriguingly, what determines which cell sub-population in pancreatic endocrine cells that will exhibit silencing and overexpression of HNF1A in a homozygous HASTER knockout context? (3) in the proposed model

wherein the authors claim that at low HNF1A concentrations, there are unhindered interactions between the HNF1A promoter and E (or other local enhancers); while at high HNF1A concentrations, HNF1A binding to HASTER limits contacts between E and HNF1A, thereby decreasing HNF1A transcription. It implicates that HNF1A binding to HASTER promoter might hijack enhancer E to prevent contact between HNF1A promoter and enhancer E. However, there is no data to show that HNF1A can regulate interaction between these loci.

5. Can the authors also perform UMI-4C in control and HASTER KO β cells? It is of relevance to the biologic phenotype. In addition, as mentioned above, to validate the model, it's worth doing UMI-4C after overexpression of HNF1A to check whether decreased contact between enhancer E and HNF1A promoter is observed.

6. What's the underlying mechanism by which HASTER promoter is able to hijack enhancer E? A plausible explanation is that high HNF1A binding affinity at HASTER promoter favors HASTER promoter-enhancer E interaction. Can the authors test it by performing UMI-4C experiment in control cells and cells where HNF1A binding motifs at HASTER promoter shown in Fig. 1f are edited or deleted?

b) modify the text to clarify that it is the noncoding locus, not the lncRNA product, that functions, as noted by referees 1 and 2, and be careful about the nomenclature of the noncoding locus, as noted by referee 3.

c) All other referee concerns pertaining to strengthening existing data, providing controls, methodological details, clarifications and textual changes, should also be addressed.

d) Finally please pay close attention to our guidelines on statistical and methodological reporting (listed below) as failure to do so may delay the reconsideration of the revised manuscript. In particular please provide:

We would be happy to consider a revised manuscript that would satisfactorily address these points, unless a similar paper is published elsewhere, or is accepted for publication in Nature Cell Biology in the meantime.

- ensure that it conforms to our format instructions and publication policies (see below and <https://www.nature.com/nature/for-authors>).

- provide a point-by-point rebuttal to the full referee reports verbatim, as provided at the end of this

2letter.

- provide the completed Reporting Summary (found here <https://www.nature.com/documents/nr-reporting-summary.pdf>). This is essential for reconsideration of the manuscript will be available to editors and referees in the event of peer review. For more information see <http://www.nature.com/authors/policies/availability.html> or contact me.

When submitting the revised version of your manuscript, please pay close attention to our [Digital Image Integrity Guidelines](https://www.nature.com/nature-research/editorial-policies/image-integrity). and to the following points below:

Nature Cell Biology is committed to improving transparency in authorship. As part of our efforts in this direction, we are now requesting that all authors identified as 'corresponding author' on published papers create and link their Open Researcher and Contributor Identifier (ORCID) with their account on the Manuscript Tracking System (MTS), prior to acceptance. ORCID helps the scientific community achieve unambiguous attribution of all scholarly contributions. You can create and link your ORCID from the home page of the MTS by clicking on 'Modify my Springer Nature account'. For more information please visit www.springernature.com/orcid.

This journal strongly supports public availability of data. Please place the data used in your paper into a public data repository, or alternatively, present the data as Supplementary Information. If data can only be shared on request, please explain why in your Data Availability Statement, and also in the correspondence with your editor. Please note that for some data types, deposition in a public repository is mandatory - more information on our data deposition policies and available repositories appears below.

[REDACTED]

We would like to receive a revised submission within six months.

3We hope that you will find our referees' comments, and editorial guidance helpful. Please do not hesitate to contact me if there is anything you would like to discuss.

Best wishes,

Jie Wang

Jie Wang, PhD
Senior Editor
Nature Cell Biology

Tel: +44 (0) 207 843 4924
email: jie.wang@nature.com

Reviewers' Comments:

Reviewer #1:

Remarks to the Author:

In "HASTER is a transcriptional stabilizer of HNF1A" the authors investigate the role of HASTER, a long non-coding RNA antisense to the promoter and first exon of HNF1A, which is a gene whose haploinsufficiency has been shown to cause diabetes mellitus. By employing both in vitro and in vivo model systems, they cleverly uncover both positive and negative feedback loops between HASTER and HNF1A, which ultimately stabilize HNF1A expression levels and maintain a proper 3D genome architecture during the early phases of liver development. They also demonstrate that the HASTER promoter per se is the main or only determinant of this activity, whereas the lncRNA molecule seems dispensable. Moreover, disruption of a ~320bp region comprising Haster TSS was shown to be sufficient to cause phenotypes comparable to diabetes mellitus, pinpointing the importance of considering non-coding loci when investigating the genetics of this disease.

This research work represents a thorough investigation of the relationship between a disease-causing gene (HNF1A) and a non-coding locus (HASTER), which results in a "balancing" interaction, which has not been reported before, as far as I know. This story will be of great interest to a broad field and therefore certainly fit the scope of Nature Cell Biology. Therefore, I have only minor comments which I propose to be addressed:

1) The graphical representations of the HASTER-HNF1A locus are sometimes confusing. For instance, while in Fig. 1B and 1F it is clear those two transcriptional units are antisense and overlapping, in others one may misunderstand and consider them as convergent and overlapping (e.g. Fig. 1A), convergent (Fig. 2A and C, Fig. 5C, D and F, Fig. 6B) or as completely separated units (Fig. 7D and F). I acknowledge that a graphical representation of two antisense and overlapping loci is conceptually challenging, but I also think an effort here can help to make sure that the readers properly interpret all the deep and careful experiments which have been done.

2) Overall, all the experiments in the manuscript converge on the promoter of HASTER being important, and as far as I can see, there is no evidence that the transcription or the RNA product are

4important. Therefore, the authors need to be careful about referring to any “lncRNA function” throughout the manuscript. Specifically, they should not explicitly say that the disruption of a lncRNA can cause diabetes mellitus (e.g. final sentence in the abstract). While it is true as they beautifully demonstrate that a deletion of HASTER promoter region cause such phenotypes, it is also true that this happens within the 5’ region of Hnf1a and the RNA product was found to be dispensable. Thus, currently there is no definitive proof that the disease-causing agent is Haster itself, and the authors do not mention human subjects with diabetes mellitus and genetic deletions in this locus.

3) Do the mice with the poly(A) insetion in HASTER (HasterSTOP) have any phenotype? Or were they only bred with HNF1A mice, and so the authors didn’t have a chance to observe homozygous mice? The authors mention that HNF1A levels are not affected, but do these mice have diabetes or any other observed phenotype? This is not essential, as the manuscript already has very extensive results, but can be of interest if the information is available or can be obtained rapidly.

4) What happens to HASTER expression if the authors over-express the HNF1A cDNA? Currently a CRISPRa experiment is presented (Fig. S3b), but its more difficult to interpret since if HASTER promoter is in physical proximity to the HNF1A promoter CRISPRa can “leak” from one promoter to another. I do not consider it an essential experiment, but if it can be performed, it can strengthen the results.

Reviewer #2:

Remarks to the Author:

Summary

Eukaryotic genome are transcribed extensively and many lncRNAs have been shown to act as local regulators to influence the expression of nearby genes. However, it’s challenging to dissect whether the regulatory function is through the lncRNA transcripts themselves or results from cis regulatory elements at lncRNA locus. In this manuscript submitted by Beucher et al., the authors focused on a specific lncRNA HASTER to study its potential biological function. Using mouse and human models, the authors showed that: (1) HNF1A is a positive regulator of HASTER; (2) HASTER is able to regulate HNF1A transcription through positive and negative feedback loops; (3) HASTER inactivation, which shows variegated HNF1A overexpression or silencing, leads to impaired insulin secretion and diabetes; (4) HASTER promoter but not HASTER transcript or transcription act, regulates HNF1A transcription; (5) HASTER promoter regulates HNF1A by remodelling a specific enhancer-promoter interaction in the 3D chromatin context.

From their results, the authors conclude that they discover a cis-regulatory element that is unlike enhancers or silencers, and instead stabilizes expression levels of a pioneer transcription factor linked to diabetes phenotype.

Review

This paper addresses an interesting topic linking an lncRNA promoter to diabetes disease. A strength of the work is using the mouse model to show the biological relevance of HASTER promoter in diabetes progression. Weaknesses include (1) lack of underlying molecular mechanism to explain the variegated expression pattern of HNF1A in a homozygous HASTER knockout pancreatic endocrine context; (2) the enhancer hijacking mechanism proposed by the authors reasonably explains the inhibitory effect of HASTER promoter on HNF1A transcription. However, what’s the mechanism contributing to HASTER activation function to HNF1A transcription shown in pancreatic endocrine

5context? More intriguingly, what determines which cell sub-population in pancreatic endocrine cells that will exhibit silencing and overexpression of HNF1A in a homozygous HASTER knockout context? (3) in the proposed model wherein the authors claim that at low HNF1A concentrations, there are unhindered interactions between the HNF1A promoter and E (or other local enhancers); while at high HNF1A concentrations, HNF1A binding to HASTER limits contacts between E and HNF1A, thereby decreasing HNF1A transcription. It implicates that HNF1A binding to HASTER promoter might hijack enhancer E to prevent contact between HNF1A promoter and enhancer E. However, there is no data to show that HNF1A can regulate interaction between these loci.

Additional comments:

1. The conclusion drawn in the abstract that “They also show that disruption of a mammalian lncRNA can cause diabetes mellitus.” is overstated. Throughout the manuscript, the authors aim to claim that HASTER promoter itself, but not HASTER transcript regulates HNF1A transcription that leads to diabetes phenotype.
2. For Supple Fig. 3a, Locked nucleic acid (LNA) gapmer ASO not only triggers RNase-H- mediated transcript cleavage but also induces premature transcription termination (See Jong-Sun Lee et al., Antisense-Mediated Transcript Knockdown Triggers Premature Transcription Termination Mol Cell 2020). Hence, the result shown in Supple Fig. 3a may be from LNA- induced premature transcription termination at HNF1A locus that also disrupts HASTER transcription. LNA sequences used for HNF1A mRNA knockdown should be listed. Western blot data examining HNF1A knockdown efficiency should be provided.
3. For Supple Fig. 3b, as HNF1A promoter is close to HASTER promoter, it’s expected to see that CRISPR-SAM strategy (which may activate both two promoters) leads to increased HASTER RNA expression. dCas9 ChIP experiment should be performed to prove that CRISPRa strategy used to target HNF1A promoter specifically targets HNF1A promoter but not HASTER promoter.
4. For scRNA-seq data, a table summarizing all the different samples that were processed for sequencing and the number of high quality nuclei sequenced should be added. Furthermore, age, sex and genetic background of the animals from which the sequencing samples were derived should be disclosed in the method section. The authors conclude that HASTER deletion in vivo caused functional HNF1A deficiency in pancreatic β cells resulting in diabetes. Have the authors tried different parameters or other clustering strategy to identify this molecularly distinct cell subtype? Why is this cell subtype not shown in control samples? It seems that the scRNA-seq data can’t explain the diabetes phenotype observed in HASTER knockout mice, as the major cluster of β cells didn’t show deficient expression of HNF1A-regulated genes (instead shows increased HNF1A and its regulated genes). The authors should discuss the scRNA-seq data (including other cell clusters shown in supple Fig. 9) in more detail in the text or discussion part.
5. Can the authors also perform UMI-4C in control and HASTER KO β cells? It is of relevance to the biologic phenotype. In addition, as mentioned above, to validate the model, it’s worth doing UMI-4C after overexpression of HNF1A to check whether decreased contact between enhancer E and HNF1A promoter is observed.
6. What’s the underlying mechanism by which HASTER promoter is able to hijack enhancer E? A plausible explanation is that high HNF1A binding affinity at HASTER promoter favors HASTER promoter-enhancer E interaction. Can the authors test it by performing UMI-4C experiment in control cells and cells where HNF1A binding motifs at HASTER promoter shown in Fig. 1f are edited or deleted?
7. Can the authors use a large window to analyze UMI-4C data to check other potential enhancers interacting with HNF1A promoter? As in WT cells, HASTER knockout doesn’t change HNF1A mRNA expression, implying other enhancers may be involved to promote HNF1A transcription in the presence of HASTER.

8. Is the "Zebrafish" labeled at the bottom of Fig. 1a and Supple Fig. 1b correct?
9. Some typographical errors should be corrected. For example, at Line 264, "Hnf1a knock our mice" should be "Hnf1a knock out mice"; at Line 328, " the inhibitory effects of HNF1A overexpression where almost completely suppressed" should be "the inhibitory effects of HNF1A overexpression were almost completely suppressed"; In Supple Fig. 12, "showing that reducing Haster elongation in failed to increase Hnf1a expression from the same C57BL/6 allele." should be "showing that reducing Haster elongation in liver failed to increase Hnf1a expression from the same C57BL/6 allele."

Reviewer #3:
Remarks to the Author:

Beucher et al., Nat Cell Biol

The study by Beucher represents an experimental tour de force to decipher the role of the HNF1A-AS1 lncRNA that is transcribed from a promoter in the first intron of diabetes-relevant gene HNF1A. The authors develop multiple new models, both in gene-targeted mice and in cell lines and apply a whole range of advanced techniques to their analysis.

It is clear from all these models that the first intron of HNF1A is important to the regulation of its expression. The findings reported are sometimes puzzling, as the authors observe both positive and negative consequences of decreasing HNF1A-AS1 transcripts, even within the same model. Fascinating but not truly explained is the observation that in some pancreatic beta cells with the same altered genetic makeup, HNF1A expression is increased while in others it is decreased. Yes, the authors conclusion that their manipulation affected the stability of HNF1A gene regulation is true, but it is not clear how. Finally, it appears in the end that the HNF1A-AS1 transcript does not matter, but the DNA element that encodes it does, suggestions that this acts more like a critical enhancer.

Abstract:

- Please remove the term "pioneer" transcription factor, as this was not formally proven for HNF1alpha. DNA binding transcription factor, or lineage-determining transcription factor is more accurate.

Introduction:

Remove the end of the introduction, where you summarize the results. While I realize that this is now common practice, it is superfluous. A single sentence stating what is being investigated here suffices.

Nomenclature: NCBI has no entry for HASTER, but multiple for HNF1A-AS1. In fact, over 20 publications in PubMed refer to the locus as HNF1A-AS1. Renaming a gene that is already annotated is poor form, and only adds to confusion in the field. Please change all references to HASTER, both in text and figures, to HNF1A-AS1

7Results:

There have been more than 20 studies on HNF1A-AS1, implicating this transcript in cancer, with multiple mechanisms that differ from the current study proposed. May be the published work is all wrong, but it needs to be discussed.

Figure 1:

Figure 1A: The HNF1A-AS1 transcripts are curious. While annotated in NCBI as a spliced gene with multiple exons, these tracks only show a single un-spliced transcript (while for HNF1A only the annotated exons have reads. How do the authors interpret this discrepancy?

In addition, for each species, the + and - strand reads need to be shown on the same scale, so that the reader can appreciate the relative abundances of the two transcripts.

Figure 1B: this is confusing. There is only one spot where transcripts for HNF1A and HNF1A-AS1 overlap, but multiple that show only HNF1A-AS1. The co-localization needs to be quantified, showing a single dot is not convincing.

Figure 1C-F, interpretation. The data clearly show that there is an HNF1A dependent cis-regulatory element that impacts HNF1A-AS1 transcription. It is also possible, however, that the element with 7 HNF1alpha binding sites also serves as an intronic enhancer controlling HNF1A expression itself. What happens to HNF1A transcripts in HNF1A null tissues?

Figure 2. Very strong data in this figure. Unfortunately, as is true with all lncRNAs/antisense transcripts near transcription factor genes, the interpretation of the data is very difficult. The data are also compatible with the alternative explanation that the DNA sequences deleted, which are contained within intron 1 of HNF1A, are controlling HNF1A expression through directly changing the binding of transcription factors or altering chromatin structure. These possibilities need to be discussed. The effect on HNF1A transcript, while statistically significant, is marginal. The effects on protein levels shown in 2f appear stronger. The Western blots needs to be quantified, and should there be a larger effect size, this divergence needs to be explained.

Figure 3: Very nice data showing that increased HNF1A levels lead to occupancy at non-canonical sites in the liver.

Figure 4 and 5: Good data, and strong models. However, the data could also be interpreted as deletion of DNA sequences in the HNF1A intron 1 alters TF binding, chromatin structure etc independent of HNF1A-AS1 promoter activity.

REFERENCES – are limited to a total of 70 for Articles, Resources, Technical Reports; and 40 for Letters. This includes references in the main text and Methods combined. References must be numbered sequentially as they appear in the main text, tables and figure legends and Methods and must follow the precise style of Nature Cell Biology references. References only cited in the Methods should be numbered consecutively following the last reference cited in the main text. References only

associated with Supplementary Information (e.g. in supplementary legends) do not count toward the total reference limit and do not need to be cited in numerical continuity with references in the main text. Only published papers can be cited, and each publication cited should be included in the numbered reference list, which should include the manuscript titles. Footnotes are not permitted.

Methods should be written concisely, but should contain all elements necessary to allow interpretation and replication of the results. As a guideline, Methods sections typically do not exceed 3,000 words. The Methods should be divided into subsections listing reagents and techniques. When citing previous methods, accurate references should be provided and any alterations should be noted. Information must be provided about: antibody dilutions, company names, catalogue numbers and clone numbers for monoclonal antibodies; sequences of RNAi and cDNA probes/primers or company names and catalogue numbers if reagents are commercial; cell line names, sources and information on cell line identity and authentication. Animal studies and experiments involving human subjects must be reported in detail, identifying the committees approving the protocols. For studies involving human subjects/samples, a statement must be included confirming that informed consent was obtained. Statistical analyses and information on the reproducibility of experimental results should be provided in a section titled "Statistics and Reproducibility".

All Nature Cell Biology manuscripts submitted on or after March 21 2016 must include a Data availability statement as a separate section after Methods but before references, under the heading "Data Availability". For Springer Nature policies on data availability see <http://www.nature.com/authors/policies/availability.html>; for more information on this particular policy see <http://www.nature.com/authors/policies/data/data-availability-statements-data-citations.pdf>. The Data availability statement should include:

- Accession codes for primary datasets (generated during the study under consideration and designated as "primary accessions") and secondary datasets (published datasets reanalysed during the study under consideration, designated as "referenced accessions"). For primary accessions data should be made public to coincide with publication of the manuscript. A list of data types for which submission to community-endorsed public repositories is mandated (including sequence, structure, microarray, deep sequencing data) can be found here <http://www.nature.com/authors/policies/availability.html#data>.
- Unique identifiers (accession codes, DOIs or other unique persistent identifier) and hyperlinks for datasets deposited in an approved repository, but for which data deposition is not mandated (see here for details <http://www.nature.com/sdata/data-policies/repositories>).
- At a minimum, please include a statement confirming that all relevant data are available from the authors, and/or are included with the manuscript (e.g. as source data or supplementary information), listing which data are included (e.g. by figure panels and data types) and mentioning any restrictions on availability.
- If a dataset has a Digital Object Identifier (DOI) as its unique identifier, we strongly encourage including this in the Reference list and citing the dataset in the Methods.

We recommend that you upload the step-by-step protocols used in this manuscript to the Protocol Exchange. More details can be found at www.nature.com/protocolexchange/about.

All imaging data should be accompanied by scale bars, which should be defined in the legend. Cropped images of gels/blots are acceptable, but need to be accompanied by size markers, and to retain visible background signal within the linear range (i.e. should not be saturated). The boundaries of panels with low background have to be demarked with black lines. Splicing of panels should only be considered if unavoidable, and must be clearly marked on the figure, and noted in the legend with a statement on whether the samples were obtained and processed simultaneously. Quantitative comparisons between samples on different gels/blots are discouraged; if this is unavoidable, it should only be performed for samples derived from the same experiment with gels/blots were processed in parallel, which needs to be stated in the legend.

- For line art, graphs, charts and schematics we prefer Adobe Illustrator (.AI), Encapsulated PostScript (.EPS) or Portable Document Format (.PDF). Files should be saved or exported as such directly from the application in which they were made, to allow us to restyle them according to our journal house style.
- We accept PowerPoint (.PPT) files if they are fully editable. However, please refrain from adding PowerPoint graphical effects to objects, as this results in them outputting poor quality raster art. Text used for PowerPoint figures should be Helvetica (preferred) or Arial.
- We do not recommend using Adobe Photoshop for designing figures, but we can accept Photoshop generated (.PSD or .TIFF) files only if each element included in the figure (text, labels, pictures, graphs, arrows and scale bars) are on separate layers. All text should be editable in 'type layers' and

line-art such as graphs and other simple schematics should be preserved and embedded within 'vector smart objects' - not flattened raster/bitmap graphics.

The total number of Supplementary Figures (not including the “unprocessed scans” Supplementary Figure) should not exceed the number of main display items (figures and/or tables (see our Guide to Authors and March 2012 editorial <http://www.nature.com/ncb/authors/submit/index.html#suppinfo>; <http://www.nature.com/ncb/journal/v14/n3/index.html#ed>). No restrictions apply to Supplementary Tables or Videos, but we advise authors to be selective in including supplemental data.

GUIDELINES FOR EXPERIMENTAL AND STATISTICAL REPORTING

REPORTING REQUIREMENTS – We are trying to improve the quality of methods and statistics reporting in our papers. To that end, we are now asking authors to complete a reporting summary that collects information on experimental design and reagents. The Reporting Summary can be found here <https://www.nature.com/documents/nr-reporting-summary.pdf> If you would like to reference the guidance text as you complete the template, please access these flattened versions at <http://www.nature.com/authors/policies/availability.html>.

We strongly recommend the presentation of source data for graphical and statistical analyses as a separate Supplementary Table, and request that source data for all independent repeats are provided when representative experiments of multiple independent repeats, or averages of two independent experiments are presented. This supplementary table should be in Excel format, with data for different figures provided as different sheets within a single Excel file. It should be labelled and numbered as one of the supplementary tables, titled “Statistics Source Data”, and mentioned in all relevant figure legends.

Author Rebuttal to Initial comments

Response to reviewers' Comments:

We truly thank all three reviewers for the encouraging comments, as well as careful reading of the manuscript and constructive suggestions. Despite a pandemic-related slight delay in carrying out the 4C experiments with inducible expression in mutant cells, we are now happy to provide a full response to all comments and suggestions. This has greatly improved the manuscript.

Please note that we switched the order in which we explain the first Haster and hnf1a KO experiments, as we felt it made more sense to start with Haster. We have marked all significant changes in blue, except typos, text reordering, or very secondary changes. We have also added array express entries for the 7 sets of NGS experiments. We will soon process Addgene entries for plasmids described here and deposit bigwig and bedfiles in our server.

Reviewer #1:

Remarks to the Author:

In "HASTER is a transcriptional stabilizer of HNF1A" the authors investigate the role of HASTER, a long non-coding RNA antisense to the promoter and first exon of HNF1A, which is a gene whose haploinsufficiency has been shown to cause diabetes mellitus. By employing both in vitro and in vivo model systems, they cleverly uncover both positive and negative feedback loops between HASTER and HNF1A, which ultimately stabilize HNF1A expression levels and maintain a proper 3D genome architecture during the early phases of liver development. They also demonstrate that the HASTER promoter per se is the main or only determinant of this activity, whereas the lncRNA molecule seems dispensable.

Moreover, disruption of a ~320bp region comprising Haster TSS was shown to be sufficient to cause phenotypes comparable to diabetes mellitus, pinpointing the importance of considering non-coding loci when investigating the genetics of this disease.

14This research work represents a thorough investigation of the relationship between a disease-causing gene (HNF1A) and a non-coding locus (HASTER), which results in a "balancing" interaction, which has not been reported before, as far as I know. This story will be of great interest to a broad field and therefore certainly fit the scope of Nature Cell Biology. Therefore, I have only minor comments which I propose to be addressed:

1) The graphical representations of the HASTER-HNF1A locus are sometimes confusing. For instance, while in Fig. 1B and 1F it is clear those two transcriptional units are antisense and overlapping, in others one may misunderstand and consider them as convergent and overlapping (e.g. Fig. 1A), convergent (Fig. 2A and C, Fig. 5C, D and F, Fig. 6B) or as completely separated units (Fig. 7D and F). I acknowledge that a graphical representation of two antisense and overlapping loci is conceptually challenging, but I also think an effort here can help to make sure that the readers properly interpret all the deep and careful experiments which have been done.

Thank you for this excellent suggestion as we had overlooked this. We have now added a simplified gene structure model in new Figure 1A and have tried to explain it more explicitly in the text. Hopefully once the reader is clear about this the other models will be understood even if they sometimes oversimplify and do not show all HNF1A exons.

2) Overall, all the experiments in the manuscript converge on the promoter of HASTER being important, and as far as I can see, there is no evidence that the transcription or the RNA product are important. Therefore, the authors need to be careful about referring to any "lncRNA function" throughout the manuscript. Specifically, they should not explicitly say that the disruption of a lncRNA can cause diabetes mellitus (e.g. final sentence in the abstract). While it is true as they beautifully demonstrate that a deletion of HASTER promoter region cause such phenotypes, it is also true that this happens within the 5' region of Hnf1a and the RNA product was found to be dispensable. Thus, currently there is no definitive proof that the disease-causing agent is Haster itself, and the authors do not mention human subjects with diabetes mellitus and genetic deletions in this locus.

We have now been much more careful to avoid referring to HASTER function as a lncRNA function. In the abstract we talk about a lncRNA promoter, and when comparing to other lncRNAs that have similar functions, we refer to them in aggregate as cis-acting lncRNA units.

3) Do the mice with the poly(A) insertion in HASTER (HasterSTOP) have any phenotype? Or were they only bred with HNF1A mice, and so the authors didn't have a chance to observe homozygous mice? The authors mention that HNF1A levels are not affected, but do these mice have diabetes or any

15

other observed phenotype? This is not essential, as the manuscript already has very extensive results, but can be of interest if the information is available or can be obtained rapidly.

Homozygous HasterSTOP mice were intended to block HASTER unidirectionally, but absolute quantification by digital droplet PCR of *Hnf1a* exon 1 vs. *Hnf1a* exons located beyond the poly(A) signal showed that the insertion of the poly(A) also directly impaired *Hnf1a* transcription in the + strand beyond the stop signal. Not surprisingly, poly(A) homozygous mice had reduced HNF1A and phenotypic features of *Hnf1a* null homozygotes (reduced body size, hepatomegaly, renal glycosuria, etc). This is of course influenced by loss of HNF1A function, because this was not observed after complete deletion of the lncRNA promoter. However, heterozygous HasterSTOP mice had no HNF1A-deficient phenotype, which allowed us to use this allele in heterozygosity in a hybrid background exclusively for the purpose of studying the role of RNA transcription and transcripts in regulating *Hnf1a* in the chromosome carrying the stop signal (**Fig. 5d, Supplementary Fig. 12**). We used it to ask if blocking Haster transcription in islets prevented *Hnf1a* transcription (exon 1, which is upstream of the poly A) in the same chromosome, as observed for germ-line Haster mutations, or if in liver this increased *Hnf1a*, as also observed for germ-line Haster mutations. We found that blocking transcription failed to affect both molecular phenotypes, which was consistent with experiments in which we activated and inhibited HASTER transcription in cell lines (**Fig. 5, Supplementary Fig. 13c,d**).

4) What happens to HASTER expression if the authors over-express the HNF1A cDNA? Currently a CRISPRa experiment is presented (Fig. S3b), but its more difficult to interpret since if HASTER promoter is in physical proximity to the HNF1A promoter CRISPRa can “leak” from one promoter to another. I do not consider it an essential experiment, but if it can be performed, it can strengthen the results.

We agree that CRISPRa experiments could theoretically have that effect. HNF1A overexpression in EndoC-BH3 cells was performed with an inducible system from an exogenous location, and this was shown to upregulate *HASTER* while downregulating *HNF1A* (**Fig 6a**). This should bypass those concerns. We have added additional experiments that further dispel doubts about this type of phenomenon. We have overexpressed HNF1A mutants that have greatly diminished transcriptional activity (IDR mutants), but still inhibit HNF1A, and added an extra IDR to HNF1B that increased HASTER RNA but did not affect inhibition of HNF1A. This data provides further evidence to uncouple HASTER transcription from HNF1A inhibition (New **Fig 6c, Supplementary Fig. 14**).

Reviewer #2: Remarks to
the Author: Summary

16Eukaryotic genome are transcribed extensively and many lncRNAs have been shown to act as local regulators to influence the expression of nearby genes. However, it's challenging to dissect whether the regulatory function is through the lncRNA transcripts themselves or results from cis regulatory elements at lncRNA locus. In this manuscript submitted by Beucher et al., the authors focused on a specific lncRNA HASTER to study its potential biological function. Using mouse and human models, the authors showed that: (1) HNF1A is a positive regulator of HASTER; (2) HASTER is able to regulate HNF1A transcription through positive and negative feedback loops; (3) HASTER inactivation, which shows variegated HNF1A overexpression or silencing, leads to impaired insulin secretion and diabetes; (4) HASTER promoter but not HASTER transcript or transcription act, regulates HNF1A transcription; (5) HASTER promoter regulates HNF1A by remodelling a specific enhancer-promoter interaction in the 3D chromatin context. From their results, the authors conclude that they discover a cis-regulatory element that is unlike enhancers or silencers, and instead stabilizes expression levels of a pioneer transcription factor linked to diabetes phenotype.

Review

This paper addresses an interesting topic linking an lncRNA promoter to diabetes disease. A strength of the work is using the mouse model to show the biological relevance of HASTER promoter in diabetes progression. Weaknesses include

(1) lack of underlying molecular mechanism to explain the variegated expression pattern of HNF1A in a homozygous HASTER knockout pancreatic endocrine context;

In this new version of the manuscript, we provide several experiments (described below) that further strengthen the molecular mechanism for how HASTER inhibits HNF1A. We also provide arguments to indicate that the mechanism whereby HASTER activates HNF1A in the endocrine pancreas is analogous to a transcriptional enhancer (see discussion below in response to point #2).

More intriguingly, what determines which cell sub-population in pancreatic endocrine cells that will exhibit silencing and overexpression of HNF1A in a homozygous HASTER knockout context?

The observed phenotype is consistent with two opposed cis-regulatory functions acting on beta cells, and leading to quasi stochastic outcomes due to failure of both functions. We hope to provide a clearer interpretation of the model we propose for the existence of HNF1A OFF or superactive states in beta cell KO cells (discussion, second paragraph):

HASTER's dual function was most compellingly illustrated by the pancreatic knock-out phenotype, in which lack of HASTER enhancer-like activity led to a reduced probability of activation of HNF1A in β

cells, while lack of negative feedback caused overexpression in β cells that succeeded in activating Hnf1a.

We note that there is no indication from our single cell RNA data, or in general from the single cell islet literature, to indicate that there is a discrete subpopulation of beta cells in which Haster KO could plausibly determine one state or the other. In the manuscript we do show, however, that the OFF state frequency increases as a function of when HASTER is deleted. In cells that have never seen HASTER (e.g. germline KOs) the OFF state is much more common than cells that have been previously exposed to HASTER (i.e. when HASTER is knocked out late in development).

(2) the enhancer hijacking mechanism proposed by the authors reasonably explains the inhibitory effect of HASTER promoter on HNF1A transcription. However, what's the mechanism contributing to HASTER activation function to HNF1A transcription shown in pancreatic endocrine context?

We have shown that HASTER exerts negative feedback in all major mouse and human cell types in which HASTER is expressed (endocrine and exocrine pancreas, liver, and not shown in gut and kidney). The only cell-specific effect is the obligate requirement of HASTER for positive regulation. Given that HASTER-dependent positive regulation acts in cis, it is located in an intronic region, and it transcribes an RNA that is actually not essential for its effects, we conclude that it is analogous to an enhancer— in fact it is supported by more experimental evidence than most enhancers in the literature. Given that enhancers form clusters that contain some degree of redundancy, the notion that an enhancer is expressed in more than one cell type but essential in only one is does not counter current views of cis-acting elements.

Concerning the precise mechanism for cis-acting enhancer activity, this is a fundamental general question in enhancer biology that is currently unsolved. Thus, while we understand that this is perceived as a limitation because it is an unsolved problem, we hope that the editor and reviewer accept that we have not solved all possible questions raised by our findings. We believe it is still important that we have conclusively demonstrated that HASTER does work in cis to ensure transcription of HNF1A during early developmental time points, and that genetic analysis shows that transcription of HASTER is not required, whereas the promoter DNA is essential for this function.

From a more speculative standpoint, there is growing evidence that enhancers are functionally heterogeneous, and a subset have a remarkable capacity to bind RNA-polymerase II. In one recent study (Thomas et al, Mol Cell, 2021 ¹) deletion of one such highly

transcribed enhancer had a greater impact than all other enhancers on target gene transcription. The positive effects of the Haster promoter are at face value analogous to this type of enhancer. We can speculate that one function might be to recruit RNA pol to a locus, but we have not addressed this further and wish to emphasize that this is not really the scope of this study.

All of this said, we have tested Haster promoter sequence for intrinsic enhancer activity in episomal luciferase assays, using plasmids carrying a minimal promoter, and found that while having a strong promoter activity when placed in front of a minimal promoter, it had only weak enhancer activity when placed downstream the luciferase gene. We and others have tested many “enhancer like” (H3K27ac+ H3K4me3-) sequences in the past and find many that do not behave as enhancers in such assays, and in fact Thomas et al, Mol Cell, 2021 showed that this is particularly true for enhancers that show strongest recruitment of RNA Pol II.

More intriguingly, what determines which cell sub-population in pancreatic endocrine cells that will exhibit silencing and overexpression of HNF1A in a homozygous HASTER knockout context?

Concerning the variegated phenotype, this is best explained by the fact that we inactivated both mechanisms, and both are essential in early pancreatic endocrine cells. In a setting that lacks mechanisms to prevent HNF1A silencing or overexpression, we observe both silencing or overexpression are possible. As far as we can see this occurs in a random fashion when HASTER is knocked out at early stages, although silencing clearly prevails in early KO settings. We now explain this proposed model in the second paragraph of the discussion:

HASTER’s dual function was most compellingly illustrated by the pancreatic knock-out phenotype, in which lack of HASTER enhancer-like activity led to a reduced probability of activation of HNF1A in β cells, while lack of negative feedback caused overexpression in β cells that succeeded in activating Hnf1a.

(3) in the proposed model wherein the authors claim that at low HNF1A concentrations, there are unhindered interactions between the HNF1A promoter and E (or other local enhancers); while at high HNF1A concentrations, HNF1A binding to HASTER limits contacts between E and HNF1A, thereby decreasing HNF1A transcription. It implicates that HNF1A binding to HASTER promoter might hijack enhancer E to prevent contact between HNF1A promoter and enhancer E. However, there is no data to show that HNF1A can regulate interaction between these loci.

Thank for this important suggestion. We have now addressed whether HNF1A can directly

regulate interactions between the *HNF1A* promoter and E. We used a lentiviral-based overexpression system for HNF1A in the human beta cell line EndoC- β H3, carrying either wild type or homozygous deletion of the *HASTER* promoter region, and performed UMI4C experiments with a viewpoint on the *HNF1A* promoter. In *HASTER* wild type cells, overexpression of HNF1A led to a significant decrease in contact frequency between *HNF1A* promoter and E, showing that HNF1A expression levels controls interactions between *HNF1A* promoters and its enhancers (fold change Dox/Vehicle = 0.89, Chi-square P = 0.05). Importantly, this was not observed in *HASTER* KO cells (fold change Dox/Vehicle = 1.00, Chi-square P = 0.78; **Fig. 7e,f**).

Additional comments:

1. The conclusion drawn in the abstract that “They also show that disruption of a mammalian lncRNA can cause diabetes mellitus.” is overstated. Throughout the manuscript, the authors aim to claim that *HASTER* promoter itself, but not *HASTER* transcript regulates HNF1A transcription that leads to diabetes phenotype.

We had used the term “lncRNA” loosely to refer promoter + transcript units, but this is easily misinterpreted and hence needs to be modified, as it might even imply that we have found a function for the RNA, or that we think that all lncRNA functions are encoded by the DNA, but of which are incorrect.

We have now reworded this sentence, and state that “disruption of a mammalian lncRNA promoter can cause diabetes mellitus” which is more accurate. We have also modified other sentences throughout the manuscript that implied that we have identified functional or disease relevance for a lncRNA, rather than a promoter.

2. For Supple Fig. 3a, Locked nucleic acid (LNA) gapmer ASO not only triggers RNase-H- mediated transcript cleavage but also induces premature transcription termination (See Jong-Sun Lee et al., Antisense-Mediated Transcript Knockdown Triggers Premature Transcription Termination Mol Cell 2020). Hence, the result shown in Supple Fig. 3a may be from LNA- induced premature transcription termination at *HNF1A* locus that also disrupts *HASTER* transcription. LNA sequences used for *HNF1A* mRNA knockdown should be listed. Western blot data examining *HNF1A* knockdown efficiency should be provided.

Thank you for bringing up this possibility. We agree that Gapmers directed against *HNF1A* could indirectly affect *HASTER* expression. While this could be true for one of the Gapmers that targets exon 1, it is less likely with the 2nd Gapmer that targets exon 8 (out of 10) of *HNF1A*. We now mention this and include the reference to highlight this in the

supplementary figure legend.

Concerning demonstrating that HNF1A knockdown. We show that gapmers downregulate *HNF1A* mRNA by ~50% (**Supplementary Fig. 5a** in the revised version). This level of downregulation can be difficult to quantify accurately by Western Blot, but we have instead analyzed a read-out of HNF1A function; *HNF4A* is a direct target of HNF1A and is functionally dependent on HNF1A, and therefore *HNF4A* mRNA levels are a good indicator of HNF1A activity (see also **Fig. 2i**, **Fig. 5g,h**). *HNF4A* levels were significantly decreased with both gapmers, showing that HNF1A was indeed downregulated.

Sequences of LNA Gapmers are provided in now **Supplementary table 7**.

We confirmed the HNF1A dosage sensitivity of *HASTER* using a TetON system to overexpress HNF1A, and observed a progressive upregulation of *HASTER* when increasing the Dox concentration.

We should also mention that HNF1A-dependence per se is not in doubt since it is demonstrated by HNF1A null mutations in various contexts.

3. For Supple Fig. 3b, as HNF1A promoter is close to *HASTER* promoter, it's expected to see that CRISPR-SAM strategy (which may activate both two promoters) leads to increased *HASTER* RNA expression. dCas9 CHIP experiment should be performed to prove that CRPSRa strategy used to target HNF1A promoter specifically targets HNF1A promoter but not *HASTER* promoter.

This experiment was meant show that HNF1A regulates *HASTER*. We performed dCas9 ChIPs, but we were unable to detect a significant enrichment of the dCas9 at the *Hnf1a* promoter. Therefore, we cannot use this approach to formally exclude an indirect activation of *Haster* promoter when targeting the *Hnf1a* promoter.

However, we think this not likely because targeting *Haster* promoter using CRISPRa does not lead to a significant change in expression of the nearby gene *Hnf1a* (new **Fig. 5h**). More importantly, overexpression of HNF1A through transfection or lentiviral transduction showed that HNF1A directly upregulated *HASTER* (**Fig. 6a,b**), confirming the *Hnf1a* CRISPRa results. Likewise, all HNF1A mouse and human KO and knockdown cells show that HNF1A positively regulates *HASTER* (and ChIP-seq shows that HNF1A directly binds to multiple HNF1A recognition sequences in *HASTER*). The CRISPR-SAM experiments is therefore one

21of several independent experiments that converge on this same conclusion.

4. For scRNA-seq data, a table summarizing all the different samples that were processed for sequencing and the number of high quality nuclei sequenced should be added.

This information was reported in **Supplementary table 2** but was only referenced in methods, we have now added a reference in the main results section.

Furthermore, age, sex and genetic background of the animals from which the sequencing samples were derived should be disclosed in the method section.

Age, sex and background were added in **Supplementary table 2**.

The authors conclude that HASTER deletion in vivo caused functional HNF1A deficiency in pancreatic β' cells resulting in diabetes. Have the authors tried different parameters or other clustering strategy to identify this molecularly distinct cell subtype?

To confirm that the HASTER KO-specific cluster can be detected independently of the method used, we analysed the scRNA-seq data using scVI that uses a different integration and clustering methods (PMID: 30504886) than Seurat v3. We now show that scVI is in fact more efficient at capturing a KO-specific cluster that has the same HNF1A-deficient signature.

Why is this cell subtype not shown in control samples?

We think that the fact that we called these cells “beta prime cells” led to confusion, as it sounded as if these are a pre-existing beta cell subtype that are predisposed to being HNF1A-deficient in the HASTER KO. This is not really a subtype of beta cells. Instead, it’s just a subset of HASTER KO beta cells that acquire an HNF1A OFF state. They do not exist in WT cells. As far as we can tell this is stochastic (see discussion above), although the earlier the KO the higher the probability of finding OFF cells.

We performed clustering with controls and KO islets, and virtually do not observe cells in this cluster in pooled controls. We see a few scattered cells in Seurat, surely contaminants of the clustering, and none with scVI. By contrast, they are clearly seen in each individual HASTER KO replicate (**Supplementary Fig. 9g**). This is consistent with the HNF1A immunofluorescence studies.

We have now explained in results and methods that the existence of HNF1A-deficient cells in HASTER KO islets was robust to different clustering methods. In particular in methods we

22

describe the use of Seurat v3 integration through canonical correlation analysis with graph-based clustering (Louvain algorithm), and scVi intergration using deep neural network and clustering (Leiden algorithm)) and parameters.

We have also renamed KO sub-clustered beta cells as HNF1A^{high} and HNF1A^{low} beta cells, which is more coherent with the IF studies.

It seems that the scRNA-seq data can't explain the diabetes phenotype observed in HASTER knockout mice, as the major cluster of β cells didn't show deficient expression of HNF1A-regulated genes (instead shows increased HNF1A and its regulated genes). The authors should discuss the scRNA-seq data (including other cell clusters shown in supple Fig. 9) in more detail in the text or discussion part.

These experiments were done in female KOs (as listed **Supplementary Table 2**) which have a much milder glucose intolerance phenotype. Still, the percentage of HNF1A-negative cells is lower in the single cell analysis than expected based by in situ immunofluorescence. With scVI we found 5-21% of β cells were "HNF1A low", whereas in male pKO mice we expected 30-60% (**Supplementary Fig. 9f, Supplementary Fig. 8a**)

Having worked with HNF1A KO mice for a couple of decades, we are well aware that HNF1A KO islet cells markedly dissociate during islet isolation (a step that precedes single cell dispersion). Dissociation of islet cells is actually very common with several TF KO mice, which leads to recovery of fewer, smaller islets. This is expected to cause negative selection of HNF1A-deficient cells. In response to this reviewer's comment we now mention this in the results.

We feel that the concept that some cells are HNF1A-deficient, and other overexpress HNF1A is proven beyond reasonable doubt in these experiments, even though the percentages vary.

5. Can the authors also perform UMI-4C in control and HASTER KO β cells? It is of relevance to the biologic phenotype. In addition, as mentioned above, to validate the model, it's worth doing UMI-4C after overexpression of HNF1A to check whether decreased contact between enhancer E and HNF1A promoter is observed.

This experiment has been performed in response to this reviewer's suggestion (**Fig. 7f and Supplementary Fig. 16b**). Deletion of HASTER promoter in β cells significantly increased HNF1A promoter-E interactions. Furthermore, we failed to detect any significant change of interactions between the HNF1A promoter and E when HNF1A was overexpressed in

23

HASTER KO β cells (fold change Dox/Vehicle = 1.00, Chi-square P = 0.78), while the interaction was reduced upon HNF1A overexpression in wild-type β cells (fold change Dox/Vehicle = 0.89, Chi-square P = 0.05).

6. What's the underlying mechanism by which HASTER promoter is able to hijack enhancer E? A plausible explanation is that high HNF1A binding affinity at HASTER promoter favors HASTER promoter-enhancer E interaction. Can the authors test it by performing UMI-4C experiment in control cells and cells where HNF1A binding motifs at HASTER promoter shown in Fig. 1f are edited or deleted?

This mechanism is indeed our model, and it is supported by new experiments including the UMI 4C with overexpression HNF1A (**Fig. 7f** and **Supplementary Fig. 16**). Furthermore, we now provide new experiments showing that “transactivation dead” HNF1A mutants still repress HNF1A, and this requires an intact HASTER promoter.

We have not been able to specifically and directly test HASTER-E interactions, because HASTER promoter is < 2.5 kb from E, which is below the resolution limit of the UMI4C.

7. Can the authors use a large window to analyze UMI-4C data to check other potential enhancers interacting with HNF1A promoter? As in WT cells, HASTER knockout doesn't change HNF1A mRNA expression, implying other enhancers may be involved to promote HNF1A transcription in the presence of HASTER.

We understand that this question refers to the enhancer knockout, rather than Haster knockout.

In human beta cells we identified 33 candidate enhancers within a ~1 Mb window that has *HNF1A* promoter contacts, and quantified contacts in a 5 kb window centred on each enhancer, as we did for E (**Supplementary Fig. 16a**). Only E showed a significant change in contact frequency with the *HNF1A* promoter after HNF1A overexpression. It is possible that other enhancers increase HNF1A expression when HASTER is inactive but with weaker effects that cannot be captured by the 4C assay. It is perhaps worth noting that the E region contains multiple clustered accessible sites.

We now state this in the main text, and believe it substantially strengthens the conclusions “*Out of 33 enhancers sites in in 1 Mb surrounding HNF1A only E showed significant HNF1A- dependent changes*”

8. Is the “Zebrafish” labeled at the bottom of Fig. 1a and Supple Fig. 1b correct?

24The label is correct for Zebrafish. Indeed the DNA sequence of the HASTER promoters is less conserved in Zebrafish. Previously we had not detected evidence for transcription in zebrafish, but analysis of new Zencode data shows antisense transcription starting from the first intron of *hnf1a*. We have added the zebrafish liver RNA-seq tracks to **Fig. 1b**.

9. Some typographical errors should be corrected. For example, at Line 264, “Hnf1a knock our mice” should be “Hnf1a knock out mice”; at Line 328, “ the inhibitory effects of HNF1A overexpression where almost completely suppressed” should be “the inhibitory effects of HNF1A overexpression were almost completely suppressed”; In Supple Fig. 12, “showing that reducing Haster elongation in failed to increase Hnf1a expression from the same C57BL/6 allele.” should be “showing that reducing Haster elongation in liver failed to increase Hnf1a expression from the same C57BL/6 allele.”

Corrected, thank you

Reviewer #3:

Remarks to the Author:

Beucher et al., Nat Cell Biol

The study by Beucher represents an experimental tour de force to decipher the role of the HNF1A-AS1 lncRNA that is transcribed from a promoter in the first intron of diabetes-relevant gene HNF1A. The authors develop multiple new models, both in gene-targeted mice and in cell lines and apply a whole range of advanced techniques to their analysis. It is clear from all these models that the first intron of HNF1A is important to the regulation of its expression. The findings reported are sometimes puzzling, as the authors observe both positive and negative consequences of decreasing HNF1A-AS1 transcripts, even within the same model. Fascinating but not truly explained is the observation that in some pancreatic beta cells with the same altered genetic makeup, HNF1A expression is increased while in others it is decreased. Yes, the authors conclusion that their manipulation affected the stability of HNF1A gene regulation is true, but it is not clear how. Finally, it appears in the end that the HNF1A-AS1 transcript does not matter, but the DNA element that encodes it does, suggestions that this acts more like a critical enhancer.

Abstract:

- Please remove the term “pioneer” transcription factor, as this was not formally proven for HNF1alpha. DNA binding transcription factor, or lineage-determining transcription factor is more accurate.

25We have removed the term “pioneer” from the abstract.

We prefer to leave the discussion in the text that biochemical studies from Ken Zaret’s lab show that HNF1A is among a list of transcription factors (including FOXA, OCT4,...) that binds nucleosomal DNA as well as naked DNA, while programming experiments show that HNF1A can derepress inactive chromatin (all of which are hallmarks of so-called pioneer factors). Since our findings show that HNF1A alone, simply by increasing concentrations, can open up chromatin and even create active promoters in regions that are completely repressed in normal liver, we feel it is of interest to mention this and use the term “pioneer-like” in this discussion. But we are happy to avoid giving this term more protagonism that needed within this article and have accordingly removed it from the abstract.

Introduction:

Remove the end of the introduction, where you summarize the results. While I realize that this is now common practice, it is superfluous. A single sentence stating what is being investigated here suffices.

We have deleted two sentences and left a single sentence that tries to describe in a nutshell what we have done.

Nomenclature: NCBI has no entry for HASTER, but multiple for HNF1A-AS1. In fact, over 20 publications in PubMed refer to the locus as HNF1A-AS1. Renaming a gene that is already annotated is poor form, and only adds to confusion in the field. Please change all references to HASTER, both in text and figures, to HNF1A-AS1

We agree that a gene that has been named should not be renamed, but HNF1A-AS1 is an automatic annotation that refers to an antisense RNA transcript, but not to the cis element we describe. All publications so far (several cited in our original manuscript) use overexpression and knockdowns. Although we set out to study a lncRNA, all of our positive findings refer to the promoter DNA. It is the promoter DNA that acts as a cis-regulatory element that stabilizes HNF1A, not the lncRNA itself. It is important that we refer to this as a stabilizer cis element, rather than a lncRNA (HASTER is an acronym for HNF1A stabilizer). Conversely, it would be wrong to name this cis DNA element as an antisense transcript.

In the original version we loosely referred to this as one type of “lncRNA function”, but reviewer comments have suggested to avoid this equivocal choice of words. In addition to complying to the suggestion of not referring to “lncRNA function” we now more carefully describe what we mean by HASTER at the outset of the results section:

This lncRNA is annotated as HNF1A-AS1, or Hnf1a-os1 and 2 in mouse. We refer to the promoter region of this lncRNA, along with transcribed products, as HASTER, for HNF1A stabilizer. HNF1A

antisense transcripts have been previously proposed to exert trans- regulation of proliferation in cell-based models²⁻⁷, but so far the transcriptional regulatory function of HASTER DNA or transcribed regions has not been characterized with genetic tools.

Results:

There have been more than 20 studies on HNF1A-AS1, implicating this transcript in cancer, with multiple mechanisms that differ from the current study proposed. May be the published work is all wrong, but it needs to be discussed.

We cited (and cite) six of those studies at the outset in the original manuscript. It is difficult to discuss all studies, but commonalities include that all of these studies which come from the same labs have used RNA inhibition and focus on cancer phenotypes. We also mention in the first paragraph that genetic studies have not been performed (and cis-regulatory function has not been explored)

To summarize our view (a) the genetic phenotype we observe is not captured by those RNA inhibition/overexpression studies, which is expected because our RNA inhibition studies have shown that the RNA is not involved in our phenotypes, and the other studies have not modified the DNA element; (b) We find it is counterintuitive for a lncRNA that exclusively resides at the site of transcription should have strong trans effects (please see improvements to the smFISH figures described below). However, beyond citing those papers, we can't really speculate on their quality, as we have not tried to perform RNA inhibition in cancer cells lines as performed in such studies.

Formally, we do not exclude that the lncRNA/transcription does have a function, namely some form of cis-regulatory function. We can only state that RNA/transcription cannot account for the genetic phenotypes we observe. We now make the following general statement in the second paragraph of the discussion

We show that HASTER's dual function emanates from a 320 bp promoter DNA sequence. Transcription was therefore not essential, although it remains possible that antisense transcripts exert modulatory functions that have not been explored.

Figure 1:

Figure 1A: The HNF1A-AS1 transcripts are curious. While annotated in NCBI as a spliced

gene with multiple exons, these tracks only show a single un-spliced transcript (while for HNF1A only the annotated exons have reads. How do the authors interpret this discrepancy?

A common feature of many nuclear lncRNAs is their complex alternative splicing with frequent intron retention^{8,9}. Due to this, the RNA-seq pileup should not be interpreted as a single exon but as the collapse of overlapping exons and retained introns. Our 3' RACE and the annotated transcripts are just the tip of the iceberg of the transcript diversity. In unpublished work we have performed long read sequencing in pancreatic islets and observed 36 distinct transcript isoforms, and even there the overlap with 3' RACE and GENCODE annotations is incomplete.

We now describe this in slightly more detail:

We used CAGE-seq, RNA-seq and 3' RACE to show that HASTER has a myriad of transcript isoforms that originate from a major upstream transcriptional start site in human islets, and an additional downstream start site in other tissues (Fig. 1a, Supplementary Fig. 1a).

In addition, for each species, the + and – strand reads need to be shown on the same scale, so that the reader can appreciate the relative abundances of the two transcripts.

This has been done

Figure 1B: this is confusing. There is only one spot where transcripts for HNF1A and HNF1A-AS1 overlap, but multiple that show only HNF1A-AS1. The co-localization needs to be quantified, showing a single dot is not convincing.

The colocalization quantifications were added to **Supplementary Fig. 2**. The colocalization between *HASTER* and the *HNF1A* locus can only be accurately assessed by smFISH in single cell alleles in which *HNF1A* is being actively transcribed and detected. Therefore, *HASTER* can be detected without the presence of the nascent *HNF1A* transcript. Importantly, in all cells in which both *HNF1A* loci were detected and *HASTER* was detected we found 100% colocalization with *HASTER*.

Supplementary Fig. 2 now also provides low powered images showing that *HASTER* is not seen in the cytoplasm, consistent with the fractionation RT-PCR studies.

Figure 1C-F, interpretation. The data clearly show that there is an *HNF1A* dependent cis-regulatory element that impacts *HNF1A-AS1* transcription. It is also possible, however, that the element with 7 *HNF1*alpha binding sites also serves as an intronic enhancer controlling *HNF1A* expression itself.

We conclude that the positive cis-acting function of this promoter is indeed analogous to an enhancer, and present several experiments showing that this element is essential in pancreatic progenitors and beta cells. We now discuss this more clearly in the second paragraph of the discussion:

HASTER's activating function also occurred in cis and was not dependent on antisense transcription. The HASTER promoter, therefore behaves as an intronic enhancer that is only essential in pancreatic endocrine cells, presumably due to cis-regulatory redundancy in other cell types.

What happens to HNF1A transcripts in HNF1A null tissues?

This is of course an interesting question. RNA-seq from adult islets of germ-line *Hnf1a* KO mice show a severe reduction HNF1A mRNA molecules (see panel a). This is consistent with HNF1A being required to activate HASTER enhancer-like activity. We know that HASTER is essential for activation of HNF1A in all islet endocrine cells during development. We would need to create a complex genetic model to prove that HNF1A is not acting through some other essential positive regulatory element, although HASTER is the main HNF1A binding site in islets. Quite remarkably, *Hnf1a* mRNAs are increased in islets from a conditional model that knocks out *Hnf1a* in adult mice, after beta cells have formed (Pdx1CreER). Overall, these findings suggest (a) a positive role of HNF1A to activate HASTER enhancer-like activity in developing beta cells, and (b) because HASTER enhancer-like activity is not essential after birth, negative feedback prevails if *hnf1a* is knocked out after birth. Lack of functional HNF1A leads to increased transcription of *Hnf1a*. We are tempted to provide this as a supplementary Figure and explain it briefly in the second paragraph of the discussion, but this would require an additional series of experiments to further characterize this effect, and prefer to share this with the reviewers.

Figure for reviewers. (a) Expression of *Hnf1a* mRNA in islets from adult *Hnf1a*^{-/-} and wildtype littermates. (b) Expression of *Hnf1a* mRNA in islets isolated from Pdx1-CreER mice,

and treated with tamoxifen for 48 hours in culture prior to RNA extraction.

Figure 2. Very strong data in this figure. Unfortunately, as is true with all lncRNAs/antisense transcripts near transcription factor genes, the interpretation of the data is very difficult. The data are also compatible with the alternative explanation that the DNA sequences deleted, which are contained within intron 1 of HNF1A, are controlling HNF1A expression through directly changing the binding of transcription factors or altering chromatin structure. These possibilities need to be discussed.

As in previous comments, we agree entirely that it is the DNA cis element that is responsible for positive regulation, and we now more clearly refer to this as an enhancer-like activity. We discuss this explicitly in the second paragraph of the discussion

The effect on HNF1A transcript, while statistically significant, is marginal. The effects on protein levels shown in 2f appear stronger. The Western blots needs to be quantified, and should there be a larger effect size, this divergence needs to be explained.

We agree that the effects on HNF1A mRNA and protein are of different magnitude, and this was consistent. We have not done a time-course, which would be very difficult in these models, so we can only speculate that this could be due to differences in RNA and protein dynamics, for example build-up of HNF1A protein could occur if the half life is much longer, or if there is increased translation of newly transcribed but short-lived mRNA. Even more speculatively it could also be a post-transcriptional effect of HASTER lncRNA that we have not explored. A wild speculation is that HASTER could promote HNF1A mRNA modifications. We take the opportunity to mention that we have not excluded the existence of RNA effects, we simply show that the major genetic phenotypes are not dependent on the RNA or transcription, and have stated this in the discussion.

We have quantified the Western blot quantified for experiment 1, and provide this in the results:

Consistent with the hESC differentiation model, Haster^{LKO} mice showed increased liver Hnf1a mRNA (1.5 ± 0.3 fold) and protein (4.5 ± 0.6 fold) (Fig. 2d-f).

Figure 3: Very nice data showing that increased HNF1A levels lead to occupancy at non-canonical sites in the liver.

Thank you

Figure 4 and 5: Good data, and strong models. However, the data could also be interpreted

as deletion of DNA sequences in the HNF1A intron 1 alters TF binding, chromatin structure etc independent of HNF1A-AS1 promoter activity.

This interpretation is indeed correct and is further supported by new data showing that an IDR within HNF1A is important to increase HASTER transcription and dispensable for the repressing activity on HNF1A (**Fig. 6c**), whereas both HASTER activation and HNF1A repression require binding of HNF1A to the HASTER promoter (**Fig. 6b, Supplementary Fig. 14**).

- 1 Thomas, H. F. et al. Temporal dissection of an enhancer cluster reveals distinct temporal and functional contributions of individual elements. *Mol Cell* **81**, 969-982 e913, doi:10.1016/j.molcel.2020.12.047 (2021).
- 2 Ding, C. H. et al. The HNF1alpha-regulated lncRNA HNF1A-AS1 reverses the malignancy of hepatocellular carcinoma by enhancing the phosphatase activity of SHP-1. *Mol Cancer* **17**, 63, doi:10.1186/s12943-018-0813-1 (2018).
- 3 Wang, C. et al. Long non-coding RNA HNF1A-AS1 promotes hepatocellular carcinoma cell proliferation by repressing NKD1 and P21 expression. *Biomed Pharmacother* **89**, 926-932, doi:10.1016/j.biopha.2017.01.031 (2017).
- 4 Wang, K. C. et al. A long noncoding RNA maintains active chromatin to coordinate homeotic gene expression. *Nature* **472**, 120-124, doi:10.1038/nature09819 (2011).
- 5 Yang, X. et al. Long non-coding RNA HNF1A-AS1 regulates proliferation and migration in oesophageal adenocarcinoma cells. *Gut* **63**, 881-890, doi:10.1136/gutjnl-2013-305266 (2014).
- 6 Zhan, Y. et al. Long non-coding RNA HNF1A-AS1 promotes proliferation and suppresses apoptosis of bladder cancer cells through upregulating Bcl-2. *Oncotarget* **8**, 76656-76665, doi:10.18632/oncotarget.20795 (2017).
- 7 Zhu, W. et al. Knockdown of lncRNA HNF1A-AS1 inhibits oncogenic phenotypes in colorectal carcinoma. *Mol Med Rep* **16**, 4694-4700, doi:10.3892/mmr.2017.7175 (2017).
- 8 Tilgner, H. et al. Deep sequencing of subcellular RNA fractions shows splicing to be predominantly co-transcriptional in the human genome but inefficient for lncRNAs. *Genome Res* **22**, 1616-1625, doi:10.1101/gr.134445.111 (2012).
- 9 Zuckerman, B. & Ulitsky, I. Predictive models of subcellular localization of long RNAs. *RNA* **25**, 557-572, doi:10.1261/rna.068288.118 (2019).

Decision Letter, first revision:

18th March 2022

Dear Dr. Ferrer,

Thank you for submitting your revised manuscript "HASTER is a cis-acting transcriptional stabilizer of HNF1A" (NCB-F45367A). It has now been seen by the original referees and their comments are below. The reviewers find that the paper has improved in revision, and therefore we'll be happy in principle to publish it in Nature Cell Biology, pending minor revisions to comply with our editorial and formatting guidelines.

The current version of your manuscript is in a PDF format. Please email us a copy of the file in an editable format (Microsoft Word or LaTeX)-- we can not proceed with PDFs at this stage.

Thank you again for your interest in Nature Cell Biology Please do not hesitate to contact me if you have any questions.

Sincerely,

Jie Wang, PhD
Senior Editor
Nature Cell Biology

Tel: +44 (0) 207 843 4924
email: jie.wang@nature.com

Reviewer #1 (Remarks to the Author):

The authors have addressed all the comments from the previous line of review in a satisfactory manner. I can now recommend publication in Nature Cell Biology

Reviewer #2 (Remarks to the Author):

32Nature Cell Biology manuscript number NCB-F45367A by Beucher et al. (Jorge)
"HASTER is a cis-acting transcriptional stabilizer of HNF1A" (revised title)

In this paper, the authors use both mouse and human models to examine HASTER, finding HASTER promoter, but not the act of HASTER transcription or the lncRNA HASTER, exerts both activating and inhibitory functions to fine tune HNF1A expression. They also find Haster mutant pancreatic β cells consequently showed variegated HNF1A overexpression or silencing, causing insulin-deficiency and diabetes.

As I noted in my previous review, this paper addresses an interesting and timely topic. The development of multiple mice models used in the paper to examine the biological function and the corresponding molecular mechanisms of the HASTER promoter is a strength of the paper.

I also noted the following that needed improvement: (1) lack of underlying molecular mechanism to explain the variegated expression pattern of HNF1A in a homozygous HASTER knockout pancreatic endocrine context, (2) more description and discussion to support mechanistic conclusions, (3) extra unbiased scRNA-seq data analysis to strengthen the statement, (4) appropriate UMI-4C experiments to clarify the proposed model.

In their revised version of the paper, the authors have addressed many of these concerns. Key improvements include:

- Additional UMI-4C experiments validating the functional importance of the HASTER promoter to inhibit HNF1A transcription;
- The proposed enhancer-like function of HASTER to explain the activation function of HASTER.
- Stronger connection between the experimental data and the proposed mechanism. This includes extra UMI-4C experiments and careful explanation of experimental data from scRNA-seq.

Overall, I think the authors have done a good job addressing my concerns. At this point, in my opinion, the novelty of conclusions outweigh any remaining deficiencies in the experimental design and presentation.

Reviewer #3 (Remarks to the Author):

The authors have addressed all concerns made in my original review, and clarified the function of HASTER as a cis regulatory element. I would like to congratulate the authors on a very carefully performed and interpreted study. Great work!

30th March 2022

Dear Dr. Ferrer,

33Thank you for your patience as we've prepared the guidelines for final submission of your Nature Cell Biology manuscript, "HASTER is a cis-acting transcriptional stabilizer of HNF1A" (NCB-F45367A). Please carefully follow the step-by-step instructions provided in the attached file, and add a response in each row of the table to indicate the changes that you have made. Ensuring that each point is addressed will help to ensure that your revised manuscript can be swiftly handed over to our production team.

We would like to start working on your revised paper, with all of the requested files and forms, as soon as possible (preferably within one week). Please get in contact with us if you anticipate delays.

In recognition of the time and expertise our reviewers provide to Nature Cell Biology's editorial process, we would like to formally acknowledge their contribution to the external peer review of your manuscript entitled "HASTER is a cis-acting transcriptional stabilizer of HNF1A". For those reviewers who give their assent, we will be publishing their names alongside the published article.

Nature Cell Biology offers a Transparent Peer Review option for new original research manuscripts submitted after December 1st, 2019. As part of this initiative, we encourage our authors to support increased transparency into the peer review process by agreeing to have the reviewer comments, author rebuttal letters, and editorial decision letters published as a Supplementary item. When you submit your final files please clearly state in your cover letter whether or not you would like to participate in this initiative. Please note that failure to state your preference will result in delays in accepting your manuscript for publication.

Cover suggestions

As you prepare your final files we encourage you to consider whether you have any images or illustrations that may be appropriate for use on the cover of Nature Cell Biology.

34Please submit your suggestions, clearly labeled, along with your final files. We'll be in touch if more information is needed.

Nature Cell Biology has now transitioned to a unified Rights Collection system which will allow our Author Services team to quickly and easily collect the rights and permissions required to publish your work. Approximately 10 days after your paper is formally accepted, you will receive an email in providing you with a link to complete the grant of rights. If your paper is eligible for Open Access, our Author Services team will also be in touch regarding any additional information that may be required to arrange payment for your article.

Please note that *Nature Cell Biology* is a Transformative Journal (TJ). Authors may publish their research with us through the traditional subscription access route or make their paper immediately open access through payment of an article-processing charge (APC). Authors will not be required to make a final decision about access to their article until it has been accepted. Find out more about Transformative Journals

Please use the following link for uploading these materials:
[REDACTED]

If you have any further questions, please feel free to contact us.

Best regards,

Ziqian Li
Editorial Assistant

35Nature Cell Biology

On behalf of

Jie Wang, PhD
Senior Editor
Nature Cell Biology

Tel: +44 (0) 207 843 4924
email: jie.wang@nature.com

Reviewer #1:

Remarks to the Author:

The authors have addressed all the comments from the previous line of review in a satisfactory manner. I can now recommend publication in Nature Cell Biology

Reviewer #2:

Remarks to the Author:

Nature Cell Biology manuscript number NCB-F45367A by Beucher et al. (Jorge)
"HASTER is a cis-acting transcriptional stabilizer of HNF1A" (revised title)

In this paper, the authors use both mouse and human models to examine HASTER, finding HASTER promoter, but not the act of HASTER transcription or the lncRNA HASTER, exerts both activating and inhibitory functions to fine tune HNF1A expression. They also find Haster mutant pancreatic β cells consequently showed variegated HNF1A overexpression or silencing, causing insulin-deficiency and diabetes.

As I noted in my previous review, this paper addresses an interesting and timely topic. The development of multiple mice models used in the paper to examine the biological function and the corresponding molecular mechanisms of the HASTER promoter is a strength of the paper.

I also noted the following that needed improvement: (1) lack of underlying molecular mechanism to explain the variegated expression pattern of HNF1A in a homozygous HASTER knockout pancreatic endocrine context, (2) more description and discussion to support mechanistic conclusions, (3) extra unbiased scRNA-seq data analysis to strengthen the statement, (4) appropriate UMI-4C experiments to clarify the proposed model.

In their revised version of the paper, the authors have addressed many of these concerns. Key improvements include:

- Additional UMI-4C experiments validating the functional importance of the HASTER promoter to inhibit HNF1A transcription;
- The proposed enhancer-like function of HASTER to explain the activation function of HASTER.

36- Stronger connection between the experimental data and the proposed mechanism. This includes extra UMI-4C experiments and careful explanation of experimental data from scRNA-seq.

Overall, I think the authors have done a good job addressing my concerns. At this point, in my opinion, the novelty of conclusions outweigh any remaining deficiencies in the experimental design and presentation.

Reviewer #3:

Remarks to the Author:

The authors have addressed all concerns made in my original review, and clarified the function of HASTER as a cis regulatory element. I would like to congratulate the authors on a very carefully performed and interpreted study. Great work!

Final Decision Letter

Dear Jorge,

I am pleased to inform you that your manuscript, "HASTER lncRNA promoter is a cis-acting transcriptional stabilizer of HNF1A", has now been accepted for publication in Nature Cell Biology. Congratulations to you and the whole team!

37If you have any questions about our publishing options, costs, Open Access requirements, or our legal forms, please contact ASJournals@springernature.com

Please note that *Nature Cell Biology* is a Transformative Journal (TJ). Authors may publish their research with us through the traditional subscription access route or make their paper immediately open access through payment of an article-processing charge (APC). Authors will not be required to make a final decision about access to their article until it has been accepted. Find out more about Transformative Journals

If you have not already done so, we strongly recommend that you upload the step-by-step protocols used in this manuscript to the Protocol Exchange (www.nature.com/protocolexchange), an open online resource established by Nature Protocols that allows researchers to share their detailed experimental know-how. All uploaded protocols are made freely available, assigned DOIs for ease of citation and are fully searchable through nature.com. Protocols and Nature Portfolio journal papers in which they are used can be linked to one another, and this link is clearly and prominently visible in the online versions of both papers. Authors who performed the specific experiments can act as primary authors for the Protocol as they will be best placed to share the methodology details, but the Corresponding

38Author of the present research paper should be included as one of the authors. By uploading your Protocols to Protocol Exchange, you are enabling researchers to more readily reproduce or adapt the methodology you use, as well as increasing the visibility of your protocols and papers. You can also establish a dedicated page to collect your lab Protocols. Further information can be found at www.nature.com/protocolexchange/about

With kind regards,
Stelios

Stylios Lefkopoulos, PhD
He/him/his
Associate Editor
Nature Cell Biology
Springer Nature
Heidelberger Platz 3, 14197 Berlin, Germany

E-mail: stylios.lefkopoulos@springernature.com
Twitter: @s_lefkopoulos

** Visit the Springer Nature Editorial and Publishing website at www.springernature.com/editorial-and-publishing-jobs for more information about our career opportunities. If you have any questions please click here.**